# Relatron: Automating Relational Machine Learning over relational databases

**Zhikai Chen**[1,*] **Han Xie**[2]**, Jian Zhang**[2]**, Jiliang Tang**[1]**, Xiang Song**[2]**,**
**Huzefa Rangwala**[2,†]
[1]Michigan State University  [2]Amazon

## Abstract

Predictive modeling over relational databases (RDBs) powers applications in various domains, yet remains challenging due to the need to capture both cross-table dependencies and complex feature interactions. Recent Relational Deep Learning (RDL) methods automate feature engineering via message passing, while classical approaches like Deep Feature Synthesis (DFS) rely on predefined non-parametric aggregators. Despite promising performance gains, the comparative advantages of RDL over DFS and the design principles for selecting effective architectures remain poorly understood. We present a comprehensive study that unifies RDL and DFS in a shared design space and conducts large-scale architecture-centric searches across diverse RDB tasks. Our analysis yields three key findings: (1) RDL does not consistently outperform DFS, with performance being highly task-dependent; (2) no single architecture dominates across tasks, underscoring the need for task-aware model selection; and (3) validation accuracy is an unreliable guide for architecture choice. This search yields a curated model performance bank that links model architecture configurations to their performance; leveraging this bank, we analyze the drivers of the RDL–DFS performance gap and introduce two task signals—RDB task homophily and an affinity embedding that captures size, path, feature, and temporal structure—whose correlation with the gap enables principled routing. Guided by these signals, we propose Relatron, a task embedding-based meta-selector that first chooses between RDL and DFS and then prunes the within-family search to deliver strong performance. Lightweight loss-landscape metrics further guard against brittle checkpoints by preferring flatter optima. In experiments, Relatron resolves the "*more tuning, worse performance*" effect and, in joint hyperparameter–architecture optimization, achieves up to 18.5% improvement over strong baselines with $10\times$ lower computational cost than Fisher information–based alternatives. Our code is available at `https://github.com/amazon-science/Automating-Relational-Machine-Learning`.

## 1 Introduction

Relational databases (Codd, 2007; Harrington, 2016) have served as the foundation of data management for decades, organizing interconnected information through tables, primary keys, and foreign keys (Harrington, 2016). Their support for data integrity, consistency, and complex SQL queries has made them essential across healthcare (White, 2020; Johnson et al., 2016), academic research (Melvin, 2025), and business applications (Stroe, 2011). However, as data volume and complexity grow, traditional analytics fall short, creating demand for machine learning to identify patterns, automate decisions, and generate scalable insights. The conventional approach requires practitioners to manually export and flatten relational data into single tables through custom joins and feature engineering (Lam et al., 2017) before applying tabular ML methods.

At the macro level, two lines of work aim to reduce manual flattening and feature engineering in RDBs: (i) deep feature synthesis (DFS) (Kanter & Veeramachaneni, 2015) and (ii) relational

---

*Work done while interning at Amazon. Email: `chenzh85@msu.edu`
†Corresponding author.

deep learning (RDL) (Robinson et al., 2024; Fey et al., 2024). Both operate on heterogeneous entity–relation graphs induced from the underlying database schema, where *rows* are represented as nodes typed by their tables and foreign-key links are represented as typed edges. DFS programmatically composes relational primitives (e.g., aggregations along join paths) to produce a single feature table on which a standard tabular learner is trained. RDL trains graph neural networks (GNNs) (Kipf & Welling, 2017; Hamilton et al., 2017) end-to-end on this heterogeneous graph, learning task-specific aggregations via message passing. Empirically, both families exploit the relational structure and can surpass relation-agnostic baselines on several RDB benchmarks (Wang et al., 2024a; Robinson et al., 2024).

However, no comprehensive comparison exists between these paradigms to clarify when each performs better or their relative advantages for different task types. [1] Practitioners currently lack principled guidelines for choosing between DFS and RDL when tackling relational database prediction tasks. Additionally, methods for selecting specific design components—such as message-passing functions in RDL or tabular models in DFS—remain largely unexplored. These gaps make architecture selection for RDB tasks a labor-intensive process that relies heavily on expert knowledge.

**Design space and evaluation.** To bridge this gap, we first propose a representative design space for RDL and DFS: for the former, we decompose models into (1) feature encoding/augmentation, (2) message passing, and (3) task-specific readouts; for DFS, we use non-parametric feature engineering paired with a tabular model. We conduct an architecture-centric search—a grid over architecture choices with sampled hyperparameters—to build a performance bank. Key findings: (1) Brute-force search outperforms from-scratch RDL baselines, validating the proposed design space. (2) At the macro level (DFS vs RDL), RDL wins on more tasks (though both have distinct strengths); at the micro level (fine-grained model architectures), neither family has a single best design. (3) Validation performance can be unreliable for selection, leading to degraded test performance.

**Automatic architecture selection.** We identify factors that drive the performance gap and use them to design the **Relatron**, an architecture selector with strong test generalization. We introduce an RDB-task homophily metric that correlates strongly with the performance gap between DFS and RDL, further enriched with training-free affinity embeddings that capture task table size, structural affinity, and temporal dynamics. We further observe that the generalization behavior of configurations (validation-selected vs. test-selected) is reflected in the loss landscape geometry. Accordingly, we propose a landscape-derived metric for more reliable post-selection. Combined, our pipeline performs strongly on real-world RDB tasks, matching or exceeding prior methods (Cao et al., 2023; Achille et al., 2019) in task-embedding quality and in predicting whether RDL or DFS is preferable; for joint hyperparameter and architecture search, it outperforms strong baselines, including search-based and task-embedding-based ones (Cao et al., 2023; Bischl et al., 2023), while using up to 10x less compute resources than task-embedding-based methods.

Our contributions can be summarized as follows.

1. We propose a representative model design space for RDB predictive tasks, featuring promising performance, and generate a model performance bank that links model architecture configurations to task performance for future research.
2. Based on a comprehensive search on the model design space, we point out the limitations of RDL, and propose a routing method to select between RDL and DFS for RDB predictive tasks automatically. Furthermore, we analyze the factors that drive the performance gap between RDL and DFS, and these insights can inspire further research, such as the development of relational foundation models.
3. Through extensive experiments, we validate the effectiveness of our pipelines in tasks such as predicting proper architectures and joint hyperparameter-architecture search.

## 2 RELATED WORK AND BACKGROUND

In this section, we present related works necessary for understanding the following paper contents, and put other related works in Appendix D.

---

[1] The graph machine learning models studied in Wang et al. (2025b) differ from RDL (Robinson et al., 2024) in implementation details, discussed further in Appendix E.1.

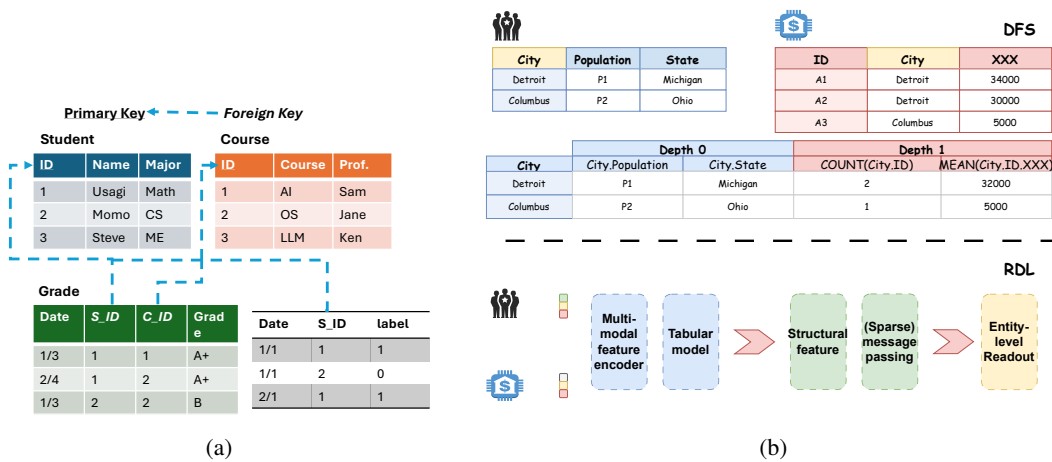

Figure 1: (a) An example of generating the task table from an RDB. The label is based on whether a student has achieved an A+ in a course before a specific timestamp. (b) Another example demonstrating the working process of DFS and RDL. For DFS, a predefined set of aggregation functions, such as MEAN and COUNT, is used to aggregate information across multiple tables based on key relationships into a final data table. For comparison, RDL is claimed to replace the manual aggregation design with an automatic message-passing-based sparse attention.

**Relational Deep Learning (RDL).**[2] (Robinson et al., 2024; Fey et al., 2024) applies graph machine learning to relational databases. RDB prediction has three key traits (Figure 1): (1) time is first-class—labels are split and conditioned on time; (2) labels are defined by time-constrained SQL over arbitrary column combinations; (3) heterogeneous column types make feature interactions richer than in text-attributed or non-attributed graphs. These choices mirror real industrial settings. Wang et al. (2024a; 2025b) offer a view closer to traditional heterogeneous GNNs; although Wang et al. (2025b) reports results on Relbench (Robinson et al., 2024), the modeling and evaluation setups differ. We therefore re-implement all methods in the unified framework for fair comparison. Some recent RDL works center on specialized models for RDL, including higher-order message passing (Chen et al., 2025a) and recommendation (Yuan et al., 2024). Transformers and LLMs (Dwivedi et al., 2025; Wu et al., 2025; Wydmuch et al., 2024) have been tested, but are resource-heavy with modest gains. Beyond RDB-specific studies, broader graph foundation model and graph-LLM literature provides useful context on transferable graph pre-training and prompting paradigms (Mao et al., 2024; Chen et al., 2023; Wang et al., 2025a). Foundation models include Griffin (Wang et al., 2025b), which uses cross-table attention yet often fails to beat GNNs, and KumoRFM (Fey et al., 2025), a graph transformer with strong performance and in-context learning, though details remain undisclosed. In this paper, we focus on efficient models training from scratch, leaving foundation models for future work.

**Deep Feature Synthesis (DFS).** Compared to RDL, DFS (Kanter & Veeramachaneni, 2015) is an often-overlooked approach that aggregates cross-table information into a single target table via automated feature engineering (Zhao et al., 2020; Lam et al., 2017; 2018). It underpins commercial systems such as getml[3]. Given a target table and a schema graph, DFS traverses foreign-key–primary-key links and composes type-aware primitives into feature definitions. Transform primitives operate on single columns, while aggregation primitives (e.g., statistics such as MAX, MIN, MODE) summarize sets of related rows; compositions along schema paths yield higher-order features. For time-indexed tasks, DFS evaluates every recipe under a per-row cutoff time, ensuring that only information available in the past contributes to the feature value, thereby avoiding leakage.

## 3   DESIGN SPACE OF MODEL ARCHITECTURES OVER RDB

Architecture selection begins by giving an architecture design space. This section introduces the task and design space, then presents evaluation results and observations from the exploration.

---

[2]RDL denotes both the learning paradigm and the problem setting; we use RDL for the former and RDB for the latter.

[3]https://getml.com/latest/

## 3.1 PREDICTIVE TASKS ON RDBS

**Problem definition.** A relational database (RDB) is a tuple $\mathcal{D} = (\mathcal{T}, \mathcal{L})$, where $\mathcal{T} = \{T_1, \ldots, T_n\}$ is a set of tables and $\mathcal{L} \subseteq \mathcal{T} \times \mathcal{T}$ is a set of links between them. Each table $T_i \in \mathcal{T}$ consists of rows (entities) $\{v_1, \ldots, v_{m_i}\}$. Links are related to primary keys (PKs) and foreign keys (FKs). A PK $p_v$ uniquely identifies a row, while a FK establishes a link to a row in another table by referencing its PK. Each row also has a set of non-key attributes, $x_v$, and an optional timestamp, $t_v$. A temporal predictive task $\Pi_{t_{pred}}$ with respect to time $t_{pred}$ can be defined over two granularities. *Entity-level prediction* learns a function $f : \mathcal{D}_{t_{pred}} \times V_{target} \to \mathcal{Y}$ that maps entities from a target set $V_{target} \subseteq T_i$ to a label space $\mathcal{Y}$. *Link-level prediction* determines the existence of a link between two entities, $v_i \in T_i$ and $v_j \in T_j$, at time $t_{pred}$ by learning a function $f : \mathcal{D}_{t_{pred}} \times T_i \times T_j \to \{0, 1\}$. RDB tables can be categorized into fact tables and dimension tables. A fact table stores events or transactions (e.g., purchases, clicks, race results) with many rows, and each typically carries foreign keys to several entities. A dimension table stores descriptive attributes about these entities (e.g., customer, product, time, circuit/driver), typically with one row per entity/state, and a PK that is referenced by facts.

**Graph perspective of RDB (Robinson et al., 2024).** Each RDB and a corresponding predictive task can be viewed as a temporal graph $\mathcal{G}_{(-\infty, T]} = (V_{(-\infty, T]}, \mathcal{E}_{(-\infty, T]}, \phi, \psi, f_V, f_E)$, paired with task labels $Y$. $V_{(-\infty, T]}$ and $\mathcal{E}_{(-\infty, T]}$ are entities and links at time $t \leq T$. $\phi$ maps each entity to its node type, $\psi$ maps each link to its link type. $f_V$ and $f_E$ are the mappings of features.

**Datasets and tasks.** We consider a diverse set of datasets and tasks from recent works (Robinson et al., 2024; Wang et al., 2024a; Chen et al., 2025b). Adopting the taxonomy from Robinson et al. (2024), we categorize these tasks into four types: entity classification, entity regression, recommendation, and autocomplete. Entity-level tasks (classification and regression) involve predicting entity properties at a given time $t_{pred}$. Recommendation tasks focus on ranking the relevance between pairs of entities at $t_{pred}$. The autocomplete task involves predicting masked information in table columns. A comprehensive description of each dataset and task is available in Appendix B. We illustrate the generation of an entity-level task in Figure 1.

## 3.2 MODEL ARCHITECTURE DESIGN SPACE

**Architecture choice.** As shown in Table 1, to enable a fair comparison between DFS and RDL on RDB benchmarks, we construct a compact, factorized design space for each family. **RDL** models are built from three modules: (i) a structural-feature encoder (partial labeling trick, learnable embedding, or no augmentation) (Yuan et al., 2024; Zhu et al., 2021), (ii) a message-passing network (PNA, HGT, SAGE, or RelGNN) (Corso et al., 2020; Hu et al., 2020b; Robinson et al., 2024; Chen et al., 2025a), and (iii) a readout head (MLP, ContextGNN, or a shallow aggregator) (Yuan et al., 2024). Standard training hyperparameters such as learning rate, dropout, normalization, and neighbor fanout are also tuned. **DFS** methods are parameterized by three main knobs: the SQL-level aggregation function (e.g., max/min/mode), the number of aggregation layers (1–3 when supported), and the backbone model (TabPFN, LightGBM, or FT-Transformer), with batch size and hidden dimension included as additional hyperparameters. Architectural components are explored via grid search, while other hyperparameters are sampled from a smaller space (see Appendix E.3).

**Design motivation.** While it is not feasible to exhaustively cover all designs in graph machine learning (GML), our design space spans representative components. For message passing alone, this includes vanilla message passing, self-attention mechanisms, multi-aggregator schemes, and higher-order approaches. Importantly, classical GML architectures gain renewed significance in RDB tasks. For example, PNA, originally devised for molecular graphs, is well-suited for RDB tasks: its multi-aggregation mechanisms naturally capture diverse feature interaction patterns, echoing the strengths of DFS-based approaches. In Appendix E.3, we provide a more detailed description of each component and explain why some GML designs are not suitable for RDB tasks.

Table 1: Search space of RDL and DFS-based methods for RDB tasks. Underline means these components will go over a grid search, while other components will be sampled.

| Name | Architecture design space | | | Hyper-parameters |
|---|---|---|---|---|
| **RDL** | Structural feature | Message passing | Readout | Learning rate, dropout, normalization, fanout... |
| | Labeling ID, Learnable embedding, None | PNA, HGT, Sage, RelGNN | MLP, ContextGNN, Shallow | |
| **DFS** | Aggregation function | Aggregation layers | Backbone | Batch size, hidden dimension |
| | Max, Min, Mode, ... (fixed) | 1, 2, 3 (if possible) | TabPFN, LightGBM, FT-Transformer | ... |

### 3.3 EMPIRICAL STUDY OF VARIOUS ARCHITECTURE DESIGNS

**Evaluation setup.** For entity-level tasks, we sample 15 configurations per architecture combination (180 per task). For recommendation, we sample 10 configurations. For DFS, we use the Robinson et al. (2024) HPO utility with 20 trials per design (TabPFN requires no tuning). Following Robinson et al. (2024), we train up to 20 epochs, capping each epoch at 1,000 steps (recommendation) or 500 (entity-level), using Adam (Kingma & Ba, 2015) with an optional exponential LR scheduler. Efficiency is not our main focus, and the only efficiency constraint is that the model can fit a single L40S GPU (48GB). See Appendix E.6 for more discussions on extending our pipelines to efficiency-aware scenarios. In this section, we only report entity-level results. The recommendation results are presented in Appendix E.5 since the architecture choice there is less important. Our evaluation reveals the following key insights:

**Observation 1. Efficacy and necessity of the design space.** We first validate our proposed design space by examining **the best possible test performance**. We compare our design space's performance with that of baseline models reported in the literature, including GraphSAGE (Robinson et al., 2024), RelGNN (Chen et al., 2025a), RelGT (Dwivedi et al., 2025), KumoRFM (Fey et al., 2025), and RelLLM (Wu et al., 2025) on 17 Relbench (Robinson et al., 2024) tasks. These strong baselines serve to highlight that our design space achieves competitive results. As shown in

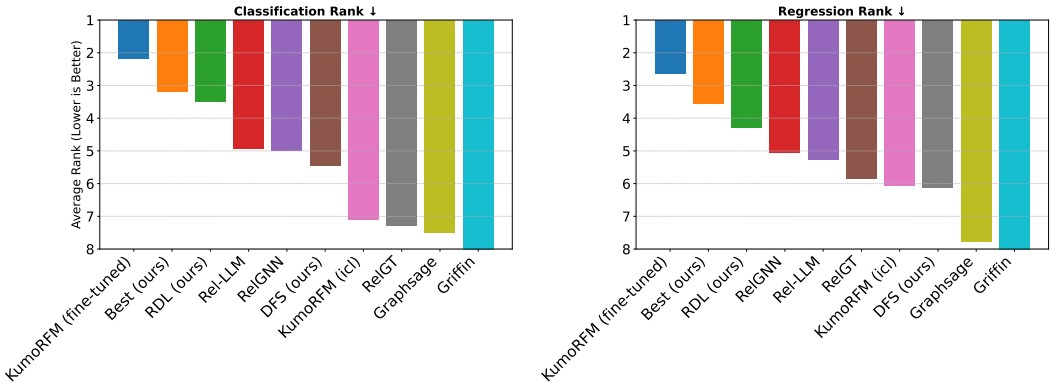

Figure 2: Performance comparison between the best configurations from our design space and baseline models on entity-level tasks. "Best (ours)" means the better value of RDL and DFS. Full numerical results can be seen in Table 11 from Appendix E.5.

Figure 2, the best configurations from our design space consistently outperform all scratch-trained baselines. Specifically, we find the following two designs improving the performance of RDL and DFS: (1) **Breaking equivariance:** learnable embeddings (e.g., partial labeling Zhu et al. (2021)) improve entity-level performance despite violating node-level permutation equivariance; (2) **Better DFS implementation:** compressing dense embeddings via incremental PCA and inserting them as numeric columns enables efficient cross-table aggregation in DFS (Wang et al., 2025b), markedly improving scalability and accuracy. These performance results justify architecture selection: **no single architecture dominates across tasks**. RDL vs. DFS rankings vary by task, and the same holds for micro-level choices (e.g., message passing; see Appendix E.5).

**Observation 2. Validation metrics can be unreliable for architecture selection.** Oracle test selection shows the upper bound of our design space, but in practice, configurations are picked by validation (or training) scores, which often leads to a gap: validation-selected models underperform test-selected ones (Table 2). In

Table 2: The performance gap between validation- and test-selected configurations.

| Task | Model | reported perf | test-selected perf | val-selected perf |
|---|---|---|---|---|
| driver-top3 (auroc) | GraphSAGE | 75.54 | 82.81 | 81.56 |
| | RelGNN | 85.69 | 85.69 | 82.61 |
| driver-position (mae) | GraphSAGE | 4.022 | 3.91 | 3.93 |
| | RelGNN | 3.798 | 3.80 | 4.35 |
| user-ignore (auroc) | GraphSAGE | 81.62 | 86.40 | 72.27 |
| | RelGNN | 86.18 | 86.18 | 78.94 |

our reproduction of RelGNN (Chen et al., 2025a), the reported gains over GraphSAGE are clear only when choosing hyperparameters by test performance; with validation selection, the advantage becomes marginal. This reliability issue is largely overlooked in graph AutoML and only briefly noted in tabular ML (Ye et al., 2024). It is pronounced in RDB settings, which are inductive and time-

aware: temporal splits induce distribution shift between validation and test periods. The problem affects both RDL and DFS; further evidence is in Appendix E.5.

## 4 PRINCIPLES AND AUTOMATION OF ARCHITECTURE SELECTION

Building on Section 3, where neither paradigm uniformly dominates, we seek a principled way to choose architectures for an RDB task. Two obstacles arise: (i) the design space is huge (180 trials per task cover only a small fraction), and (ii) validation performance—the usual selection proxy—can be unreliable, so more search can even degrade test performance. We address this by leveraging the *model performance bank* (Section 3.3): for a new task, transfer information from similar tasks to reduce the search space. This requires a task embedding to capture the properties of tasks.

### 4.1 FROM OBSERVATIONS TO TASK EMBEDDINGS

We begin with two observations from the performance bank: (1) RDL–DFS performance gaps vary across tasks; and (2) validation-selected vs. test-selected performance gaps differ across model types. The first implies data factors driving a task's affinity for certain model classes; the second motivates analyzing model properties to explain generalization.

#### 4.1.1 DATA-CENTRIC PERSPECTIVE

We begin with *homophily* as the first axis of task characterization, since it is the lowest-order relational signal and reflects a task's favored inductive bias (Ma et al., 2022). Label-induced properties empirically outperform label-agnostic ones (e.g., degree) for performance prediction (Li et al., 2023; Zheng et al., 2024; Mao et al., 2023), and labels are directly available in RDB tasks via the materializing SQL query. Extending homophily to RDB tasks is non-trivial because: (1) labels evolve over time; (2) labels may be continuous; and (3) the PK–FK graph is schema-driven—labels usually come from a single fact table, so naively computing edge homophily on raw PK–FK links is ill-posed (always equal 1). We therefore propose **RDB task homophily**. Starting from the PK–FK graph used for training, we temporally aggregate labels to per-entity means $\hat{y}_v \in \mathbb{R}^C$, then augment the graph as in Figure 3 to form self-looped metapaths; for scalability, we restrict to one-hop metapaths.

**Definition 1** (RDB task homophily)**.** *Given an augmented heterogeneous graph $\mathcal{G} = (V, \mathcal{E})$ induced from an RDB task, labeled entity type $\mathsf{F}$ and $V_\mathsf{F}$ its nodes, each with mean label $\hat{y}_v$. Let $\mathcal{M}$ be a finite set of self-looped metapaths $m$ starting and ending with $\mathsf{F}$, and let $\mathcal{E}_m$ be the set of edges induced by $m$. Given a label metric $\mathcal{K}$, the* RDB task homophily *for metapath $m$ is*

$$H(\mathcal{G}; m) = \frac{1}{|\mathcal{E}_m|} \sum_{\{u,v\} \in \mathcal{E}_m} \mathcal{K}(\hat{y}_u, \hat{y}_v).$$

**Label metric design.** For classification tasks, the label metric can be the dot product $\mathcal{K}(\hat{y}_u, \hat{y}_v) = \hat{y}_u^\top \hat{y}_v$, which reduces to traditional edge homophily $\mathbf{1}\{\hat{y}_u = \hat{y}_v\}$ when there are no duplicate entities in the task table. We denote this measure by $H_{\text{edge}}(\mathcal{G}; m)$. For regression tasks, we instead use a correlation-based label metric: letting $\tilde{y}_u = (\hat{y}_u - \mu)/\sigma$ denote standardized predictions over labeled nodes (with empirical mean $\mu$ and variance $\sigma^2$), we define $\mathcal{K}(\hat{y}_u, \hat{y}_v) = \tilde{y}_u \tilde{y}_v = \frac{(\hat{y}_u - \mu)(\hat{y}_v - \mu)}{\sigma^2}$ to measure Pearson-style correlation of labels along edges of each metapath. We may further extend the homophily definition to account for class imbalance. A notable extension is the adjusted homophily (Platonov et al., 2023). For a classification task, it can be defined as

$$H_{\text{adj}}(\mathcal{G}; m) = \frac{H_{\text{edge}}(\mathcal{G};m) - \sum_{k=1}^{C} \left( \frac{D_k^{(m)}}{2|\mathcal{E}_m|} \right)^2}{1 - \sum_{k=1}^{C} \left( \frac{D_k^{(m)}}{2|\mathcal{E}_m|} \right)^2}, \text{ where } D_k^{(m)} \text{ is the}$$

degree of class $k$. To obtain a global measure across the whole graph, we aggregate over metapaths using statistical functions such as MEAN and STD.

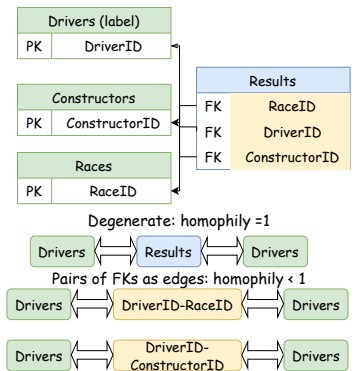

Figure 3: Augmenting REL-F1 databases. We should treat the set of FKs as a hyperedge (each pair of FKs is appended to the original PK-FK graphs as a new edge type) rather than relying solely on PK-FK edges.

**Anchor-based affinity properties.** Homophily is most suited to node-level equivariant message-passing and overlooks path-based models with labeling tricks and non-structural factors (feature quality, temporal dynamics). We therefore add anchor-based affinities to estimate which model families a task favors. If random path-based aggregation already separates labels well, path-based models should excel. See Appendix C for the relationship between random hashing and randomly initialized NBFNet. Compared to task embeddings requiring backpropagation (Cao et al., 2023), these features only require a forward pass and closed-form fitting, which is much more efficient.

1. **Path/neighborhood affinity.** Use randomly initialized GraphSAGE and NBFNet (Zhu et al., 2021) as random hashers; after one forward pass, fit a closed-form linear head (ridge or LDA). A random NBFNet achieves $> 82$ AUROC on USER-BADGE, indicating strong path affinity.
2. **Feature affinity.** Use TabPFN validation performance (no training) as a proxy for feature quality.
3. **Temporal affinity.** Since mean-label homophily ignores time, add simple timeline statistics (e.g., majority label over time), which effectively capture dynamics (Cornell et al., 2025).

**Number of training rows.** Among basic RDB statistics, the number of training rows $N_{\text{train}}$ plays a uniquely important role. Unlike other dataset statistics (e.g., the number of test rows or table counts), $N_{\text{train}}$ directly determines the amount of context a model can access during learning. This is especially critical for foundation-model-based approaches such as DFS+TabPFN, where $N_{\text{train}}$ governs how much historical information the model can attend to in its context window. We therefore include $\log(N_{\text{train}})$ as a dedicated feature alongside the affinity properties.

**Correlating heuristics with the RDL-DFS performance gap.** We conduct a nonparametric correlation analysis relating RDB task characteristics to the performance differential between the best RDL and DFS models. We identify that $\log(N_{\text{train}})$, representing the logarithm of training row counts, and adjusted homophily are the most significant predictors. Notably, in classification tasks, adjusted homophily displays a strong negative correlation with the RDL-DFS gap, with Spearman's $\rho = -0.43$ ($p < 0.05$). (See Appendix C for complete heuristic definitions). This implies that RDL's nonlinear aggregation is particularly advantageous for low homophily tasks. Moreover, RDL requires substantial supervision signals to learn appropriate aggregation functions. We evaluate other heuristics, including the influence of labeling tricks and path affinity, in Appendix E.5.

**(Informal) Theoretical insights.** The correlation between the relative performance of RDL and DFS, homophily-induced features, and the train task table size (number of available labels) can be elucidated from a graph-theoretical perspective. If we formulate the RDB as a heterogeneous graph and expand it into a multi-relational graph via metapaths, both DFS and RDL can be viewed as mechanisms that aggregate neighbor information into a score for each task-table row. However, DFS relies on fixed linear averages of neighbor signals, whereas RDL learns relation-specific nonlinear transformations (e.g., amplification, clipping, or sign flips) prior to combination. We briefly discuss two regimes here and provide rigorous proofs in Appendix C.2.

1. **Low-homophily classification.** When labels are strongly homophilous along most metapaths, simple averaging is close to optimal and DFS already captures most relational signal. When some metapaths are weakly homophilous, heterophilous, or nearly random, linear averaging mixes positive and negative evidence and tends to cancel signal. RDL can instead learn to down-weight uninformative relations and flip the contribution of systematically "opposite-label" metapaths, effectively increasing the signal-to-noise ratio exactly when adjusted homophily is small or negative. This explains why the RDL–DFS gap grows on low-homophily classification tasks.
2. **Dependence on the number of training rows.** The same view clarifies why the number of training rows, $N_{\text{train}}$, is a key moderator. DFS has a relatively small hypothesis class: aggregators are fixed and only a tabular model is learned, so its estimation error shrinks quickly with sample size. RDL introduces many additional parameters (per-relation weights, gates, nonlinearities), which in principle allow it to recover the "right" aggregation but also make it easier to overfit when supervision is scarce. In low-data regimes, these learned gates are noisy and can hurt performance compared to the stable DFS averages; once $\log(N_{\text{train}})$ is large enough, the gates can be estimated reliably and RDL's extra flexibility turns into a consistent advantage.

### 4.1.2 MODEL-CENTRIC PERSPECTIVE

To understand validation-test selection gaps, we analyze checkpoints that exhibit good and poor generalization. After conducting intuitive visualization-based analysis (shown in Appendix E.5), we

probe generalization via the local loss landscape $L : \mathbb{R}^d \to \mathbb{R}$ around a checkpoint $w_0$ (Chiang et al., 2023). This is also consistent with recent evidence that mode connectivity in GNNs is informative for understanding optimization geometry and model robustness (Li et al., 2025). Fix an orthonormal 2D subspace $\Pi = \mathrm{span}(e_1, e_2)$ and sample a grid $\Gamma = \{(s_i, t_j)\} \subset [-\rho, \rho]^2$. Each grid point defines $w_{ij} = w_0 + s_i e_1 + t_j e_2$ with $L_{ij} = L(w_{ij})$. We summarize with three indicators spanning increasing smoothness scales (Garipov et al., 2018; Li et al., 2018a; Ghorbani et al., 2019):

1. **First-order** $P_1$: $\max_{|i-k|+|j-l|=1} \frac{|L_{ij}-L_{kl}|}{\sqrt{(s_i-s_k)^2+(t_j-t_l)^2}}$ (worst finite-difference slope on $\Pi$), detecting brittle checkpoints near loss cliffs.
2. **Second-order** $P_2$: $\lambda_{\max}(H_\Pi(w_0))$, where $H_\Pi(w_0) = E^\top \nabla^2 L(w_0)E$, $E = [\,e_1 \ e_2\,]$ (sharpness along $\Pi$), measuring basin curvature; high $P_2$ signals narrow minima sensitive to distribution shift (Li et al., 2018a).
3. **Energy barrier** $P_{\mathrm{bar}}$: $\max_{(i,j)} \max_{t \in [0,1]} L(w_0 + t(w_{ij} - w_0)) - \max\{L(w_0), L_{ij}\}$ (barrier to departing $w_0$ along rays within $\Pi$), probing global basin geometry rather than local curvature.

These three metrics span local slope, curvature, and global basin shape, and can occasionally conflict (e.g., moderate $P_1$ but high $P_2$ when the surface curves steeply beyond the immediate neighborhood). On DRIVER-TOP3 (Table 3), when validation–test gaps are large all indicators consistently favor the test-optimal checkpoint; when gaps are small they may disagree, motivating their joint use. These metrics are comparable within a model family (RDL or DFS) but not across families due to scale differences, and are effective only for well-fitted models. Since these signals are post-hoc, we use them to refine the final checkpoint choice among top-validation candidates.

### 4.1.3 AUTOMATIC ARCHITECTURE SELECTION THROUGH RELATRON

Based on these findings, we introduce **Relatron**, an architecture selector that maps *task embeddings* to *meta-predictions* about which model design to use. Given an RDB task, Relatron considers two types of architecture selection.

Table 3: Example landscape properties and model performance. Smaller values typically indicate a more benign landscape.

| Selection | Model type | Val_auroc | Test_auroc | $P_{bar}$ | $P_1$ | $P_2$ |
|---|---|---|---|---|---|---|
| Val | RDL | 89.48 | 82.41 | 2.77 | 1.49 | 4.23 |
| Test | RDL | 86.05 | **85.94** | **0.41** | **1.22** | **0.22** |
| Val | DFS | 83.44 | 84.69 | **0.384** | 0.041 | 1.32 |
| Test | DFS | 83.76 | **85.71** | 0.495 | **0.03** | **0.50** |

**Macro-level selection (RDL vs. DFS).** We train a meta-classifier on the performance bank to map task embeddings to the empirically winning family, using homophily-based task embeddings. At inference, we (1) compute the novel task's embedding and (2) apply the meta-classifier to choose between RDL and DFS.

**Joint architecture selection and HPO with a query budget.** For standard HPO with a query-budget setting, the query budget is appended to the task embedding as an additional feature. (1) A macro-level meta-predictor first chooses between RDL and DFS. (2) An HPO routine (e.g., TPE (Bergstra et al., 2011) or Autotransfer (Cao et al., 2023)) generates candidate checkpoints within the selected family. (3) Loss-landscape metrics are applied for post-selection among candidates with top validation performance. An insight here is that the favored model type is related to the query budget. Although RDL often attains higher best-case performance on tasks such as STUDY-OUTCOME, under tight search budgets, its average performance can lag behind DFS because good RDL configurations are harder to find. Moreover, as shown in Section 4.2, we surprisingly find that the macro-level meta-predictor (DFS or RDL) addresses most issues: after selecting the appropriate model branch, search efficiency improves and the validation–test gap narrows.

**Design space of task features.** In terms of task feature design, there are three categories, with computation budgets ranging from small to large. A detailed introduction can be found in Appendix E.

1. **Model-free embeddings**: Model-free embedding requires no training and is extremely fast to compute. This includes homophily-based features, the performance of simple heuristic baselines, basic database statistics, and temporal-related correlations.
2. **Training-free Model-based embeddings**: These embeddings use training-free models' performance as embeddings. This includes the performance of DFS-TabPFN and that of randomly initialized GraphSAGE or NBFNet.
3. **Anchor-based embeddings (Cao et al., 2023)**: This refers to task2vec (Achille et al., 2019)-based anchor model-based embeddings, such as Auto-transfer. Though the original paper claims

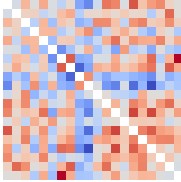

Figure 4: Ground truth GraphGym similarity

Table 4: Experimental results for task-embedding similarity, leave-one-out (LOO) accuracy, and task-embedding computation efficiency. "AT" stands for Autotransfer. For "winner by val," we still report test performance, but the representative checkpoints are selected by validation performance.

| Task embedding design | Mean Kendall's corr (no g) | Mean Kendall's corr (g) | Winner by val | Winner by test | Average time (min) |
|---|---|---|---|---|---|
| Model-free (homophily + stats + temporal) | **0.066** | **0.163** | **87.5%** | **79.2%** | **0.48** |
| Model-free (simple heuristic performance) | 0.027 | 0.031 | 70.8% | 75.0% | 0 |
| Model-based (training-free) | -0.038 | -0.030 | 66.7% | 66.7% | 5 |
| Anchor-based (Autotransfer) | -0.049 | -0.011 | 66.7% | 66.7% | 50 |

that these embeddings require little time to obtain, we find that computing the Fisher information matrix is actually very time-consuming for RDB tasks, given the number of required anchors.

## 4.2 EXPERIMENTAL EVALUATIONS

We then evaluate the proposed Relatron on three experiments. (1) *A sanity check of task embeddings*: First, we evaluate the effectiveness of different task embeddings by comparing the task similarity calculated by task embedding and the ground truth Graphgym similarity (You et al., 2020). This experiment is mainly used to verify the correctness of task embeddings. (2) *Macro-level architecture search:* Second, we check whether task embeddings can help identify the proper architecture for an RDB task. (3) *Joint selection of architectures and hyperparameters*: Third, we consider a more practical scenario, in which task embeddings and meta-predictor are used to enhance the effectiveness of joint hyperparameter and architecture search.

**Can task embeddings reflect ground-truth task similarity?** Using all trials in the performance bank, we: (1) derive ground truth GraphGym similarity (You et al., 2020) by intersecting model configurations across tasks, ranking them by per-task performance, and defining pairwise similarity as Kendall's $\tau$ between the two rank signatures. As a sanity check, the top 3 similar tasks are USER-BADGE, USER-ENGAGEMENT, and USER-CHURN, which aligns with the similarity of their homophily metrics. (2) for each embedding, form a task–task matrix via cosine similarity on normalized features and report Kendall's rank correlation with the GraphGym matrix; (3) optionally learn a projection $g$ (as in Cao et al. (2023)) using a margin-ranking meta-objective that pulls together tasks with similar performance profiles and pushes apart dissimilar ones. Results (Table 4): Our proposed homophily and affinity-based embedding achieves the best ranking correlation. On the other hand, we also need to point out that none of these task embeddings present significantly high correlation, which can partially explain why transfer-based HPO is not very effective in Table 6.

**Can task embeddings help predict RDL vs. DFS winner?** We then investigate whether task embeddings can predict which method—RDL or DFS—performs better on a novel task. We consider winners selected by validation performance and directly extracted using test perfor-

Table 5: Ablation on performance-bank size for macro-level predictors (LOO accuracy).

| Bank size | Model-free (ours) | Training-free model-based | Anchor-based |
|---|---|---|---|
| 8 | 68.9% | 56.9% | 62.9% |
| 12 | 64.8% | 50.4% | 56.1% |
| 16 | 76.2% | 53.1% | 62.5% |
| All | 87.5% | 66.7% | 66.7% |

mance. The former one is exactly the setting for architecture selection during the HPO. For each target task, we fit the model using other tasks in the model performance bank and evaluate the target task with leave-one-out cross-validation. As shown in Table 6, our proposed model-free task features are surprisingly most effective despite the low computation cost. If we further look at the incorrect samples, we find that most of them have a small performance gap between RDL and DFS (within 2.5%), indicating that these tasks are inherently hard to distinguish. Moreover, as shown in Table 5, we also study the influence of model performance bank size. As expected, larger banks lead to better predictors since they cover more diverse tasks. At the same time, our proposed model-free task features consistently outperform other embeddings under different bank sizes.

**Joint hyperparameter and architecture search.** We evaluate our full pipeline in a joint HPO–architecture–search setting. As *baselines* for trial generation, we use random search, TPE (Bergstra et al., 2011), Hyperband (Li et al., 2018b), and Autotransfer (Cao et al., 2023). We report results on three representative datasets—DRIVER-TOP3, DRIVER-POSITION, and USER-CHURN—and use the remaining tasks as the performance bank; the first two have small training tables and are prone to overfitting. Our pipeline trains a TabPFN-based meta-predictor that selects between RDL and DFS from the query budget and homophily-based task embeddings. Conditional

Table 6: Joint architecture and hyperparameter optimization result. Best results are highlighted with an underline and the second are **bold**. "Only predictor" means only using the meta-predictor. Relbench's default refers to the default architecture and hyperparameters in their pipelines. Best fixed refers to the configurations with the best mean rank of performance across tasks.

| Strategy | Budget | driver-top3 (ROC-AUC) ↗ | | driver-position (MAE) ↘ | | user-churn (ROC-AUC) ↗ | |
|---|---|---|---|---|---|---|---|
| Random | 3 | 82.67 ± 2.19 | | 3.6810 ± 0.4255 | | 68.56 ± 1.23 | |
| | 10 | 77.80 ± 4.79 | | 3.8576 ± 0.5035 | | 68.60 ± 1.20 | |
| | 30 | 77.28 ± 2.42 | | 4.2793 ± 0.1483 | | 69.54 ± 0.43 | |
| TPE | 3 | 82.67 ± 2.19 | | 3.6810 ± 0.4255 | | 68.56 ± 1.23 | |
| | 10 | 81.45 ± 0.44 | | 3.7897 ± 0.4271 | | 68.60 ± 1.20 | |
| | 30 | 77.92 ± 5.12 | | 4.1724 ± 0.0519 | | 69.54 ± 0.43 | |
| Hyperband | 3 | 82.67 ± 2.19 | | 3.6810 ± 0.4255 | | 68.43 ± 1.33 | |
| | 10 | 80.68 ± 0.74 | | 3.7897 ± 0.4271 | | 68.60 ± 1.20 | |
| | 30 | 74.37 ± 9.59 | | 4.0948 ± 0.1420 | | 69.32 ± 0.34 | |
| Autotransfer | 3 | 77.11 ± 4.43 | | 4.2916 ± 0.0952 | | 69.09 ± 0.86 | |
| | 10 | 78.71 ± 2.88 | | 4.3645 ± 0.2105 | | 70.28 ± 0.27 | |
| | | *Only predictor* | *Full* | *Only predictor* | *Full* | *Only predictor* | *Full* |
| **Ours** | 3 | 83.80 ± 0.34 | NA | 3.3986 ± 0.0877 | NA | 68.78 ± 1.31 | NA |
| | 10 | 83.28 ± 1.45 | 83.30 ± 1.17 | 3.3934 ± 0.1389 | **3.3553 ± 0.0862** | 69.66 ± 0.89 | 69.61 ± 0.86 |
| | 30 | **84.00 ± 0.34** | 84.33 ± 0.06 | 3.3339 ± 0.1563 | 3.3339 ± 0.1563 | **70.05 ± 0.34** | 70.05 ± 0.34 |
| **DFS + TabPFN** | | 82.24 | | 3.43 | | 67.79 | |
| **Relbench's default** | | 73.19 | | 5.02 | | 68.12 | |
| **Best fixed** | | 83.72 | | 4.35 | | 69.84 | |
| **Griffin** | | 77.95 | | 4.20 | | 68.4 | |

on this choice, we run TPE—the best-performing trial generator—within the selected family, then post-select among the top-3 validation models using landscape measures.

As shown in Table 6, we observe: (1) our pipeline generally achieves better performance. Using only the meta-predictor to choose the model family already yields strong results, suggesting that large validation–test gaps often arise from selecting an unsuitable architecture. Unlike other methods that suffer from the undesirable "the more you train, the worse you get" phenomenon, our performance continues to improve as the number of trials increases. Post-hoc selection offers only limited gains, likely because hard voting over numeric landscape metrics still introduces noise. Addressing checkpoint selection may require pre-trained priors similar to architecture search; we leave this to future work. (2) For search acceleration, unlike Cao et al. (2023), embedding-based configuration retrieval is typically ineffective, with the sole exception of USER-CHURN. Consequently, we rely on TPE for trial generation. The limited effectiveness of Autotransfer suggests that RDB data distributions are more complex than those of standard graph benchmarks, and that a larger, more diverse task bank would be needed—impractical given the scarcity of public data. Synthetic tasks, in the spirit of Hollmann et al. (2023), may therefore be a promising direction to improve search quality. (3) After hyper-parameter tuning, we demonstrate that models trained from scratch can outperform foundation models like DFS with TabPFN.

## 5 CONCLUSION, LIMITATIONS, AND FUTURE DISCUSSION

In this paper, we systematically study the design space of relational machine learning models for RDB tasks and collect a model performance bank. Based on this study, we show that the advantage of RDL over DFS is related to task properties, like the RDB task homophily. Then, we propose a meta-predictor based on the model performance bank and our proposed selector, Relatron, which demonstrates promising performance in both macro- and micro-level architecture search.

**Limitations and future work.** Our study does not explore LLM-based methods, either as encoders (Wang et al., 2025b) or predictors (Wu et al., 2025). While these approaches excel on certain databases, they often perform similarly or worse than baselines on other databases, leaving their role an open question. Although we do not propose new architectures, our results highlight design insights: GNNs with labeling tricks can boost entity-level prediction, and DFS-based methods often outperform RDL, suggesting that current RDL designs may be suboptimal. Yet DFS remains a non-parametric, hand-crafted approach, contrasting with deep learning trends. Designing novel architectures inspired by DFS thus represents a promising direction.

# 6 ETHICS STATEMENT

We acknowledge that we and all co-authors of this work have read and commit to adhering to the ICLR Code of Ethics. Our study relies solely on publicly available benchmark datasets such as Relbench (Robinson et al., 2024). We believe this work raises no direct ethical risks beyond standard concerns associated with machine learning research.

# 7 REPRODUCIBILITY STATEMENT

We provide training settings in Section 3.3, Section 4, and Appendix E. To validate the observations in this paper, we include theoretical discussions in Appendix C.

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

## A   USAGE OF LARGE LANGUAGE MODELS

We utilize large language models to refine our writing and also employ a large language model-based coding agent to assist with code writing. We have reviewed the generated content provided by large language models and will be responsible for the correctness of the polished content.

## B   SUPPLEMENTARY INFORMATION FOR DATASETS AND TASKS

Our adopted datasets and tasks are summarized in Table 7. It should be noted that we skip the **rel-amazon** datasets because the source is crawled from Amazon, which contains sensitive information and does not present a valid license. We then briefly introduce each dataset and task as follows:

- `rel-hm` (Robinson et al. (2024)): This database contains comprehensive customer and product data from online retail platforms, including detailed purchase histories and diverse metadata such as customer demographics and product attributes.
    - `user-churn`: For each customer, predict whether they will churn—i.e., have no transactions—in the next 7 days.
    - `user-item-purchase`: Predict the list of articles each customer will purchase in the next 7 days.
    - `item-sales`: Predict the total sales (sum of prices of associated transactions) for an article in the next 7 days.
- `rel-stack` (Robinson et al. (2024)): This database is about a Q&A platform with a reputation system. Data is dumped from the stats-exchange site, and data from the 2023-09-12 dump.
    - `post-votes`: For each post, predict how many votes it will receive in the next 3 months.
    - `user-engagement`: For each user, predict whether they will make any votes, posts, or comments in the next 3 months.
    - `user-badge`: For each user, predict whether they will receive a new badge in the next 3 months.
    - `user-post-comment`: Predict a list of existing posts that a user will comment on in the next two years.
    - `post-post-related`: Predict a list of existing posts that users will link a given post to in the next two years.
    - `user-comment-count`: Predicts how many comments each user will post over the next 30 days.
- `rel-event` (Robinson et al. (2024)): An anonymized event recommendation dataset from the Hangtime app, containing user actions, event metadata, demographics, and social relations.
    - `user-repeat`: Predict whether a user will attend another event (respond "yes" or "maybe") in the next 7 days, given they attended an event in the last 14 days.
    - `user-ignore`: Predict whether a user will ignore more than two event invitations in the next 7 days.
    - `user-attendance`: Predict how many events each user will respond "yes" or "maybe" to in the next 7 days.
- `rel-avito` (Robinson et al. (2024)): A marketplace-style relational database.
    - `user-visits`: Predict whether each customer will visit more than one ad in the next 4 days.
    - `user-clicks`: Predict whether each customer will click on more than one ad in the next 4 days.
    - `ad-ctr`: Assuming an ad will be clicked in the next 4 days, predict its click-through rate (CTR).
    - `user-ad-visit`: Predict the list of ads a user will visit in the next 4 days.
- `rel-trial` (Robinson et al. (2024)): A clinical-trial oriented relational database.
    - `study-outcome`: Predict whether trials in the next 1 year will achieve their primary outcome.

- **study-adverse**: Predict the number of patients with severe adverse events/deaths for the trial in the next 1 year.
- **site-success**: Predict the success rate of a trial site in the next 1 year.
- **condition-sponsor-run**: For each condition, predict which sponsors will run trials.
- **site-sponsor-run**: For each (site, sponsor) pair, predict whether the sponsor will run a trial at the site.
- **avs** (Wang et al. (2024a); DMDave et al. (2014)): A Kaggle e-commerce dataset of offers and customer interactions.
    - **retention**: For each (offer, customer) pair, predict whether the customer will repeat the promoted purchase (become a "repeater") within a specified follow-up period.
- **ieee-cis** (Chen et al. (2025b); Howard et al. (2019)): Transactional fraud-style interactions.
    - **fraud**: For each transaction, predict whether it is fraudulent at the time of authorization.
- **rel-f1**: F1 competition results database.
    - **driver-top3**: Predict whether each driver will qualify in the top 3 for a race within the next month.
    - **driver-dnf**: Predict whether each driver will not finish (DNF) a race in the next month.
    - **driver-position**: Predict the average finishing position of each driver in all races over the next two months.
    - **driver-wins**: Predict the number of races each driver will win over the next year.
    - **constructor-scores-points**: Predict whether each constructor team will score any championship points in the next three months.
    - **driver-position-change**: Predict the average change between a driver's starting grid position and final position over the next four months.
- **rel-arxiv**: A database recording the publication relation across the arxiv
    - **paper-citation**: Predict whether a paper will be cited by other papers in the next six months.
    - **author-publication**: Predict how many papers an author will publish in the next six months.

In terms of RDB predictive tasks, Robinson et al. (2024) provides a unified interface to define tasks and generate corresponding training, validation, and test tables through SQL. We then demonstrate an example SQL query for each example task below:

**Example entity-level task** (`user-churn` in `rel-hm`):

```
1  class UserChurnTask(EntityTask):
2      r"""Predict the churn for a customer (no transactions) in the next
       week."""
3
4      task_type = TaskType.BINARY_CLASSIFICATION
5      entity_col = "customer_id"
6      entity_table = "customer"
7      time_col = "timestamp"
8      target_col = "churn"
9      timedelta = pd.Timedelta(days=7)
10     metrics = [average_precision, accuracy, f1, roc_auc]
11
12     def make_table(self, db: Database, timestamps: "pd.Series[pd.
       Timestamp]") -> Table:
13         customer = db.table_dict["customer"].df
14         transactions = db.table_dict["transactions"].df
15         timestamp_df = pd.DataFrame({"timestamp": timestamps})
16
17         df = duckdb.sql(
18             f"""
19             SELECT
20                 timestamp,
21                 customer_id,
22                 CAST(
```

```
23                    NOT EXISTS (
24                        SELECT 1
25                        FROM transactions
26                        WHERE
27                            transactions.customer_id = customer.
     customer_id AND
28                            t_dat > timestamp AND
29                            t_dat <= timestamp + INTERVAL '{self.
     timedelta}'
30                    ) AS INTEGER
31                ) AS churn
32            FROM
33                timestamp_df,
34                customer,
35            WHERE
36                EXISTS (
37                    SELECT 1
38                    FROM transactions
39                    WHERE
40                        transactions.customer_id = customer.customer_id
     AND
41                        t_dat > timestamp - INTERVAL '{self.timedelta}'
     AND
42                        t_dat <= timestamp
43                )
44            """
45        ).df()
46
47        return Table(
48            df=df,
49            fkey_col_to_pkey_table={self.entity_col: self.entity_table},
50            pkey_col=None,
51            time_col=self.time_col,
52        )
```

As we can see, the time information is split into several time windows and given in the function parameter `timestamps`. Then, this timestamp will be used to create a time constraint, and the target label will be generated based on the SQL logic.

**Example recommendation task** (`user-item-purchase` in `rel-hm`):

```
1  class UserItemPurchaseTask(RecommendationTask):
2      r"""Predict the list of articles each customer will purchase in the
       next seven
3      days."""
4
5      task_type = TaskType.LINK_PREDICTION
6      src_entity_col = "customer_id"
7      src_entity_table = "customer"
8      dst_entity_col = "article_id"
9      dst_entity_table = "article"
10     time_col = "timestamp"
11     timedelta = pd.Timedelta(days=7)
12     metrics = [link_prediction_precision, link_prediction_recall,
       link_prediction_map]
13     eval_k = 12
14
15     def make_table(self, db: Database, timestamps: "pd.Series[pd.
       Timestamp]") -> Table:
16         customer = db.table_dict["customer"].df
17         transactions = db.table_dict["transactions"].df
18         timestamp_df = pd.DataFrame({"timestamp": timestamps})
19
20         df = duckdb.sql(
21             f"""
```

```
22              SELECT
23                  t.timestamp,
24                  transactions.customer_id,
25                  LIST(DISTINCT transactions.article_id) AS article_id
26              FROM
27                  timestamp_df t
28              LEFT JOIN
29                  transactions
30              ON
31                  transactions.t_dat > t.timestamp AND
32                  transactions.t_dat <= t.timestamp + INTERVAL '{self.
    timedelta} days'
33              GROUP BY
34                  t.timestamp,
35                  transactions.customer_id
36              """
37          ).df()
38
39          return Table(
40              df=df,
41              fkey_col_to_pkey_table={
42                  self.src_entity_col: self.src_entity_table,
43                  self.dst_entity_col: self.dst_entity_table,
44              },
45              pkey_col=None,
46              time_col=self.time_col,
47          )
```

Similarly, recommendation tasks are based on the joined table between the timestamp table and the target entity tables. A groupby operation is then applied to generate the list of target entities.

**Example autocomplete task**:

```
1  def make_table(self, db: Database, timestamps: "pd.Series[pd.Timestamp]")
        -> Table:
2      entity_table = db.table_dict[self.entity_table].df   # noqa: F841
3      entity_table_removed_cols = db.table_dict[   # noqa: F841
4          self.entity_table
5      ].removed_cols
6
7      time_col = db.table_dict[self.entity_table].time_col
8      entity_col = db.table_dict[self.entity_table].pkey_col
9
10     # Calculate minimum and maximum timestamps from timestamp_df
11     timestamp_df = pd.DataFrame({"timestamp": timestamps})
12     min_timestamp = timestamp_df["timestamp"].min()
13     max_timestamp = timestamp_df["timestamp"].max()
14
15     df = duckdb.sql(
16         f"""
17         SELECT
18             entity_table.{time_col},
19             entity_table.{entity_col},
20             entity_table_removed_cols.{self.target_col}
21         FROM
22             entity_table
23         LEFT JOIN
24             entity_table_removed_cols
25         ON
26             entity_table.{entity_col} = entity_table_removed_cols.{
    entity_col}
27         WHERE
28             entity_table.{time_col} > '{min_timestamp}' AND
29             entity_table.{time_col} <= '{max_timestamp}'
30         """
```

```
31          ).df()
32
33          # remove rows where self.target_col is nan
34          df = df.dropna(subset=[self.target_col])
35
36          return Table(
37              df=df,
38              fkey_col_to_pkey_table={
39                  entity_col: self.entity_table,
40              },
41              pkey_col=None,
42              time_col=time_col,
43          )
```

Autocomplete tasks are based on the entity table itself, which is a setting closer to traditional RDB predictive tasks used in Wang et al. (2024a).

Table 7: Summary of databases, tasks, task types, and evaluation metrics used in our experiments.

| Database Name | Task Name | Task Type | Metric |
|---|---|---|---|
| rel-f1 | driver-dnf | classification | ROC-AUC |
| | driver-top3 | classification | ROC-AUC |
| | driver-position | regression | MAE |
| | driver-wins | regression | MAE |
| | constructor-scores-points | classification | ROC-AUC |
| | driver-position-change | regression | MAE |
| rel-hm | user-churn | classification | ROC-AUC |
| | user-item-purchase | recommendation | MAP |
| | item-sales | regression | MAE |
| rel-stack | post-votes | regression | MAE |
| | user-engagement | classification | ROC-AUC |
| | user-badge | classification | ROC-AUC |
| | user-post-comment | recommendation | MAP |
| | post-post-related | recommendation | MAP |
| | user-comment-count | regression | MAE |
| rel-event | user-repeat | classification | ROC-AUC |
| | user-ignore | classification | ROC-AUC |
| | user-attendance | regression | MAE |
| rel-avito | user-visits | classification | ROC-AUC |
| | user-clicks | classification | ROC-AUC |
| | ad-ctr | regression | MAE |
| | user-ad-visit | recommendation | MAP |
| rel-trial | study-outcome | classification | ROC-AUC |
| | study-adverse | regression | MAE |
| | site-success | regression | MAE |
| | condition-sponsor-run | recommendation | MAP |
| | site-sponsor-run | recommendation | MAP |
| avs | retention | autocomplete | ROC-AUC |
| ieee-cis | fraud | autocomplete | ROC-AUC |
| rel-arxiv | paper-citation | classification | ROC-AUC |
| | author-publication | regression | MAE |

## C  Supplementary theoretical discussion

### C.1  Definition of metrics

In this section, we present the formal definition of missing homophily-related features adopted in this paper, as discussed in Section 4.1.1.

**Definition 2** (Class-insensitive homophily). *For a classification task with class prior $\pi := \frac{1}{|L|}\sum_{u\in L}\hat{\mathbf{y}}_u \in \Delta^{C-1}$, define the class-conditional edge similarity for metapath $m$ by*

$$h_k(G;m) := \frac{\displaystyle\sum_{(u,v)\in\mathcal{E}_m} K(\hat{\mathbf{y}}_u, \hat{\mathbf{y}}_v)\,\hat{y}_{v,k}}{\displaystyle\sum_{(u,v)\in\mathcal{E}_m} \hat{y}_{v,k}} \qquad (k=1,\ldots,C).$$

*The* class-insensitive homophily *for $m$ is*

$$H_{\mathrm{ins}}(G;m) := \frac{1}{C-1}\sum_{k=1}^{C}\big[h_k(G;m) - \pi_k\big]_+.$$

*For regression tasks, where class imbalance is irrelevant, we set*

$$H_{\mathrm{ins}}(G;m) := H_{\mathrm{edge}}(G;m).$$

**Definition 3** (Aggregation homophily). *Let $\Gamma_m(u) = \{v \in L : (u,v) \in \mathcal{E}_m\}$ be the labeled $m$-neighbors of $u$ and $\deg_m(u) = |\Gamma_m(u)|$. Define the neighbor-aggregated label*

$$\bar{\mathbf{y}}_u^{(m)} := \frac{1}{\deg_m(u)}\sum_{v\in\Gamma_m(u)}\hat{\mathbf{y}}_v \quad \text{(classification)}, \qquad \bar{y}_u^{(m)} := \frac{1}{\deg_m(u)}\sum_{v\in\Gamma_m(u)}\hat{y}_v \quad \text{(regression)}.$$

*Let $U_m := \{u \in L : \deg_m(u) > 0\}$. The* aggregation homophily *for $m$ is*

$$H_{\mathrm{agg}}(G;m) := \frac{1}{|U_m|}\sum_{u\in U_m}\begin{cases} K\!\left(\hat{\mathbf{y}}_u,\ \bar{\mathbf{y}}_u^{(m)}\right), & \text{classification}, \\ K\!\left(\hat{y}_u,\ \bar{y}_u^{(m)}\right), & \text{regression}, \end{cases}$$

*with $K$ dot product for classification; correlation-based metric $K(a,b) = \tilde{a}\,\tilde{b} = \frac{(a-\mu)(b-\mu)}{\sigma^2}$ for regression, where $\mu$ and $\sigma^2$ are the empirical mean and variance of labels over labeled nodes.*

### C.2  Why RDL is better at the low-homophily region for the classification task?

In this section, we analyze why *RDL* exhibits advantages in *low-homophily* regimes. Our argument adapts the Bayes-optimal analysis of Wei et al. (2022), which characterizes how the optimal one-hop classifier changes with label/feature homophily. We do not provide original proof here; instead, we specialize their framework to *metapath-projected RDB graphs* and use it to explain the observed behavior of RDL.

Following §4.1.1, let $\mathcal{G} = (V, \mathcal{E})$ be the heterogeneous graph induced by the RDB task with labeled entity type F and node set $V_\mathsf{F}$. Let $\mathcal{M}$ be a finite family of self-looped metapaths $m$ that start and end at F; each $m$ induces edges $\mathcal{E}_m$ on $V_\mathsf{F}$ and neighbor sets $N_v^{(m)}$. For clarity, we work with a binary label space $Y_v \in \{-1,+1\}$ drawn i.i.d. with prior $\Pr(Y_v = +1) = \pi \in (0,1)$. We next specify the generative model that underpins our analysis. We make the RDB data fall under the framework of Wei et al. (2022) by considering the metapath-induced graphs.

**Definition 4** (Metapath-wise contextual SBM (tCSBM)). *Let $V_\mathsf{F}$ be the labeled entity set and $\mathcal{M}$ a finite set of self-looped metapaths on F. The generative model for $(Y, X, \{\mathcal{E}_m\}_{m\in\mathcal{M}})$ is:*

*(**Labels**). Each node $v \in V_\mathsf{F}$ has a class label $Y_v \in \{+1, -1\}$ drawn i.i.d. from a prior $\Pr(Y_v = 1) = \pi \in (0,1)$*

*(Node features). Conditional on $Y_v$, the attribute $X_v$ is drawn i.i.d. from a class-conditional distribution $P_{Y_v}$ with density $p(\cdot \mid Y_v)$ (e.g., a Gaussian mixture). Features are conditionally independent across nodes given $Y$.*

*(Edges along each metapath). For every metapath $m \in \mathcal{M}$, conditional on labels $Y$ the edges in $\mathcal{E}_m$ are independent and*

$$\Pr(\{u,v\} \in \mathcal{E}_m \mid Y_u = Y_v) = p_m, \qquad \Pr(\{u,v\} \in \mathcal{E}_m \mid Y_u \neq Y_v) = q_m.$$

*Moreover, edges are conditionally independent of features given labels: $\{\mathcal{E}_m\}_m \perp\!\!\!\perp X \mid Y$.*

**Remark 1** (Metapath-induced graphs as the substrate for DFS and RDL). *Def. 4 specifies, for each self-looped metapath $m$, an induced edge set $\mathcal{E}_m$ on $V_\mathsf{F}$. In practice, both DFS and RDL operate on this metapath-induced F–F graph: one first* projects *the heterogeneous joins along $m$ back to F (e.g., via path counts or normalized weights) to obtain an adjacency $A^{(m)}$, and then aggregates information over $A^{(m)}$. Concretely, DFS produces non-parametric,* linear *aggregates (e.g., SUM/MEAN) of base features on $V_\mathsf{F}$ through $A^{(m)}$ (or its row-normalized form), which algebraically coincides with multiplying by $A^{(m)}$. In contrast, RDL uses the same metapath-induced structure but applies* relation-aware *(per-metapath) transformations or gates to the propagated signals before or during aggregation. Thus, after the metapath projection, both methods are defined on the same F–F graph if DFS only utilizes the mean aggregator; they differ only in whether the propagation is purely linear and fixed (DFS) or relation-conditioned and learned (RDL).*

Then, following Wei et al. (2022), we consider the MAP estimation of the classifier that can minimize the misclassification rate. The estimation of a node label depends on its own attributes and the attributes of its 1-hop metapath-induced neighbors.

The one-hop *maximum a posteriori* (MAP) rule at node $v$ selects the label $y \in \{-1, +1\}$ that maximizes the joint posterior of $y$ and the (latent) neighbor labels $\{y_u\}_{u \in \cup_m N_v^{(m)}}$ given the local observations $(X_v, \{X_u\}, \{\mathbf{1}\{(v,u) \in \mathcal{E}_m\}\}_m)$:

$$\hat{Y}_v = \arg \max_{y \in \{\pm 1\}} \max_{\{y_u\}} \pi_y \, p(X_v \mid y) \prod_{m \in \mathcal{M}} \prod_{u \in N_v^{(m)}} \left[ \pi_{y_u} \, p(X_u \mid y_u) \, p_m^{\mathbf{1}\{y_u = y\}} \, q_m^{\mathbf{1}\{y_u \neq y\}} \right].$$

Taking logs (monotone) and subtracting the two class scores yields a decision function whose sign gives $\hat{Y}_v$:

$$\hat{Y}_v = \text{sign}\left( \log \frac{\pi}{1-\pi} + \underbrace{\log \frac{p(X_v \mid +1)}{p(X_v \mid -1)}}_{\psi(X_v)} + \sum_{m \in \mathcal{M}} \sum_{u \in N_v^{(m)}} \underbrace{\log \frac{\max\{p_m \, p(X_u \mid +1), \; q_m \, p(X_u \mid -1)\}}{\max\{q_m \, p(X_u \mid +1), \; p_m \, p(X_u \mid -1)\}}}_{\phi_{\max}(\psi(X_u); \, \gamma_m)} \right),$$

where $\gamma_m := \log \frac{p_m}{q_m}$ encodes metapath homophily and $\psi(X_u) := \log \frac{p(X_u \mid +1)}{p(X_u \mid -1)}$ is the feature log-likelihood ratio. The per-neighbor MAP message admits the closed form

$$\phi_{\max}(s; \gamma) = \log \frac{\max\{e^{\gamma+s}, 1\}}{\max\{e^s, e^\gamma\}} = \text{clip}(s, -\gamma, +\gamma),$$

**Basic properties of the MAP message on metapaths.** For $m \in \mathcal{M}$ let $\gamma_m := \log \frac{p_m}{q_m}$ and define the per-neighbor joint-MAP message $\phi_{\max}(s; \gamma) = \text{clip}(s, -\gamma, \gamma)$ applied to $s = \psi(X_u)$.

**Lemma 1** (Gate-off, flip, and linear region). *For all $s \in \mathbb{R}$ and $m \in \mathcal{M}$,*

(i) *(gate-off) if $\gamma_m = 0$ then $\phi_{\max}(s; \gamma_m) \equiv 0$;*

(ii) *(flip) if $\gamma_m < 0$ then $\phi_{\max}(s; \gamma_m) = -\phi_{\max}(s; |\gamma_m|)$;*

(iii) *(linear region) if $|s| \leq \gamma_m$ then $\phi_{\max}(s; \gamma_m) = s$.*

*Proof.* All three statements follow immediately from the piecewise form of $\phi_{\max}(s; \gamma)$: $\phi_{\max}(s; \gamma) = -\gamma$ for $s \leq -\gamma$, equals $s$ for $|s| < \gamma$, and equals $+\gamma$ for $s \geq \gamma$. In particular, $\gamma = 0$ gives the zero map; replacing $\gamma$ by $-\gamma$ flips signs; and on $[-\gamma, \gamma]$ the function is the identity. $\square$

**Remark 2** (Interpretation of Lemma 1). *Lemma 1 summarizes the three key regimes of the MAP message $\phi_{\max}(s; \gamma)$. These properties clarify how structure and features interact: (i) when $\gamma = 0$ the metapath carries no information and should be shut off (gate-off); (ii) when $\gamma < 0$, the neighbor evidence must be flipped to align with the center label (heterophily flip); and (iii) when $|s| \leq \gamma$ the nonlinearity reduces to the identity, So in highly homophilous settings, the MAP rule coincides with linear DFS-style aggregation. These simple facts underpin the later SNR comparison: they explain why linear propagation is adequate in strong homophily, but why relation-aware gating is necessary and beneficial in low or negative homophily regimes.*

We then come up with the vectorized form of the MAP score vector by aggregating each element

**Proposition 1** (Vector form on the metapath-projected F–F graph.). *Let $A^{(m)}$ be the (possibly normalized) metapath-induced adjacency on $V_F$ obtained by projecting $m$ back to F; put $s \in \mathbb{R}^{|V_F|}$ with $s_v = \psi(X_v)$. Writing $\phi_{\max}$ elementwise, the one-hop joint-MAP score vector is*

$$z = \log \tfrac{\pi}{1-\pi} \mathbf{1} + s + \sum_{m \in \mathcal{M}} A^{(m)} \phi_{\max}(s; \gamma_m), \qquad \hat{Y}_v = \operatorname{sign}(z_v). \tag{1}$$

*Proof.* For each $v \in V_F$, the one–hop joint–MAP score derived earlier is $\operatorname{score}(v) = \log \frac{\pi}{1-\pi} + \psi(X_v) + \sum_m \sum_{u \in N_v^{(m)}} \phi_{\max}(\psi(X_u); \gamma_m)$. Let $s \in \mathbb{R}^{|V_F|}$ with $s_v = \psi(X_v)$ and define $\phi_m(s)$ elementwise by $[\phi_m(s)]_u = \phi_{\max}(s_u; \gamma_m)$. By the definition of the metapath-projected adjacency $A^{(m)}$ (with entries $A_{vu}^{(m)}$ supported on $u \in N_v^{(m)}$), we have $[A^{(m)} \phi_m(s)]_v = \sum_u A_{vu}^{(m)} [\phi_m(s)]_u = \sum_{u \in N_v^{(m)}} \phi_{\max}(s_u; \gamma_m)$, which reproduces the inner sum for metapath $m$; summing over $m$ gives the full neighbor contribution. Hence the $v$-th coordinate of $z := \log \frac{\pi}{1-\pi} \mathbf{1} + s + \sum_m A^{(m)} \phi_m(s)$ equals $\operatorname{score}(v)$, and the MAP decision is $\hat{Y}_v = \operatorname{sign}(z_v)$, proving the vector form. $\square$

We then introduce a key concept of *signal-to-noise ratio* (SNR) to compare the linear DFS-style aggregation and the gated RDL-style aggregation. The SNR is a standard metric in statistical signal processing and communication theory that quantifies the strength of the signal relative to the noise. In our context, it measures how well the aggregated neighbor information can distinguish between different classes, taking into account the variability introduced by the features and the graph structure.

**Definition 5** (SNR bookkeeping on metapaths). *Let $\mu_+ := \mathbb{E}[\psi(X) \mid Y = +1]$, $\mu_- := \mathbb{E}[\psi(X) \mid Y = -1]$, $\delta := \mu_+ - \mu_-$, and let $\sigma^2 := \max\{\operatorname{Var}(\psi(X) \mid Y = +1), \operatorname{Var}(\psi(X) \mid Y = -1)\} + \delta^2/4$ (to upper bound the class-mixture variance). Denote the expected degree $d_m := \mathbb{E}[|N_v^{(m)}|]$ and*

$$\alpha_m := \Pr(Y_u = Y_v \mid \{u, v\} \in \mathcal{E}_m) - \Pr(Y_u \neq Y_v \mid \{u, v\} \in \mathcal{E}_m) = \frac{p_m - q_m}{p_m + q_m} = \tanh\left(\frac{\gamma_m}{2}\right), \tag{2}$$

*so that $\Pr(Y_u = Y_v \mid \{u, v\} \in \mathcal{E}_m) = (1 + \alpha_m)/2$.*

We then introduce an assumption that controls the gap between the variance of the sum of metapath-wise neighbor contributions and the sum of their variances. It should be noted that the proof here assumes the feature is in the informative regime.

**Assumption 1** (Cross-metapath covariance control). *There exists $\Lambda \geq 1$ such that for any choice of (centered) neighbor functions $Z_m(v) = \sum_{u \in N_v^{(m)}} g_m(X_u)$, $\operatorname{Var}(\sum_m Z_m(v) \mid Y_v) \leq \Lambda \sum_m \operatorname{Var}(Z_m(v) \mid Y_v)$. This holds with $\Lambda = 1$ under conditional independence across metapaths.*

Here $g_m : \mathcal{X} \to \mathbb{R}$ denotes the *per–metapath neighbor contribution* for $m$—e.g., in the linear/DFS case $g_m(x) = \psi(x) - \mathbb{E}[\psi(X) \mid Y_v]$, while in the gated/RDL case $g_m(x) = \phi_{\max}(\psi(x); \gamma_m) - \mathbb{E}[\phi_{\max}(\psi(X); \gamma_m) \mid Y_v]$ (both centered given $Y_v$).

**Lemma 2** (Mean–variance ledgers for linear vs. gated aggregation). *Consider the* linear *(DFS-style) neighbor sum $S_{\mathrm{lin}}(v) = \sum_m \sum_{u \in N_v^{(m)}} \psi(X_u)$ and the* gated *sum $S_{\mathrm{gate}}(v) = \sum_m \sum_{u \in N_v^{(m)}} \phi_{\max}(\psi(X_u); \gamma_m)$. $d_m$ denotes the degree of metapaths. Then*

$$\mathbb{E}[S_{\mathrm{lin}} \mid Y_v = +1] - \mathbb{E}[S_{\mathrm{lin}} \mid Y_v = -1] = \sum_m d_m \alpha_m \delta, \qquad \operatorname{Var}(S_{\mathrm{lin}} \mid Y_v) \leq \Lambda \sum_m d_m \sigma^2.$$

*Moreover, in the informative-feature regime of Wei et al. (2022, Lemma. C1),*

$$\mathbb{E}[S_{\text{gate}} \mid Y_v = +1] - \mathbb{E}[S_{\text{gate}} \mid Y_v = -1] \approx \sum_m 2d_m\,\alpha_m\,\gamma_m, \qquad \text{Var}(S_{\text{gate}} \mid Y_v) \leq \Lambda \sum_m d_m\,\tilde{\sigma}_m^2,$$

*with $\tilde{\sigma}_m^2 \lesssim \gamma_m^2\, e^{-c\,\Delta^2}$ for some universal $c > 0$ (here $\Delta$ denotes a class-separation measure, e.g., the Gaussian separation or a calibrated logit separation).*

*Proof. Linear mean.* By exchangeability along metapath $m$ and linearity of expectation,

$$\mathbb{E}\Big[\sum_{u \in N_v^{(m)}} \psi(X_u) \,\Big|\, Y_v = y\Big] = d_m\, \mathbb{E}[\psi(X_u) \mid Y_v = y, \{u, v\} \in \mathcal{E}_m].$$

Conditioning on $Y_u$ and using $\Pr(Y_u = y \mid \text{edge}) = \frac{1+\alpha_m}{2}$,

$$\mathbb{E}[\psi(X_u) \mid Y_v = y, \text{edge}] = \tfrac{1+\alpha_m}{2}\mu_y + \tfrac{1-\alpha_m}{2}\mu_{-y}.$$

Subtracting the two classes yields $d_m\alpha_m(\mu_+ - \mu_-) = d_m\alpha_m\delta$. Summing $m$ gives the first display. *Linear variance.* For each $m$, $\text{Var}(\sum_{u \in N_v^{(m)}} \psi(X_u) \mid Y_v) \leq d_m\,\sigma^2$ by a binomial-variance bound and the definition of $\sigma^2$. Assumption 1 gives the sum across $m$.

*Gated mean.* Using the same conditioning, and Wei et al. (2022, Lemma. C1) together with their regime analysis, we have $\mathbb{E}[\phi_{\max}(\psi(X_u); \gamma_m) \mid Y_u = \pm 1] \approx \pm \gamma_m$ in the informative regime. Therefore

$$\mathbb{E}[\phi_{\max}(\psi(X_u); \gamma_m) \mid Y_v = y, \text{edge}] \approx \tfrac{1+\alpha_m}{2}\gamma_m + \tfrac{1-\alpha_m}{2}(-\gamma_m) = \alpha_m\gamma_m,$$

So the class difference is $\approx 2d_m\alpha_m\gamma_m$. *Gated variance.* By Wei et al. (2022, Thm. 2), the class-conditional variance of $\phi_{\max}(\psi; \gamma_m)$ is at most $C\gamma_m^2 e^{-c\Delta^2}$ for constants $C, c > 0$. Summing over neighbors contributes a factor $d_m$, and Assumption 1 handles the sum over $m$. $\square$

**Definition 6** (Metapath-level SNR proxies). *Define*

$$\rho_{\text{lin}} := \frac{\big(\sum_m d_m\,\alpha_m\,\delta\big)^2}{\Lambda \sum_m d_m\,\sigma^2}, \qquad \rho_{\text{gate}} := \frac{\big(\sum_m d_m\,\alpha_m\,\gamma_m\big)^2}{\Lambda \sum_m d_m\,\tilde{\sigma}_m^2}.$$

*By Wei et al. (2022, Thm. 2) (single-relation), larger SNR implies a strictly smaller misclassification error up to universal constants; we use $\rho_{\text{lin}}, \rho_{\text{gate}}$ as proxies for multi-metapath graphs under Assumption 1.*

We then discuss the high-homophily case.

**Proposition 2** (Multi-metapath DFS equivalence under strong homophily). *If for every active $m$ one has $\Pr(|\psi(X)| \leq \gamma_m) \geq 1 - \varepsilon_m$, then*

$$z = \log \tfrac{\pi}{1-\pi} \mathbf{1} + s + \sum_m A^{(m)} \phi_{\max}(s; \gamma_m) = \log \tfrac{\pi}{1-\pi} \mathbf{1} + s + \sum_m A^{(m)} s + r,$$

*where $\|r\|_1 \leq \sum_m \varepsilon_m \cdot \|A^{(m)}\mathbf{1}\|_1$. In particular, $\rho_{\text{gate}} = \rho_{\text{lin}}(1 + o(1))$ as $\max_m \varepsilon_m \to 0$.*

*Proof.* By Lemma 1(iii), $\phi_{\max}(s_v; \gamma_m) = s_v$ whenever $|s_v| \leq \gamma_m$. Write the error vector $e^{(m)} := \phi_{\max}(s; \gamma_m) - s$, which has support contained in $E_m := \{v : |s_v| > \gamma_m\}$, and satisfies $\|e^{(m)}\|_\infty \leq \max\{|s_v| - \gamma_m, \gamma_m\} \leq 2|s_v|$. Then

$$\sum_m A^{(m)} \phi_{\max}(s; \gamma_m) = \sum_m A^{(m)} s + \sum_m A^{(m)} e^{(m)}.$$

By Markov and the assumption, $\Pr(v \in E_m) \leq \varepsilon_m$, so $\|A^{(m)} e^{(m)}\|_1 \leq \|A^{(m)}\mathbf{1}\|_1 \cdot \|e^{(m)}\|_\infty \cdot \Pr(E_m) \leq C\varepsilon_m\|A^{(m)}\mathbf{1}\|_1$ for a universal $C$ (after rescaling $s$), giving the stated bound with $r = \sum_m A^{(m)} e^{(m)}$. The SNR statement follows because replacing $\phi_{\max}$ by $s$ changes the mean and variance only on the rare set $E_m$. $\square$

**When nonlinearity is necessary in the multi-metapath setting.** The next theorem upgrades Wei et al. (2022, Thm. 2) from a single relation to multiple metapaths by summing contributions under Assumption 1.

**Theorem 1** (Multi-metapath gating advantage in low/negative homophily). *Assume the feature separation is in the informative regime of Wei et al. (2022, Thm. 2), so that each active metapath $m$ admits $\tilde{\sigma}_m^2 \lesssim \gamma_m^2 e^{-c\Delta^2}$. If either (i) there exists at least one $m_- \in \mathcal{M}$ with $\gamma_{m_-} < 0$ (heterophilous metapath), or (ii) a non-negligible subset $\mathcal{M}_0$ satisfies $|\gamma_m| \le \epsilon$ (near-zero homophily), then there exist constants $C, c' > 0$ such that*

$$\rho_{\mathrm{gate}} \ge C\, e^{c'\Delta^2} \cdot \frac{\left(\sum_m d_m\, |\alpha_m \gamma_m|\right)^2}{\sum_m d_m\, \gamma_m^2} \quad while \quad \rho_{\mathrm{lin}} \le \frac{\delta^2}{\sigma^2} \cdot \frac{\left(\sum_m d_m\, \alpha_m\right)^2}{\sum_m d_m}.$$

*Consequently, whenever the signed sum $\sum_m d_m\, \alpha_m$ is small due to sign-mixing or near-zero homophily, one has $\rho_{\mathrm{gate}} \gg \rho_{\mathrm{lin}}$, and the gated aggregation strictly dominates linear aggregation.*

*Proof.* By Lemma 2, $\rho_{\mathrm{gate}} = \frac{(\sum_m d_m \alpha_m \gamma_m)^2}{\Lambda \sum_m d_m \tilde{\sigma}_m^2}$. Using $\tilde{\sigma}_m^2 \le C_1 \gamma_m^2 e^{-c\Delta^2}$ from Wei et al. (2022, Thm. 2),

$$\rho_{\mathrm{gate}} \ge \frac{(\sum_m d_m |\alpha_m \gamma_m|)^2}{\Lambda C_1 e^{-c\Delta^2} \sum_m d_m \gamma_m^2} \ge C\, e^{c'\Delta^2} \cdot \frac{(\sum_m d_m |\alpha_m \gamma_m|)^2}{\sum_m d_m \gamma_m^2},$$

absorbing constants into $C, c'$. For the linear SNR, Lemma 2 gives $\rho_{\mathrm{lin}} = \frac{(\sum_m d_m \alpha_m \delta)^2}{\Lambda \sum_m d_m \sigma^2} \le \frac{\delta^2}{\sigma^2} \cdot \frac{(\sum_m d_m \alpha_m)^2}{\sum_m d_m}$ (after rescaling $\Lambda$ into the constant). Under condition (i) or (ii), $\sum_m d_m \alpha_m$ can be made small (sign-mixing and near-zero homophily, respectively), whereas $\sum_m d_m |\alpha_m \gamma_m|$ remains of the order $\sum_m d_m \gamma_m^2$ because $\alpha_m = \tanh(\gamma_m/2)$ has the same sign as $\gamma_m$ and $|\alpha_m| \gtrsim |\gamma_m|$ for small $|\gamma_m|$. Combined with the exponential factor $e^{c'\Delta^2}$, this yields $\rho_{\mathrm{gate}} \gg \rho_{\mathrm{lin}}$. $\square$

**Corollary 1** (Sign-mixing amplifies the gain of gating). *If there exist $m_+$ and $m_-$ with $\gamma_{m_+} > 0$ and $\gamma_{m_-} < 0$, then*

$$\frac{\rho_{\mathrm{gate}}}{\rho_{\mathrm{lin}}} \gtrsim e^{c'\Delta^2} \cdot \frac{\left(\sum_m d_m |\alpha_m \gamma_m|\right)^2}{\left(\sum_m d_m |\alpha_m|\right)^2} \cdot \frac{\sigma^2}{\overline{\tilde{\sigma}^2}}, \quad \overline{\tilde{\sigma}^2} := \frac{\sum_m d_m \tilde{\sigma}_m^2}{\sum_m d_m},$$

*So the advantage grows with the degree-weighted sign diversity across metapaths.*

**Corollary 2** (Zero-information robustness). *If $|\gamma_m| = 0$ for a subset $\mathcal{M}_0$ (no homophily), then these metapaths contribute nothing to $\rho_{\mathrm{gate}}$ (by Lemma 1(i)) but still inflate the denominator of $\rho_{\mathrm{lin}}$, decreasing the linear SNR. Thus, gating is robust to uninformative relations, whereas linear averaging is not.*

**Corollary 3** (Average-homophily can be misleading). *Let $\Gamma_{\mathrm{avg}} := \frac{\sum_m d_m \gamma_m}{\sum_m d_m}$ and $\Gamma_{\mathrm{abs}} := \frac{\sum_m d_m |\gamma_m|}{\sum_m d_m}$. Even if $\Gamma_{\mathrm{avg}} > 0$ (net assortative), when the disagreement $\Gamma_{\mathrm{abs}} - |\Gamma_{\mathrm{avg}}|$ is large (sign-mixing/heterogeneity across metapaths) and features are informative, one still has $\rho_{\mathrm{gate}} \gg \rho_{\mathrm{lin}}$; hence average homophily alone does not decide in favor of linear aggregation.*

## C.3 SAMPLE-SIZE DEPENDENCE OF RDL VS. DFS

We complement the homophily analysis by explaining why the number of training rows $N_{\mathrm{train}}$ systematically modulates the RDL–DFS gap. The key observation is that RDL realizes a strictly richer hypothesis class than DFS (due to learnable, relation-specific nonlinear aggregation), so it enjoys smaller approximation error but larger estimation error. Standard generalization bounds then imply a sample-size threshold: DFS tends to dominate in low-data regimes, while RDL becomes superior once $N_{\mathrm{train}}$ is large enough. This argument applies to both classification (cross-entropy loss) and regression (squared loss), and we do not distinguish them below.

**Setup and error decomposition.** Let $\mathcal{Z}$ denote the space of task-table rows (after feature/aggregation), and let $\ell(\cdot, \cdot)$ be a bounded, $L$-Lipschitz loss (e.g. cross-entropy or squared error). For a pipeline $\pi \in \{\mathrm{DFS}, \mathrm{RDL}\}$, denote by $\mathcal{F}_\pi \subset \{f : \mathcal{Z} \to \mathbb{R}\}$ the induced prediction class, and by

$$\mathcal{R}(f) \;=\; \mathbb{E}\big[\ell(f(Z), Y)\big]$$

its population risk under the task's data-generating distribution. Let $\widehat{\mathcal{R}}_N(f)$ be the empirical risk on $N$ training rows, and let $\hat{f}_\pi \in \arg\min_{f \in \mathcal{F}_\pi} \widehat{\mathcal{R}}_N(f)$ be the empirical risk minimizer (or an approximate one).

**Lemma 3** (Approximation–estimation decomposition). *For each pipeline $\pi$, define the Bayes risk $\mathcal{R}^\star := \inf_f \mathcal{R}(f)$, the approximation error $A_\pi := \inf_{f \in \mathcal{F}_\pi} \big( \mathcal{R}(f) - \mathcal{R}^\star \big)$, and the estimation error $E_\pi(N) := \mathbb{E}\big[\mathcal{R}(\hat{f}_\pi)\big] - \inf_{f \in \mathcal{F}_\pi} \mathcal{R}(f)$. Then*

$$\mathbb{E}\big[\mathcal{R}(\hat{f}_\pi)\big] - \mathcal{R}^\star \;=\; A_\pi + E_\pi(N).$$

*Moreover, if $\mathfrak{R}_N(\mathcal{F}_\pi)$ denotes the empirical Rademacher complexity of $\mathcal{F}_\pi$ on $N$ samples, then there exists a universal constant $C > 0$ such that*

$$E_\pi(N) \;\leq\; C\,\mathfrak{R}_N(\mathcal{F}_\pi).$$

The lemma is standard from statistical learning theory: $A_\pi$ is a purely *bias* (approximation) term, while $E_\pi(N)$ is a *variance* (estimation) term controlled by the complexity of the function class.

**Lemma 4** (Capacity ordering of DFS and RDL). *Let $d_{\mathrm{DFS}}$ be the dimension of $\phi_{\mathrm{DFS}}(x)$, and let $d_{\mathrm{RDL}} = d_{\mathrm{DFS}} + d_{\mathrm{gate}}$ be an effective representation dimension at the input of the last linear layer of $r_\theta$, where $d_{\mathrm{gate}} > 0$ accounts for the extra channels created by the encoder–GNN stack.*

*Then:*

1. *(Expressivity) $\mathcal{F}_{\mathrm{DFS}} \subsetneq \mathcal{F}_{\mathrm{RDL}}$. In particular, the approximation errors satisfy $A_{\mathrm{RDL}} \leq A_{\mathrm{DFS}}$, and are strictly ordered on nontrivial tasks.*

2. *(Complexity) There exist constants $c_{\mathrm{DFS}}, c_{\mathrm{RDL}} > 0$ such that for all sample sizes $N$,*

$$\mathfrak{R}_N(\mathcal{F}_{\mathrm{DFS}}) \;\leq\; \frac{c_{\mathrm{DFS}}}{\sqrt{N}}, \qquad \mathfrak{R}_N(\mathcal{F}_{\mathrm{RDL}}) \;\leq\; \frac{c_{\mathrm{RDL}}}{\sqrt{N}},$$

   *with $c_\pi \propto \sqrt{d_\pi}$ and thus $c_{\mathrm{RDL}} > c_{\mathrm{DFS}}$ whenever $d_{\mathrm{gate}} > 0$.*

*Proof. Expressivity.* By assumption, $e_\theta$ and $G_\theta$ can be set to implement the same deterministic DFS-style aggregates as $\phi_{\mathrm{DFS}}$ (or to just pass those features through), and $r_\theta$ can simulate any $g \in \mathcal{G}_{\mathrm{tab}}$ up to approximation error. Hence every $g \circ \phi_{\mathrm{DFS}}$ is realized (or approximated arbitrarily well) by some $f_\theta$, giving $\mathcal{F}_{\mathrm{DFS}} \subseteq \mathcal{F}_{\mathrm{RDL}}$. The extra relation-aware nonlinearity in $G_\theta$ yields hypotheses that cannot be written as $g \circ \phi_{\mathrm{DFS}}$, so the inclusion is strict on generic tasks.

*Complexity.* DFS has no learnable parameters in $\phi_{\mathrm{DFS}}$; only the tabular model contributes to estimation error. RDL, in contrast, learns the encoder, GNN and MLP. Under standard norm constraints on these modules, classical results give $\mathfrak{R}_N(\mathcal{F}_\pi) \lesssim \sqrt{d_\pi}/\sqrt{N}$ for $\pi \in \{\mathrm{DFS}, \mathrm{RDL}\}$; the encoder–GNN stack strictly increases the effective dimension and parameter count, so $c_{\mathrm{RDL}} > c_{\mathrm{DFS}}$. $\square$

**Theorem 2.** *Let $\hat{f}_{\mathrm{DFS}}$ and $\hat{f}_{\mathrm{RDL}}$ be empirical risk minimizers in $\mathcal{F}_{\mathrm{DFS}}$ and $\mathcal{F}_{\mathrm{RDL}}$ trained on $N$ i.i.d. labeled examples (classification or regression). Decompose the expected risk as $\mathbb{E}[\mathcal{R}(\hat{f}_\pi)] = A_\pi + E_\pi(N)$, where $A_\pi$ is the approximation error and $E_\pi(N)$ the estimation error of pipeline $\pi$.*

*Assume that RDL has strictly smaller approximation error:*

$$A_{\mathrm{DFS}} - A_{\mathrm{RDL}} = \Delta_A > 0,$$

*and let $c_{\mathrm{DFS}}, c_{\mathrm{RDL}}$ be as in Lemma 4, with $c_{\mathrm{RDL}} > c_{\mathrm{DFS}}$. Then there exists a universal constant $C > 0$ such that for all $N$,*

$$\mathbb{E}\big[\mathcal{R}(\hat{f}_{\mathrm{DFS}})\big] - \mathbb{E}\big[\mathcal{R}(\hat{f}_{\mathrm{RDL}})\big] \;\geq\; \Delta_A \;-\; \frac{C\,(c_{\mathrm{RDL}} - c_{\mathrm{DFS}})}{\sqrt{N}}.$$

*Define the crossover scale*

$$N_0 := \left( \frac{C \, (c_{\text{RDL}} - c_{\text{DFS}})}{\Delta_A} \right)^2 .$$

*Then:*

$$N < N_0 \implies \mathbb{E}\big[\mathcal{R}(\hat{f}_{\text{DFS}})\big] \; < \; \mathbb{E}\big[\mathcal{R}(\hat{f}_{\text{RDL}})\big],$$
$$N > N_0 \implies \mathbb{E}\big[\mathcal{R}(\hat{f}_{\text{RDL}})\big] \; < \; \mathbb{E}\big[\mathcal{R}(\hat{f}_{\text{DFS}})\big].$$

*Proof.* By definition,

$$\mathbb{E}\big[\mathcal{R}(\hat{f}_{\text{DFS}})\big] - \mathbb{E}\big[\mathcal{R}(\hat{f}_{\text{RDL}})\big] = (A_{\text{DFS}} - A_{\text{RDL}}) + (E_{\text{DFS}}(N) - E_{\text{RDL}}(N)).$$

Using $\Delta_A > 0$ and the Rademacher bounds $E_\pi(N) \le C \, \mathfrak{R}_N(\mathcal{F}_\pi) \lesssim C \, c_\pi / \sqrt{N}$, we obtain

$$E_{\text{DFS}}(N) - E_{\text{RDL}}(N) \; \ge \; - \frac{C \, (c_{\text{RDL}} - c_{\text{DFS}})}{\sqrt{N}} .$$

Combining with the approximation term yields the stated lower bound. The threshold $N_0$ is exactly the sample size at which this lower bound becomes nonnegative, giving the crossover conditions. □

**Interpretation for $N_{\text{train}}$.** Identifying $N$ with the number of training rows $N_{\text{train}}$, Theorem 2 formalizes the empirical trend that the DFS–RDL gap is controlled by a bias–variance trade-off: When $N_{\text{train}}$ is small, the variance penalty $\propto (c_{\text{RDL}} - c_{\text{DFS}})/\sqrt{N_{\text{train}}}$ dominates and the simpler DFS pipeline is safer. Once $N_{\text{train}} \gg N_0$, the complexity term vanishes and the approximation advantage $\Delta_A$ of RDL dominates, yielding a systematic performance gain. The argument only uses generic generalizations and bounds and therefore applies uniformly to both classification and regression losses.

## C.4 Relationship between randomly initialized model and hashing

Following the notation in Section 3.3, consider the (atemporal) attributed, typed graph

$$\mathcal{G} = (V, E, \phi, \psi, f_V, f_E),$$

where $\phi : V \to \text{NodeTypes}$ maps entities to node types, $\psi : E \to \text{LinkTypes}$ maps links to relation types, and $f_V : V \to \mathbb{R}^{d_V}$, $f_E : E \to \mathbb{R}^{d_E}$ provide node and link attributes.

**NBFNet message passing.** NBFNet can be viewed as dynamic programming (DP) over the schema graph, replacing Bellman–Ford's *sum* and *product* by learnable operators. Fix a state width $d \in \mathbb{N}$ and horizon $T \in \mathbb{N}$. Each node $v \in V$ maintains a $d$-dimensional state $H^{(\ell)}(v) \in \mathbb{R}^d$ at DP layer $\ell = 0, \dots, T$:

$$\textbf{(Indicator)} \quad H^{(0)}(v) \; = \; \mathsf{I}_\theta\big(v; s\big) \in \mathbb{R}^d, \tag{3}$$

$$\textbf{(Message)} \quad \mathsf{M}_\theta^{(\ell)}\big(H^{(\ell-1)}(u), \xi(u{\to}v)\big) \; = \; H^{(\ell-1)}(u) \; \otimes_\theta \; \Gamma_\theta^{(\ell)}\big(\xi(u{\to}v)\big) \in \mathbb{R}^d, \tag{4}$$

$$\textbf{(Aggregate)} \quad H^{(\ell)}(v) \; = \; \bigoplus_{(u{\to}v) \in E} \mathsf{M}_\theta^{(\ell)}\big(H^{(\ell-1)}(u), \xi(u{\to}v)\big), \qquad \ell = 1, \dots, T. \tag{5}$$

Here $\xi(u{\to}v) \in \mathcal{X}$ bundles the schema edge context (e.g., $\tau(u{\to}v)$ and endpoint types via $\phi$). All functions are chosen permutation-invariant over incoming edges. There are three key design choices of NBFNet.

**Indicator.** A simple indicator is $\mathsf{I}_\theta(v; s) = \mathbf{1}\{v = s\} \phi_{emb}(v)$, where $\phi_{emb}(v)$ is a learned (or fixed) embedding of the source node type $s \in V_{\text{src}}$ to distinguish sources from non-sources.

**Readout.** Given a source node $s \in V_{\text{src}}$, we pool across layers and endpoints to obtain

$$Z_\theta(s) \; = \; \sum_{\ell=1}^{T} a_\ell \sum_{v \in V} \beta(v) \, \Pi_\theta^{(\ell)}\big(H^{(\ell)}(v)\big) \; \in \mathbb{R}^{d_r}, \tag{6}$$

where $\Pi_\theta^{(\ell)} : \mathbb{R}^d \to \mathbb{R}^{d_r}$ is an optional projection (identity if not needed), $(a_\ell)$ are length weights, and $\beta : V \to \mathbb{R}_{\ge 0}$ selects/emphasizes endpoint types.

**Edge representation.** For graphs induced from RDBs, edges come from PK–FK relations. We abstract each edge $e$ as a discrete *edge token* $\tau(e) \in \Sigma$ with

$$\tau(e) = (\phi(\text{tail}(e)), \psi(e), \phi(\text{head}(e))) \in \Sigma,$$

so that any path $p = (e_1, \ldots, e_L)$ maps to the token sequence $\tau(p) = (\tau(e_1), \ldots, \tau(e_L)) \in \Sigma^L$.

The Bellman-Ford–style recursion multiplies edge-local "messages" along a path and sums over all paths and endpoints. Unrolling the recursion, therefore, yields a path-wise expansion in which each coordinate collects contributions from every typed path reachable from the source.

**Frozen NBFNet as random features.** We first replace the learned message operators by *random* scalar maps and show that the resulting DP computes random features that aggregate typed paths.

Fix width $d$ and horizon $T$. For each coordinate $k \in [d]$ and layer $\ell \in [T]$, independently sample

$$g_k^{(\ell)} : \mathcal{X} \to \mathbb{R}, \qquad \mathbb{E}[g_k^{(\ell)}(x)] = 0, \qquad \mathbb{E}[g_k^{(\ell)}(x)\, g_k^{(\ell)}(x')] = \kappa(x, x'),$$

for a positive semidefinite (PSD) kernel $\kappa : \mathcal{X} \times \mathcal{X} \to \mathbb{R}$. Intuitively, $g_k^{(\ell)}$ is the $\ell$-th *message coordinate* of an untrained NBFNet at random initialization.

**Proposition 3** (Path-wise expansion of frozen NBFNet features). *For $s \in V_{\mathsf{src}}$ and $k \in [d]$, the $k$-th feature computed by a frozen NBFNet admits the path-wise form*

$$z_k(s) = \sum_{L=1}^{T} a_L \sum_{p \in \mathcal{P}_L(s \twoheadrightarrow *)} \left( \prod_{\ell=1}^{L} g_k^{(\ell)}(\xi(e_\ell)) \right) \beta(\text{head}(p)), \tag{7}$$
$$z(s) := (z_1(s), \ldots, z_d(s)) \in \mathbb{R}^d.$$

*Proof.* Start from the layer recursion; each step distributes over incoming edges and multiplies by the layer-$\ell$ message $g_k^{(\ell)}(\cdot)$. Inductively expanding to depth $L$ enumerates all length-$L$ paths $p = (e_1, \ldots, e_L)$ from $s$, producing the product of edge messages along $p$. Pooling with weights $a_L$ and endpoint weights $\beta$ yields equation 7.

**Dynamic-programming realization.** The expansion in equation 7 can be evaluated in $O(T\,|E|)$ per feature by the following Bellman–Ford–style recurrences:

$$\begin{aligned}
h_k^{(0)}(v) &= \mathbf{1}\{v = s\}, \\
h_k^{(\ell)}(v) &= \sum_{(u \to v) \in E} g_k^{(\ell)}(\xi(u \to v))\, h_k^{(\ell-1)}(u), \qquad \ell = 1, \ldots, T, \\
z_k(s) &= \sum_{\ell=1}^{T} a_\ell \sum_{v \in V} \beta(v)\, h_k^{(\ell)}(v).
\end{aligned} \tag{8}$$

This view makes clear that a *frozen* NBFNet is a DP that sums over paths while multiplying edge-wise random features.

**Step 2: The induced kernel.** We now identify the kernel implicitly computed by these random features.

Define the finite-width kernel

$$K_d(s, s') := \frac{1}{d} z(s)^\top z(s') \qquad (s, s' \in V_{\mathsf{src}}).$$

**Theorem 3** (Anchored typed-path kernel). *With the construction above,*

$$\mathbb{E}\big[K_d(s, s')\big] = \sum_{L=1}^{T} a_L^2 \sum_{\substack{p \in \mathcal{P}_L(s \twoheadrightarrow *) \\ q \in \mathcal{P}_L(s' \twoheadrightarrow *)}} \left( \prod_{\ell=1}^{L} \kappa\big(\xi(e_\ell), \xi(f_\ell)\big) \right) \beta(\text{head}(p))\, \beta(\text{head}(q)), \tag{7}$$

*where $p = (e_1, \ldots, e_L)$ and $q = (f_1, \ldots, f_L)$. The right-hand side defines a PSD kernel $K$ on $V_{\mathsf{src}}$.*

Theorem 3 follows by expanding $z(s)^\top z(s')$, observing that mixed lengths cancel due to zero mean, and using layer-wise independence to factor expectations across edges. Thus, a randomly initialized NBFNet computes random features for the typed-path kernel $K$.

*Proof.* Expanding $z_k(s)z_k(s')$ and taking expectations over $\{g_k^{(\ell)}\}$, mixed-length terms vanish by zero mean; for equal lengths, independence across layers yields the product of second moments $\prod_{\ell=1}^L \kappa(\xi(e_\ell), \xi(f_\ell))$. Summing over paths and averaging over $k$ gives equation 7. PSD follows since $K$ is an expectation of Gram matrices. $\square$

**Step 3: Concentration at finite width.** Having identified the limiting kernel, we quantify how fast $K_d$ concentrates around $K$.

**Lemma 5** (Concentration). *Assume each $g_k^{(\ell)}(x)$ is subgaussian uniformly in $x$ with proxy $\sigma^2$, and set $\nu = \sup_x \kappa(x,x) < \infty$. Let $N_L(s \to *) = |\mathcal{P}_L(s \to *)|$. If $\sum_{L=1}^T a_L^2 \, \nu^L \, N_L(s \to *) < \infty$ for every $s$, then there exist constants $c, C > 0$ such that for all $\varepsilon > 0$,*

$$\Pr\big(\big|K_d(s,s') - \mathbb{E}K_d(s,s')\big| \geq \varepsilon\big) \leq 2\exp\big(-c\,d\,\varepsilon^2/C^2\big).$$

*Hence $K_d \to K$ in probability at rate $O_\mathbb{P}(1/\sqrt{d})$.*

Lemma 5 ensures that training only a linear classifier atop $z(\cdot)$ realizes a standard random-feature approximation to the RKHS induced by $K$. We now specialize the kernel choice to make the hashing connection explicit.

**Remark 3.** *Training* only *a linear classifier on $z(\cdot)$ implements random-feature learning for the kernel $K$; as $d \to \infty$, the solution converges to the kernel method in $\mathcal{H}_K$.*

**Step 4: Discrete edge tokens and the Dirac kernel.** Let $\tau(e) \in \Sigma$ be a discrete edge token (e.g., $\psi(e)$ or $(\phi(\mathrm{tail}(e)), \psi(e), \phi(\mathrm{head}(e)))$) and consider the Dirac (identity) kernel

$$\kappa_\delta(x, x') = \mathbf{1}\{\tau(x) = \tau(x')\}.$$

Realize $\kappa_\delta$ with *Rademacher codes* by drawing, for each layer $\ell$, random maps $r^{(\ell)} : \Sigma \to \{\pm 1\}^d$ independently across $\ell$ and setting $g_k^{(\ell)}(x) = r_k^{(\ell)}(\tau(x))$ [4]. Then Theorem 3 yields

$$\mathbb{E}\big[K_d(s,s')\big] = \sum_{L=1}^T a_L^2 \sum_{\substack{p \in \mathcal{P}_L(s \to *) \\ q \in \mathcal{P}_L(s' \to *)}} \mathbf{1}\{\tau(e_1) = \tau(f_1), \ldots, \tau(e_L) = \tau(f_L)\}\,\beta(\mathrm{head}(p))\,\beta(\mathrm{head}(q)).$$

$$(8)$$

**Bag-of-typed-paths view.** Let $\Psi_s \in \mathbb{R}^{\Sigma^{\leq T}}$ be the *bag of typed path sequences* out of $s$, where the coordinate for $\sigma = (\sigma_1, \ldots, \sigma_L)$ is

$$\Psi_s[\sigma] \;=\; a_L \times \big(\text{count of length-}L\text{ paths } p \text{ with } \tau(p) = \sigma, \text{ weighted by } \beta(\mathrm{head}(p))\big).$$

Then equation 8 is exactly the inner product $\mathbb{E}[K_d(s,s')] = \langle \Psi_s, \Psi_{s'} \rangle$. Moreover, each random coordinate implements a multiplicative sign code over sequences:

$$z_k(s) = \sum_{\sigma \in \Sigma^{\leq T}} \Psi_s[\sigma] \underbrace{\prod_{i=1}^{|\sigma|} r_k^{(i)}(\sigma_i)}_{=:\, r_k(\sigma)}.$$

$$(9)$$

---

[4]Normalization: we deliberately *omit* a $1/\sqrt{d}$ factor inside $g_k^{(\ell)}$; the outer $1/d$ in $K_d$ provides the correct scaling. Inserting $1/\sqrt{d}$ inside each layer would undesirably shrink longer paths by $d^{-L/2}$.

**Sparse CountSketch/TensorSketch realization.** We then bridge the bag-of-typed view to the countsketch algorithm. First, introduce pairwise independent hash functions

$$h^{(i)} : \Sigma \to [d], \qquad s^{(i)} : \Sigma \to \{\pm 1\}, \qquad i = 1, \ldots, T.$$

For a typed sequence $\sigma = (\sigma_1, \ldots, \sigma_L)$ define combined bucket and sign

$$H(\sigma) = \left(h^{(1)}(\sigma_1) + \cdots + h^{(L)}(\sigma_L)\right) \bmod d, \qquad S(\sigma) = \prod_{i=1}^{L} s^{(i)}(\sigma_i),$$

and the sketch $y(s) \in \mathbb{R}^d$ by

$$y_j(s) = \sum_{\sigma \in \Sigma^{\leq T}} \Psi_s[\sigma] \, S(\sigma) \, \mathbf{1}\{H(\sigma) = j\}, \qquad j \in [d]. \tag{10}$$

This is the standard *TensorSketch* construction specialized to sequences (metapaths). Then:

**Proposition 4** (Unbiased CountSketch of typed-path bags)**.** *With the construction in equation 10 using independent $h^{(i)}$ and $s^{(i)}$ with pairwise independence, we have*

$$\mathbb{E}\big[\langle y(s), y(s')\rangle\big] = \langle \Psi_s, \Psi_{s'}\rangle,$$

*and*

$$\mathrm{Var}\big(\langle y(s), y(s')\rangle\big) \lesssim \frac{\|\Psi_s\|_2^2 \, \|\Psi_{s'}\|_2^2}{d}.$$

*Proof.* Expand $\langle y(s), y(s')\rangle = \sum_j \sum_{\sigma,\sigma'} \Psi_s[\sigma]\Psi_{s'}[\sigma']S(\sigma)S(\sigma')\mathbf{1}\{H(\sigma) = H(\sigma') = j\}$ and take expectations. The sign hashes kill cross terms ($\sigma \neq \sigma'$) by zero mean, while bucket collisions contribute only when $H(\sigma) = H(\sigma')$; pairwise independence ensures these events occur with probability $1/d$ and cancel with the outer sum over $j$. Variance follows from standard CountSketch/TensorSketch analyses using limited independence. □

**How frozen NBFNet implements the sketch.** The dense realization (A) is exactly what equation 8 computes when $g_k^{(\ell)}(x) = r_k^{(\ell)}(\tau(x))$: each DP layer multiplies by layer-$\ell$ signs, and aggregation sums across paths—precisely the linear form in equation 9.

**Takeaway.** Steps 1–3 show that a randomly initialized NBFNet realizes random features for a typed-path kernel; Step 4 reveals that, under a Dirac edge kernel, those features are *precisely* CountSketch-style projections of the bag of typed metapaths reachable from $s$. In short: *frozen NBFNet = DP-powered CountSketch of typed-path counts*. This perspective clarifies both the inductive bias (which typed patterns are matched) and the approximation behavior (controlled by $d, T$, and the path growth rates).

## D MORE RELATED WORKS

### D.1 RELATIONAL DEEP LEARNING MODELS

To effectively address the challenges in RDB benchmarks, Robinson et al. (2024); Wang et al. (2024a) propose GNN-based pipelines with two main components: (1) transforming original tabular features into a unified latent space using type-specific encoders, and (2) aggregating latent features with a temporal-aware GNN conditioned on primary key-foreign key relationships. Chen et al. (2025a) further extended the message passing function to capture higher-order information by introducing atomic routes. Yuan et al. (2024) adapts the original GNN for recommendation tasks by implementing a path-based routing mechanism combining an ID-based GNN (Zhu et al., 2021) with shallow learnable embedding-based retrieval. Dwivedi et al. (2025) explores the potential of transformer-based backbones for RDL tasks; however, these currently require significantly more computational resources than GNN-based methods for limited performance gain, so we do not include them in our design space.

Wu et al. (2025); Wydmuch et al. (2024) investigate the potential of large language models for predictive tasks on RDBs. Currently, they exhibit a much lower performance-to-resource ratio than GNN-based methods, and it is difficult to evaluate the influence of duplication between pre-training

knowledge and downstream tasks. We thus leave their study to future work. At a broader scope, recent graph foundation model and graph-LLM works (Mao et al., 2024; Chen et al., 2023; Wang et al., 2025a) offer useful perspectives on transferability and pre-training, but their implications for temporally grounded SQL-defined RDB prediction remain under-explored.

Compared to these models trained from scratch, Fey et al. (2025); Wang et al. (2025b) propose foundation models for RDB tasks. Wang et al. (2025b) relies on a cross-table attention module that mimics DFS aggregation, but its performance cannot consistently outperform state-of-the-art GNN-based methods. Fey et al. (2025) utilizes a graph transformer-based backbone and delivers superior performance and in-context learning capabilities. However, it is not yet open-sourced, and training details are not revealed. With the help of tabular foundation models like TabPFN (Hollmann et al., 2023), it is also possible to achieve in-context learning by utilizing online DFS to achieve promising performance.

## D.2 AutoML for Graph Machine Learning

AutoML (Hutter et al., 2019) seeks to automate expert tasks—data engineering, model engineering, and evaluation—into an end-to-end machine learning pipeline. Model engineering typically encompasses neural architecture search (NAS) (Zoph & Le, 2017) and hyper-parameter optimization (HPO) (Bischl et al., 2023). Many works have adapted AutoML ideas to GML: for example, Gao et al. (2019) and Zhou et al. (2022) use reinforcement learning to search architectures, while Yoon et al. (2020) applies Bayesian optimization to improve search efficiency together with an algorithm budget constraint. One-shot NAS approaches first train a supernet and then prune it to obtain target architectures (Li & King, 2020; Guan et al., 2022; Qin et al., 2021), and Zhang et al. (2023) extends this paradigm to dynamic heterogeneous graphs. However, supernet-based methods are not well-suited to RDB settings due to the heterogeneity of model designs required there.

Beyond model-centric search, data-centric AutoML leverages dataset properties to guide selection. MetaGL (Park et al., 2023a;b) uses structural embeddings and graph statistics as task embeddings for meta-learned GNN selection. GraphGym and AutoTransfer (You et al., 2020; Cao et al., 2023) follow a knowledge-transfer strategy; AutoTransfer in particular constructs loss-landscape–based task embeddings and employs pre-trained embeddings to steer HPO. In contrast, our work is driven by empirical observations and targets architecture selection at both macro and micro levels, especially across heterogeneous model classes (RDL and DFS) in relational-database predictive tasks.

Bai et al. (2021); Luo et al. (2021); Lam et al. (2021) propose dedicated systems for AutoML over relational data. These efforts share DFS's motivation for automatic feature engineering but generally lack ready-to-use open-source implementations; accordingly, we adopt DFS as a representative, practical framework for automatic feature engineering.

Ranjan et al. (2024) studies post-hoc model selection and argues that picking what to deploy purely by the "best base validation score" is brittle.

## D.3 Real-world graph machine learning benchmarks

Benchmarking is essential for evaluating methods in graph machine learning. Representative datasets include the Open Graph Benchmark (Hu et al., 2020a) and TUDataset (Morris et al., 2020). However, recent work questions whether these benchmarks, like predicting the category of academic papers, reflect real-world tasks (Bechler-Speicher et al., 2025). To address this gap, benchmarking GNNs on relational-database (RDB) predictive tasks has become increasingly popular; notable examples are 4DBInfer (Wang et al., 2024a), H2GB (Lin et al., 2024), and RelBench (Robinson et al., 2024). In particular, RelBench provides a SQL-based framework to standardize task generation, which has helped it become a widely used benchmark for RDB predictive tasks.

In addition to model-centric benchmarks, recent works propose benchmarks focused on graph construction (Chen et al., 2025b; Choi et al., 2025). Jointly studying automatic graph construction and automatic model selection is a promising direction for future research.

# E  Supplementary information for models

In this section, we present more details on model designs, including model parameter design space and more training details.

## E.1  Discussion on the implementation difference between 4dbinfer and relbench

It is noteworthy to discuss the difference between the implementation details of 4dbinfer (Wang et al., 2024a) and relbench (Robinson et al., 2024), which are two main frameworks used to research RDB prediction tasks. We find that these implementation discrepancies are essential for a fair comparison between macro-level and micro-level architecture comparisons.

**Feature encoding part**. First, Relbench and 4DBinfer both adopt a type-specific encoder to project categorical/numerical/text into a latent space with the same dimension. However, after this transformation, Relbench further adopts a tabular encoder to transform the latent embeddings, which is not present in 4DBInfer. Furthermore, due to the implementation differences between Pyg and DGL, 4DBInfer does not utilize relative positional encoding when encoding temporal features. Another difference is that 4DBInfer will normalize all features. To align this with Relbench metrics, we save the scaler and do the inverse transform at the test stage.

**Neighborhood sampling part.** Both frameworks adopt a temporal sampling strategy to avoid temporal leakage. However, there is a difference in implementation details. For relbench, it adopts the temporal sampling of PyG, which generates disjoint subgraphs (when using the latest neighborhood sampling). This aligns with the time dynamics of the relbench task, such as user behavior over the past few months. For 4dbinfer, it is similar to standard subgraph-based sampling with a time mask.

**Evaluation setting.** There is also a difference in how the two frameworks do the evaluation, especially for the recommendation task. 4dbinfer adopts a pre-selected negative sample set, whereas relbench uses the entire target set as candidates. Moreover, 4dbinfer focuses on bipartite graphs, while for relbench, there are some tasks where the source and target lie between several-hop metapaths. This makes the method design not compatible across two types of problems. For example, any methods requiring pairwise information are not scalable for relbench settings.

## E.2  Detailed task feature designs

In this subsection, we introduce the detailed task feature designs.

**Simple heuristic performance** It characterizes the target entity distribution across different data splits. Specifically, we compute `entity_mean_val`, `entity_median_val`, `entity_mean_train`, and `entity_median_train`, which capture the central tendency of target entities in both validation and training sets.

For example, for entity mean, we first generate the following prediction,

```
1 fkey = list(train_table.fkey_col_to_pkey_table.keys())[0]
2 df = train_table.df.groupby(fkey).agg({task.target_col: "mean"})
3 df.rename(columns={task.target_col: "__target__"}, inplace=True)
4 df = pred_table.df.merge(df, how="left", on=fkey)
5 pred = df["__target__"].fillna(0).astype(float).values
```

Then we take the performance on the validation set as a feature.

**Homophily + stats + temporal** It aggregates 13 model-free heuristics designed to capture intrinsic properties of the relational structure and task characteristics without requiring any model training. This group synthesizes four distinct categories of structural features:

**(1) Homophily Features**: We compute five adjacency-based correlation statistics—`h_adjs_corr_mean`, `h_adjs_corr_max`, `h_adjs_corr_min`, `h_adjs_corr_mode`, and `h_adjs_corr_weighted_mean`. The weighted mean variant accounts for edge importance based on degrees.

**(2) Temporal Autocorrelation Features**: We include two lag-based autocorrelation measures—`lag1_autocorr_corr` and `lag2_autocorr_corr`—that capture temporal dependencies in time-aware relational tasks. These features measure whether target values at time $t$ correlate with values at $t-1$ and $t-2$, respectively.

**(3) DFS homophily Features**: This is the "homophily" metric calculated in the DFS manner. By doing a random walk from the task table, based on the retrieved context, we calculate how many target columns share the same label as the seed ones. `mean_same_class_ratio_ignore` computes the average proportion of same-class neighbors while ignoring unlabeled nodes; `adjusted_mean_same_class_ratio` provides a normalized version accounting for class imbalance; `sparsity_ratio` quantifies the density of the relational graph; and `mean_past_task_nodes` captures the average number of historical entities available for temporal tasks.

**(4) Stats**: We include `log_total_rows`, computed as $\log(1 + \text{train\_rows} + \text{val\_rows})$, which captures the logarithmic scale of the dataset. The log transformation ensures that the feature values are comparable across tasks with vastly different sizes.

**AutoTransfer Features**   We use 24 anchors in total. 12 for RDL and 12 for DFS.

**Model-Based Features**   It comprises eight performance indicators derived from lightweight model probes on the target task itself.

**(1) TabPFN Features**: We compute `tabpfn_1hop` and `tabpfn_2hop` by evaluating TabPFN on 1-hop and 2-hop neighborhood aggregations of the relational features.

**(2) Random Initialization Features**: We include six features derived from randomly initialized graph neural networks: `rfr_randomsage_1`, `rfr_randomsage_2`, `rfr_randomsage_3`, `rfr_randomnbfnet_1`, `rfr_randomnbfnet_2`, and `rfr_randomnbfnet_3`. These probes test whether the task's relational structure is inherently easy to exploit (even without learning), which can indicate task difficulty and the potential benefit of sophisticated architectures.

### E.3   DETAILED DESIGN SPACE

In this paper, we consider two classes of models: end-to-end learning (relational deep learning) models and non-parametric graph-based feature synthesis (DFS) models. Specifically, when the number of propagation hops for the latter is 1, the corresponding model will be a relation-agnostic one. We then elaborate on the module design inside each class.

**RDL.** Following wisdom in the design of the GNN architecture (You et al., 2020; Luo et al., 2024), we modularize the RDL module into the following parts: feature encoding, (optional) structural feature, message passing module, (optional) architecture design tricks, readout function, hyperparameters, and training objective. The design choices and rationales are detailed below.

1. For feature encoding, we stick to the ResNet-based encoder used in Robinson et al. (2024).

2. The structure feature is particularly useful for link-level prediction, which aims to break the original symmetry of the GNN designed for node-level tasks. We consider learnable embedding and partial labeling tricks because of their effectiveness demonstrated in existing benchmarks (Robinson et al., 2024; Yuan et al., 2024). Other features, such as random embeddings, are neglected because of their limited effectiveness.

3. For message passing, we consider all alternatives used in the existing literature to study the correlation between the message passing function and task performance.

4. For architecture design tricks, we consider residual connection (He et al., 2016) because of their effectiveness shown in Luo et al. (2024). However, under the RDB setting, we find that these tricks do not always improve performance and result in much more computation overhead.

5. Readout is another important module in architectural design. For entity-level tasks, we only consider MLP as the readout function. For link-level tasks, we consider ContextGNN and Shallow-Item (Yuan et al., 2024), which integrates graph-free learnable embeddings to

Table 8: Design space of a unified architecture for end-to-end RDL models. "Grid" indicates that the module is selected from a predefined grid, whereas "random" indicates that the module is randomly sampled.

| Module name | Possible choices |
|---|---|
| Feature encoding | ResNet (Robinson et al., 2024) |
| Structural features (grid) | Learnable embeddings (for the destination table, or for both the source and destination tables) (Yuan et al., 2024; Ma et al., 2024); partial-labeling tricks from NBFNet (Zhu et al., 2021) |
| Message passing (grid) | Sparse message passing: GraphSAGE (Robinson et al., 2024; Hamilton et al., 2017), HGT (Wang et al., 2024a; Hu et al., 2020b), PNA (Corso et al., 2020; Wang et al., 2024a); sparse message passing with higher-order information: Rel-GNN (Chen et al., 2025a) |
| Architecture design trick (random) | Residual connections (He et al., 2016) |
| Readout (grid) | MLP, ContextGNN, Shallow-Item (Yuan et al., 2024) |
| Hyperparameters (random) | Learning rate, weight decay, batch size, dropout rate, number of layers, hidden dimension, temporal sampling strategy, number of sampled neighbors |
| Training objective | Classification tasks: cross-entropy; regression tasks: MSE or MAE; link-level tasks: cross-entropy, BPR, or margin-based losses |

mitigate the pitfalls of GNNs on link-level tasks. It should be mentioned that all pairwise methods like NCN (Wang et al., 2024b), SEAL (Zhang & Chen, 2018) are not applicable because of the complexity.

6. Training objective is designed based on common loss functions for different task formats.

The detailed hyper-parameter search space is presented as follows:

```
1  RDL_SEARCH_SPACE = {
2      ## these will go through a grid search
3      "full_entities": {
4          'pre_sf': ['src_dst', 'zero_learn', 'none'],
5          'mpnn_type': ['relgnn', 'sage', 'hgt', 'pna'],
6          'post_sf': {'link': ['none', 'shallow', 'contextgnn'], 'node': ['
   none']}
7      },
8      "model_config": {
9          "encoder_num_layers": [4],
10         "torch_frame_model_cls": ['resnet'],
11         "batch_size": [128, 256],
12         "gnn_config": {
13             # src: learnable embedding for src, src_dst: learnable
   embedding for src and dst,
14             "loss_fn": {
15                 "binary_classification": ["bce"],
16                 "regression": ["mse", "mae"],
17                 "recommendation": ['bpr'],
18                 "multiclass_classification": ["ce"]
19             },
20             "hidden_channels": [64, 128, 256],
21             "num_heads": [1, 4],
22             "dropout": [0.0, 0.5],
23             "norm": ["layernorm", "batchnorm", 'none'],
24             "aggregation": ["mean", "sum"],
25             # jk is turned off because it almost always leads no
   performance gain with huge computation overhead
26             "jk": [False],
27             "skip_connection": [True, False]
28         },
```

```
29          "sampler_config": {
30              "temporal": ["uniform", "last"],
31              "num_neighbors": [32, 64, 128],
32              "num_layers": [1, 2, 3, 4],
33              "loader_type": ["node", "edge"]
34          },
35          "optimizer_config": {
36              "lr": [1e-3, 1e-4, 1e-2],
37              "weight_decay": [0.0, 1e-5],
38              "scheduler": ["exponential"],
39              "gamma": [.8, .9, 1.]
40          }
41      }
42 }
```

### E.4 GRAPH-INDUCED NON-PARAMETRIC FEATURE SYNTHESIS MODEL

**DFS.** The graph-induced non-parametric feature synthesis approach follows a different paradigm compared to end-to-end RDL models. Instead of learning parameters through gradient descent, DFS models leverage graph topology and statistical aggregation to synthesize features. We modularize the DFS framework into the following components: feature aggregation strategy, propagation depth, model backbone, and hyperparameters. The design choices and rationales are detailed below.

Table 9: Design space of graph-induced non-parametric feature synthesis (DFS) models. Grid means the module is selected from a predefined grid, while random means the module is randomly sampled.

| Module name | Possible choices |
|---|---|
| Feature aggregation strategy (fixed) | Mean, sum, max, min, count, weighted mean, target encoding |
| Propagation depth (grid) | Number of hops (1 = relation-agnostic): 1, 2, 3 |
| Downstream predictor (grid) | TabPFN (Hollmann et al., 2023), FT-Transformer, Light-GBM |
| Hyper-parameters (random) | Learning rate, weight decay, scheduler gamma, hidden dimension, number of attention heads, normalization |
| Training objective | Classification tasks: cross-entropy loss; Regression tasks: MSE or MAE |

1. Feature aggregation strategies determine how information flows through the graph structure. We follow Wang et al. (2024a) to adopt a fixed set of aggregation strategies.

2. Propagation depth controls the scope of information aggregation. When the number of hops is 1, the model becomes relation-agnostic and only uses entity-level features.

3. We consider three typical types of downstream task predictors: tabular foundation model, gradient boosting tree, and neural networks.

4. Training objectives are task-dependent and align with the evaluation metrics used in the benchmark datasets.

The detailed hyper-parameter search space is presented as follows: LightGBM and TabPFN, in general, do not require many hyper-parameters, so model-related hyper-parameters are only applied to FT-Transformer.

```
1 DFS_SEARCH_SPACE = {
2     ## dfs_layer will go through a grid search
3     "dfs_layer": [1,2,3],
4     "model_type": ["tabpfn", "ft_transformer", "lgbm"],
5     "batch_size": [128, 256],
6     "model_config": {
7         ## only for ft_transformer
8         "hidden_size": [128, 256, 512],
9         "dropout": [0.0, 0.5],
10        "num_layers": [1, 2, 3, 4],
```

```
11          "attn_dropout": [0.0, 0.5],
12          "num_heads": [1, 4],
13          "normalization": ["layernorm", "batchnorm", 'none'],
14          "loss_fn": {
15              "binary_classification": ["bce"],
16              "regression": ["mse", "mae"],
17              "ranking": ['bce', 'bpr'],
18              "multiclass_classification": ["ce"]
19          }
20      },
21      "optimizer_config": {
22          "lr": [1e-3, 1e-4],
23          "weight_decay": [0.0, 1e-5],
24          "scheduler": ["exponential"],
25          "gamma": [.8, .9, 1.]
26      }
27 }
```

### E.5    Supplementary experimental details

#### E.5.1    Experimental results for recommendation tasks

Here, we discuss the recommendation tasks skipped in the main text. First, we want to emphasize that the design space of recommendation tasks is much smaller since DFS-based methods cannot work well on recommendation tasks. The main reason is that the recommendation is more about capturing the collaborative signal across pairs of entities, which goes beyond the feature synthesis patterns of DFS. To make DFS work on recommendation tasks, we need to design common neighborhood-based or path-based features, which makes it no longer "automatic" but requires substantial feature engineering efforts.

In terms of RDL, we also need to emphasize that only a small portion of graph-related models can work under the RDB settings. Revisiting the traditional link prediction tasks on OGB (Hu et al., 2020a), the positive and negative sample pairs are usually pre-defined, with negative samples coming only from a small portion of the whole set. This makes it possible to use pairwise models like NCN (Wang et al., 2024b) and SEAL (Zhang & Chen, 2018). However, in RDB settings, the candidate set is usually the entire target table, which makes it impossible to use these pairwise models. That's why NBFNet (Zhu et al., 2021), a source-only model, is first considered in Yuan et al. (2024). Such a scalability problem also affects the implementation of vanilla GNN. Unlike using a link-level sampler in Wang et al. (2024a), we have to use two node-level loaders, one for the source type and one for the target type, for representation computation. These properties make it only possible to use vanilla GNN, shallow embedding, NBFNet, or a combination of them in the current stage.

We then present the HPO experiments based on these methods. As shown in Figure 5, we observe the following phenomena:

1. Overall, node-based loader dominates the link-based loader (or more accurately, the loader based on source and target types). One potential reason is that the node-based loader is closer to the idea of path-based retrieval, which is more effective in RDB recommendation tasks with rich relational paths. A potential exception is the REL-AMAZON datasets, which we do not use here. For these kinds of datasets whose path patterns are super sparse, the path-based collaborative signal may live in distant neighbors. As a result, on these tasks, we typically need a neighbor loader with more than 6 hops with dense neighbors to get good performance.
2. For the number of layers, unlike entity-level tasks, it presents that the deeper the better within a certain range. This is because, for path-based retrieval, deeper layers can capture more distant signals.
3. For message passing designs, it is also somewhat different from the phenomenon in entity-level tasks. Here, Relgnn presents clearly better performance. The reason is that there are some tables with multiple foreign keys. Semantically, these tables are closer to an edge, while in the PK-FK graphs, they are treated as nodes. RelGNN can simulate transforming these tables into

hyperedges, and thus makes the model capture more distant signals. HGT and PNA are better at capturing feature interactions, which are more important for entity-level tasks.

4. For structural features, partial labeling tricks of NBFNet are more effective. One noteworthy phenomenon is that on many tasks, a node-level loader without any structural features can also deliver good performance. The reason is that the original features in the graph can act as implicit type embeddings, which makes a multi-source path-based retrieval.

5. For readout functions, contextgnn does not always bring a performance boost. However, it will not degrade performance either. This is also based on the sparsity of path patterns.

**Takeaways for recommendation.** Across RelBench-style recommendation tasks, ContextGNN is a robust default that delivers competitive performance with modest tuning. This largely reflects task properties: path-aware retrieval with shallow ID embeddings plus GNN context works well when collaborative signals are captured via multi-hop relational paths. On some other recommendation tasks on bipartite graphs, two-tower (dual-encoder) architectures may scale training and inference more effectively and simplify candidate generation, though they typically require task-specific components (e.g., hard-negative mining, retrieval infrastructure, and reranking) to reach top accuracy. Overall, these observations suggest that for recommendation, fully automatic architecture design may be less effective than crafting a task-tailored framework—consistent with prevailing industrial practice.

Table 10: Validation-selected and test-selected performance gap. For RDL, we show the top 2 architectures with the largest gap.

| Method name | Mean test clf perf | Mean val clf perf | Mean gap |
|---|---|---|---|
| RDL Overall | 78.21 | 76.73 | 1.49 |
| Sage (top1) | 76.90 | 72.77 | 4.13 |
| HGT (top2) | 77.19 | 74.95 | 2.24 |
| DFS overall | 76.90 | 75.22 | 1.68 |
| TabPFN | 75.91 | 75.70 | **0.21** |
| FT-Transformer | 75.87 | 74.45 | 1.42 |

| Method name | Mean test reg perf | Mean val reg perf | Mean gap |
|---|---|---|---|
| RDL Overall | 6.9292 | 7.0578 | 0.1287 |
| HGT (top1) | 6.9407 | 7.1036 | 0.1630 |
| PNA (top2) | 7.1376 | 7.1795 | 0.0419 |
| DFS overall | 3.6881 | 3.7135 | 0.0254 |
| TabPFN | 3.7544 | 3.7692 | **0.0148** |
| FT-Transformer | 4.0630 | 4.0883 | 0.0254 |

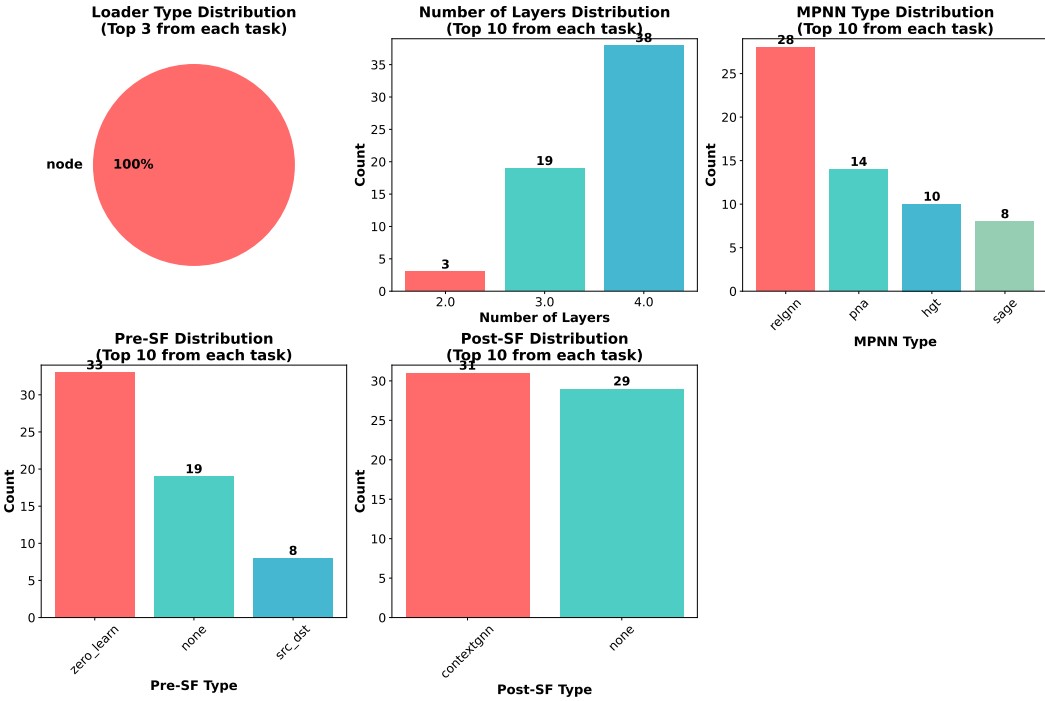

Figure 5: HPO results for recommendation tasks.

### E.5.2 SUPPLEMENTARY EXPERIMENTAL RESULTS FOR THE MAIN TEXT

**Full experimental results for Figure 2.** The full result for Figure 2 is presented in Table 11.

**Influence of micro-level design choices.** Looking further into the influence of micro-level architecture choices shown in Figure 6, we can observe that: (1) different design choices present in the top performing configurations, underscoring the importance of architecture design; (2) compared to

Table 11: Full experimental results for Figure 2.

| Task | Type | RelGNN | RelGT | GraphSAGE | Rel-LLM | KumoRFM (icl) | KumoRFM (fine-tuned) | RDL (val-selected) | RDL (ours) | DFS (val-selected) | DFS (ours) | Best (ours) | Griffin |
|---|---|---|---|---|---|---|---|---|---|---|---|---|---|
| driver-top3 | classification | 85.69 | 83.52 | 75.54 | 82.22 | 91.07 | 99.62 | 82.41 | 85.94 | 84.70 | 85.71 | 85.94 | 77.95 |
| driver-dnf | classification | 75.29 | 75.87 | 72.62 | 77.15 | 82.41 | 82.63 | 74.35 | 77.20 | 76.89 | 79.42 | 79.42 | 70.91 |
| driver-position | regression | 3.798 | 3.920 | 4.022 | 3.967 | 2.747 | 2.731 | 4.0491 | 3.8029 | 3.3660 | 3.2730 | 3.2730 | 4.2000 |
| user-churn | classification | 70.93 | 69.27 | 69.88 | 70.55 | 67.71 | 71.23 | 70.98 | 70.98 | 65.29 | 68.23 | 70.98 | 68.04 |
| item-sales | regression | 0.0540 | 0.0536 | 0.0560 | 0.0520 | 0.0400 | 0.0340 | 0.0511 | 0.0509 | 0.0780 | 0.0750 | 0.0509 | 0.0810 |
| post-votes | regression | 0.0650 | 0.0654 | 0.0650 | 0.0620 | 0.0650 | 0.0650 | 0.0665 | 0.0651 | 0.0680 | 0.0660 | 0.0651 | 0.0622 |
| user-engagement | classification | 90.75 | 90.53 | 90.59 | 91.21 | 87.09 | 90.70 | 88.95 | 90.56 | 78.47 | 87.28 | 90.56 | 87.56 |
| user-badge | classification | 88.98 | 86.32 | 88.86 | 89.64 | 80.00 | 89.86 | 88.41 | 88.51 | 85.17 | 86.47 | 88.51 | 85.99 |
| user-repeat | classification | 79.61 | 76.09 | 76.89 | 79.26 | 76.08 | 80.64 | 81.25 | 82.89 | 77.20 | 79.26 | 82.89 | 77.93 |
| user-ignore | classification | 86.18 | 81.57 | 81.62 | 83.74 | 89.20 | 89.43 | 83.66 | 86.77 | 77.20 | 84.43 | 86.77 | 82.35 |
| user-attendance | regression | 0.2380 | 0.2500 | 0.2580 | 0.2510 | 0.2640 | 0.2380 | 0.2397 | 0.2397 | 0.2630 | 0.2380 | 0.2380 | 0.3336 |
| user-visits | classification | 66.18 | 66.78 | 66.20 | 67.01 | 64.85 | 78.30 | 66.77 | 66.87 | 65.29 | 66.74 | 66.87 | 64.68 |
| user-clicks | classification | 68.23 | 68.30 | 65.90 | 66.74 | 64.11 | 66.83 | 67.16 | 68.77 | 62.34 | 69.19 | 69.19 | 63.30 |
| ad-ctr | regression | 0.0370 | 0.0345 | 0.0410 | 0.0370 | 0.0350 | 0.0340 | 0.0346 | 0.0340 | 0.0380 | 0.0370 | 0.0340 | 0.0639 |
| study-outcome | classification | 71.24 | 68.61 | 68.60 | 71.04 | 70.79 | 71.76 | 71.41 | 74.13 | 70.59 | 71.82 | 74.13 | 69.08 |
| study-adverse | regression | 44.681 | 43.990 | 44.473 | 43.682 | 58.231 | 44.225 | 44.5706 | 43.9880 | 49.9500 | 44.1100 | 43.9880 | 45.2100 |
| site-success | regression | 0.3010 | 0.3260 | 0.4000 | 0.3970 | 0.4170 | 0.3010 | 0.3932 | 0.3236 | 0.3910 | 0.3490 | 0.3236 | 0.3765 |

older models like HGT and PNA, RelGNN does not present advantages in terms of prediction performance, which inspires us to revisits the wisdom of past research. (3) For RDL-based methods, learnable embeddings are mainly required to achieve top performance. (4) For DFS-based methods, the FT-Transformer works better for large-scale tasks, while TabPFN can well fit small-scale ones. Tree-based methods, such as LightGBM, are not optimal in most cases; therefore, we do not consider them in the following text.

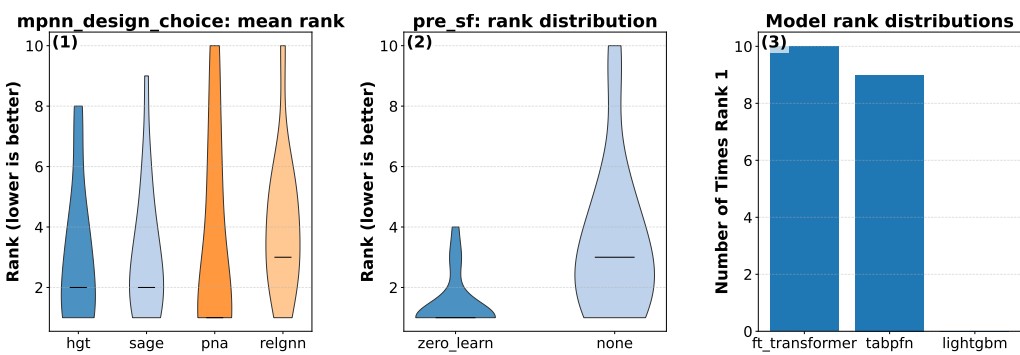

Figure 6: Relationship between test performance ranking and micro-level architecture choices. For RDL-based methods, we filter the original search result with the top 10 performing configurations on each task. For DFS-based methods, we filter the original search result with the top 1 performing configurations on each task. The best configurations are selected based on test performance directly. "zero_learn" is the labeling trick adopted by NBFNet. (a) A violin plot of MPNN types shows that PNA achieves the best mean ranking across different tasks. (b) When comparing labeling tricks, although equivariant models appear more frequently in top rankings, a labeling trick is still needed to achieve the top spot. (c) For DFS, ft_transformer and tabpfn present unique strengths, where the former can leverage more training samples, and the latter usually works better in small-scale settings and can conduct in-context learning.

**Performance gap between validation-selected and test-selected configurations.** As shown in Table 10, we can see that both RDL and DFS suffer from this performance gap. Specifically, RDL presents a much larger gap for regression tasks. TabPFN, the tabular foundation model utilizing in-context learning for inference, shows an advantage in mitigating such performance drift.

**Analysis of affinity-based features.** Here, we show some empirical results on the analysis of affinity-based features and empirical performance. For each task from the model performance bank, we have three task-level *anchor* scores (TabPFN, RandomSAGE, RandomNBFNet) and mean test performance for two model families (RDL, DFS). Anchors are constant within a task; performances are task-wise means over validated runs. For RDL we also compare two preprocessing options, pre_sf $\in \{$zero_learn, none$\}$; define the per-task difference $\Delta = \text{RDL}_{\text{zero\_learn}} - \text{RDL}_{\text{none}}$.

*(i) RDL vs DFS from TabPFN vs graphs.* Using all tasks, a log–log fit shows

$$\log(\text{RDL}/\text{DFS}) \approx 0.091 - 0.262 \log(\text{TabPFN}/\text{NBFNet}) \quad (R^2 \approx 0.58,\ n = 19),$$

so when TabPFN exceeds the graph anchors, DFS tends to outperform RDL; simple thresholds TabPFN/NBFNet $\geq 1.10$ or TabPFN/SAGE $\geq 1.18$ classified the winner at about 79% accuracy. *(ii) RDL pre_sf from graph–graph ratio.* With $R = \max(\text{NBFNet})/\max(\text{SAGE})$, the

linear association with $\Delta$ is small (Pearson $\approx -0.114$, $n = 19$), but as a one-bit chooser it is useful: AUC(`zero_learn better`) $\approx 0.718$, and the rule $R \geq 0.977 \Rightarrow$ choose `zero_learn` (else `none`) attains $\sim 0.789$ accuracy (base rate $\sim 0.684$ favoring `zero_learn`).

**Visualization of loss landscape.** Our first-step analysis is to plot the loss landscape of a series of models presenting different val-selected and test-selected performance gaps. An example is shown in Figure 7. We can see that the DFS, which generalizes better, presents a much flatter loss landscape.

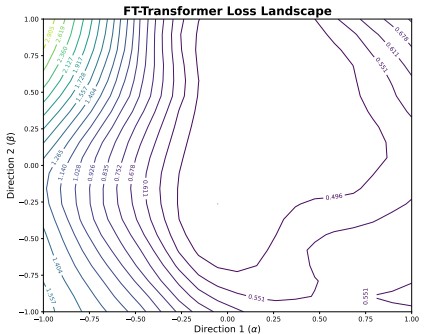
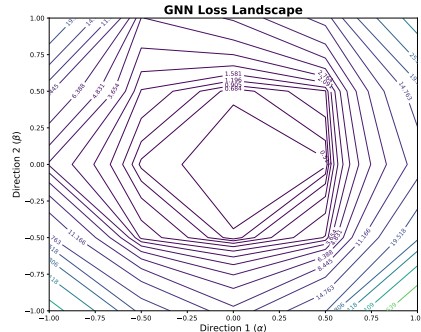

(a) Loss landscape of DFS + FT-Transformer on DRIVER-TOP3

(b) Loss landscape of RDL on DRIVER-TOP3

Figure 7: Loss landscapes

### E.6 EFFICIENCY

In the main text, we skip the discussion of efficiency-related concerns, such as running time and memory consumption. One reason is that efficiency depends on the backbone implementation. For example, we implement SQL using in-memory databases in this work. One cannot say DFS or RDL is more efficient merely based on this implementation. In reality, DFS can potentially be accelerated via tools like Spark. Nonetheless, we still present efficiency-related results here, with the following main contents: (1) Average running time of RDL and DFS pipelines, which includes the time for model training and dataset pre-processing. (2) The way to extend our current methods to incorporate efficiency-related concerns.

As shown in Table 12, we consider the running time of two representative tasks: driver-dnf and study-outcome. The former database is a small-scale one, while the latter contains lots of columns. We consider the simple RDL models SAGE and the complicated ones HGT. For DFS, we consider three propagation depths: 1, 2, and 3. We can see that DFS is relatively more efficient when the propagation depth is small. The main backbone is just the feature encoder part of the RDL, so it will be much faster during the inference stage. Moreover, without our proposed PCA compression strategy, DFS is usually unusable for large-scale tasks. Generally, both RDL and DFS do not meet significant efficiency concerns.

Table 12: Preprocessing and training times by method for driver-dnf and study-outcome.

|  | driver-dnf | | | | | study-outcome | | | | |
|---|---|---|---|---|---|---|---|---|---|---|
|  | RDL (SAGE) | RDL (HGT) | DFS-1 (no p/p) | DFS-2 (no p/p) | DFS-3 (no p/p) | RDL (SAGE) | RDL (HGT) | DFS-1 (no p/p) | DFS-2 (no p/p) | DFS-3 (no p/p) |
| Preprocessing time | 60 s | 60 s | 7 s/12 s | 18 s/17 s | 310 s/41 s | 240 s | 240 s | 240 s/36 s | 353 s/48 s | 965 s/95 s |
| Training time (per epoch) | 6 s | 8 s | < 1 s/< 1 s | < 1 s/< 1 s | < 1 s/< 1 s | 10 s | 11 s | < 1 s/< 1 s | < 1 s/< 1 s | < 1 s/< 1 s |

To extend our current methods to incorporate efficiency-related concerns, we consider the following two strategies: (1) Rule-of-thumb. Since we know the number of training samples, when the scale is limited, then directly utilizing TabPFN and DFS is usually the most efficient approach. Moreover, for RDL, HGT is obviously the most expensive model considering its complicated attention mechanism. (2) Joint optimization of efficiency and effectiveness. We can consider a multi-objective optimization framework, where we can consider the validation performance and training time as two objectives. First, we can train a meta-model to predict the training time based on architecture designs. For example, we can list the following efficiency-related hyperparameters: number of lay-

ers, hidden dimension, number of attention heads, and batch size. Then, we can train a model whose input feature is the hyper-parameter configuration, and the output is the training time. To estimate the pre-processing time of DFS, it is approximately proportional to the number of SQL operations multiplied by the size of training tables. We then demonstrate one formula to do joint optimization of efficiency and effectiveness.

$$
\begin{aligned}
(\pi^*, \theta^*) = \operatorname*{arg\,max}_{\substack{\pi \in \{\mathrm{RDL,\,DFS}\} \\ \theta \in \mathcal{S}_\pi}} \quad & \widehat{\mathrm{Perf}}(\theta, \pi) \\
& - \lambda \cdot \frac{\mathbb{1}[\pi = \mathrm{DFS}]\, c_{\mathrm{sql}} \cdot \mathrm{ops}(\theta) \cdot \mathrm{rows}}{T_{\mathrm{ref}}} \\
& - \lambda \cdot \frac{\mu_{\mathrm{time}}(\theta, \pi) + \beta\, \sigma_{\mathrm{time}}(\theta, \pi)}{T_{\mathrm{ref}}}
\end{aligned}
$$

$\pi$ chooses the pipeline (RDL vs. DFS) and restricts the search space to $\mathcal{S}_\pi$; $\widehat{\mathrm{Perf}}(\theta, \pi)$ is predicted validation effectiveness; $\lambda > 0$ trades time for performance; $T_{\mathrm{ref}}$ normalizes time; the DFS pre–processing cost is activated by $\mathbb{1}[\pi = \mathrm{DFS}]$ and modeled as $c_{\mathrm{sql}} \cdot \mathrm{ops}(\theta) \cdot \mathrm{rows}$; $\mu_{\mathrm{time}}(\theta, \pi)$ and $\sigma_{\mathrm{time}}(\theta, \pi)$ are the meta–model's mean and uncertainty for training time; $\beta \geq 0$ adds risk aversion to slow/uncertain runs. $\pi$ can be trained based on a model performance bank similar to the meta-predictor.

