# OpenReview forum: "Relatron: Automating Relational Machine Learning over Relational Databases"
_ICLR.cc/2026/Conference — ICLR 2026 Poster_

### Official Review · Reviewer_Yptq · 2025-10-31

**Soundness:** 3
**Presentation:** 3
**Contribution:** 2
**Rating:** 4
**Confidence:** 4

**Summary:**

This paper presents an empirical study comparing Relational Deep Learning (RDL) and Deep Feature Synthesis (DFS) for predictive modeling on relational databases. The authors construct a large-scale design space for both model families and build a "model performance bank" by running an architecture-centric search across numerous RDB tasks. Relatron uses diffferent signals in a meta-predictor to first choose between RDL and DFS (macro-selection) and then prune the search within that family.

**Strengths:**

- The paper targets an important problem, relational deep learning (RDL), in the machine learning.
- The experimental analysis across models and datasets is interesting.

**Weaknesses:**

- The recommendation task is not involved. Only classification and regression task types from Relbench.
- The results of a foundation model, KumoRFM, outperforms Relatron that selects between models. This questions the practical usage of Relatron. Since there is already a stronger foundation model, is it still necessary to design and train a model selector?

**Questions:**

- The paper provides a theoretical argument in Appendix C.2 for RDL's strength in low-homophily settings. It would be better to explain the other, more surprising half: why DFS is so strong in high-homophily regimes.
- Could Relatron also include KumoRFM, the current state-of-the-art foundation model, into its selection pool?

---

> ### Author Response · Authors · 2025-11-22
> **Response to Q1**
>
> **The recommendation task is not involved. Only classification and regression task types from Relbench.**
>
> We consider recommendation tasks in the paper. Because of space limitations in the main text, we present these results in Appendix E 4.1.
>
> We briefly summarize our observations here:
> 1. DFS does not apply to relbench-style recommendation tasks. You need to hand-craft structural features like a common neighborhood, which only works for bipartite graphs like Movielens, but not for RDBs from Relbench.
> 2. Unlike entity-level tasks, where task-specific model selection is necessary and beneficial, RelGNN, augmented with NBFNet's labeling trick (backbone) and ContextGNN (readout head), generally achieves the best performance across all RDB recommendation tasks in the Relbench setting.
> 3. For the number of layers, unlike entity-level tasks, it presents a phenomenon that the deeper the better within a specific range. For path-based retrieval, deeper layers can capture more distant signals.
>
>
> These phenomena are related to the task properties of Relbench. Compared to a traditional recommendation task, the source and target entities don't always form a bipartite graph (like Movielens). For example, in rel-trial tasks, you need a 4-hop walk to reach the target. This makes path-based retrieval methods, such as NBFNet, effective. Meanwhile, traditional graph link prediction methods based on common neighbors (link-level models), such as NCN and BUDDY, are not applicable here because the number of target entities is usually much larger than that of source entities, and only node-level models, such as NBFNet or ID embedding, are applicable.
> As a result, the central theme of the recommendation is possibly designing specialized models rather than selecting a pre-defined model. For example, the ContextGNN head uses path-based ensembling (if the target can't be reached via a path, it uses a learnable embedding to retrieve). On databases with rich signals, such as rel-stack or rel-trial, the NBFNet branch works well. On databases like rel-amazon with sparse signals, the NBFNet branch requires more than 6 layers to achieve non-trivial results. Then, the learnable shallow embedding can be used to retrieve targets. As a result, it can be a versatile solution for relbench-style recommendations at this stage.
>
> If we want to study model selection with some budget constraints, for example, whether a 4-layer model can work well for rel-amazon, then we may use the metric locality [1] (here $s$ refers to the locality, k refers to h hop, $\mathcal{L}$ is the source entity set, $\mathcal{R}$ is the target entity set, $\mathcal{N}_k$ is the k-hop neighborhood. $\mathcal{Y}_v$ is the positive link)
>
> $$s_k^{(T, T+i]}=\frac{1}{|\mathcal{L}|} \sum_{v \in \mathcal{L}}=\frac{\left|\mathcal{N}_k^{(-\infty, T]}(v) \cap \mathcal{R} \cap \mathcal{Y}_v^{[T, T+i)}\right|}{\left|\mathcal{Y}_v^{[T, T+i)}\right|}$$
>
> which measures the ratio of links that fall within the k-hop neighborhood of the source entity, to train a meta-predictor. As shown in the following table, it can somehow explain why RelGNN is generally better.  Taking user-post-comment as an example, we can see that converting an RDB using the "table-to-node" or "table-to-node-or-hyperedge" strategy, the latter strategy gets clearly better locality, and thus better recommendation performance. RelGNN actually just shares a similar philosophy.
>
> |                                     | user-post-comment |
> | ----------------------------------- | ----------------- |
> | Table-to-node locality              | 0.3034            |
> | Table-to-node-or-hyperedge locality | 0.3229            |
> | SAGE + NBFNet performance           | 12.72             |
> | RelGNN + NBFNet performance         | 14.00             |
>
>
> [1] Yuan, Yiwen, et al. "ContextGNN: Beyond two-tower recommendation systems." arXiv preprint arXiv:2411.19513 (2024).

---

> ### Author Response · Authors · 2025-11-22
> **Response to Q2**
>
> **The results of a foundation model, KumoRFM, outperforms Relatron that selects between models. This questions the practical usage of Relatron. Since there is already a stronger foundation model, is it still necessary to design and train a model selector?**
>
> Thanks for this good question. We think the study in Relatron is still valuable because: 1. KumoRFM doesn't reveal technical details and isn't always effective (especially in the ICL setting); 2. We provide a principled framework for understanding the weaknesses of both simple and foundation models. 3. Our model can help with the selection of foundation models, too. (We put this part in the last response.)
>
> **KumoRFM is still not perfect** First, KumoRFM is a closed-source, closed-weight model that can only be accessed through its API. Moreover, they only offer an in-context learning interface. If we look at their "magic" fine-tuning performance, we can see that on tasks like driver-top3, the ROC-AUC is already 99, suggesting some overlap between pre-training and downstream tasks. If we look at their ICL performance, we can see that the current KumoRFM still presents limitations:
> 1. Compared to DFS+TabPFN, which doesn't have any RDB-related pre-training, it is outperformed on 4/8 tasks below.
> 2. KumoRFM performs poorly on the user-badge task—a heterophilous problem where, in theory, RDL models (like KumoRFM, which is based on a relational graph transformer [1]) should outperform DFS when there are plenty of training labels. Despite this, KumoRFM’s results are even worse than those of the DFS+TabPFN approach. This suggests KumoRFM’s in-context learning (ICL) may only support a limited context length (fewer than 1,000). Meanwhile, training a GraphSage model from scratch can yield better performance (>85) with less than 1,000 training samples. The new TabPFN v2.5 now supports context lengths exceeding 50,000, while scaling RDL architectures remains difficult due to the need for online graph sampling. Beyond Relbench, many tasks require much larger context lengths than 1,000, further highlighting the scaling limitations of KumoRFM.
>
> | RDB name   | rel-f1     | rel-f1      | rel-hm     | rel-trial     | rel-stack  | rel-stack       | rel-avito  | rel-avito   |
> | ---------- | ---------- | ----------- | ---------- | ------------- | ---------- | --------------- | ---------- | ----------- |
> | task name  | driver-dnf | driver-top3 | user-churn | study-outcome | user-badge | user-engagement | user-visit | user-clicks |
> | KumoRFM    | 81.98      | 90.64       | 68.82      | 71.54         | 84.09      | 86.9            | 64.85      | 64.11       |
> | DFS+TabPFN | 76.87      | 82.24       | 67.72      | 71.82         | 84.79      | 82.98           | 66.28      | 66.44       |
>
> **Second, our analysis can inspire the design of relational foundation models**. RDL and DFS are general concepts that can cover all existing relational foundation models. For example, RT [2] can be understood as a neural DFS. As a result, we can understand why, after pre-training, their performance on specific tasks on rel-avito and rel-stack still lags behind RDL training from scratch. Theoretically, keeping both RDL and DFS-like may be necessary to work well on all tasks. This suggests two potential designs: 1. NBFNet-like attention designs, which try to inject those inductive logic programming-like path inductive bias; 2. An MoE-like architecture that can do ensembling wisely. A pre-trained model selector, together with multiple relational foundation models with different architectures, can be a potential solution.
>
> Third, our framework has the potential to help with relational foundation model selection. We show this in response to Q4. Based on these points, it's still necessary to study and understand how to do model selection.
>
> [1] Dwivedi, Vijay Prakash, et al. "Relational Graph Transformer." arXiv preprint arXiv:2505.10960 (2025).
> [2] Ranjan R, Hudovernik V, Znidar M, et al. Relational Transformer: Toward Zero-Shot Foundation Models for Relational Data[J]. arXiv preprint arXiv:2510.06377, 2025.

---

> ### Author Response · Authors · 2025-11-22
> **Response to Q3**
>
> **The paper provides a theoretical argument in Appendix C.2 for RDL's strength in low-homophily settings. It would be better to explain the other, more surprising half: why DFS is so strong in high-homophily regimes.**
>
> Thanks for this good question. 1. We want first to clarify the observations with the newly added tasks and the new evaluation. 2. We explain the theoretical insights for this new observation.
>
> **New experiments and observations** During the submission phase of this paper, we are also curious why DFS can be so good for regression tasks. So, we introduce more regression tasks and try different evaluation settings. We then have the following observations: 1. If you fully rely on MAE to select checkpoints for regressions, you may end up with some models with "overfitted" MAE while very low R2. This may make DFS "overly" high in some tasks. We have changed the strategy to select checkpoints based on both MAE and R2 (MAE is a good metric for comparison, but you can't fully rely on it when you apply clamps). It seems some recent works also noticed this problem [1]. 2. The original Gaussian kernel-based homophily doesn't present a strong correlation with RDL-DFS performance gap on regression tasks. Instead, we find that the correlation between center and neighbor labels is a much stronger signal. Based on this new study, we have the following conclusion:
> 1. RDL is more "expressive" than DFS, while it can only present this strength in a non-low-data regime
> 2. In a low data regime, DFS can effectively eliminate noise. Moreover, this extends to newer models like RT [1], which learns a neural DFS. As long as the aggregation is linear (whether predefined or parametric), this limit holds. **As a result, DFS is not better at high-homophily tasks; it is better at low-data regimes. ****
>
> We can validate this by conducting a controlled study on a low-homophily task using a user-badge from rel-stack; we can see that in the low-data regime, DFS is still better despite low homophily. Then, RDL's performance grows much faster while DFS's performance saturates.
>
> | Database name | task name  | Method | Perf (100) | Perf (500) | Perf (10000) |
> | ------------- | ---------- | ------ | ---------- | ---------- | ------------ |
> | rel-stack     | user-badge | Sage   | 75.39      | 83.67      | 86.35        |
> | rel-stack     | user-badge | DFS    | 77.62      | 77.22      | 82.18        |
>
> For some informal insights, DFS can beat RDL in the low-data regime by trading off expressiveness for *sample efficiency*. From a learning-theory view, RDL has a much larger hypothesis space: it doesn’t just learn a predictor on top of fixed relational features, it also learns how to combine relations, how strongly to trust each metapath, and how to gate or flip heterophilous neighbors. That gives RDL, in principle, a lower *approximation error* (it can represent the Bayes-optimal, relation-wise-gated rule and even simulate DFS as a special case). Still, it also means higher *estimation error*: you need many more labeled examples to reliably pin down all those parameters without overfitting. DFS, by contrast, has a much smaller adequate capacity because the relational part is hard-coded and only a compact tabular head is learned, so classical generalization bounds (VC/Rademacher/bias–variance story) favor it when the number of training rows is small. Intuitively, homophily and degree control how many “clean coin flips” you see per relation; when this evidence is weak or scarce, RDL can’t confidently decide which relations to flip or ignore, and its extra flexibility just turns into noise, while the simpler DFS estimator stays more stable and therefore wins in the low-data regime.
>
> We will soon also update the theoretical part of the paper.

---

> ### Author Response · Authors · 2025-11-22
> **Response to Q4**
>
> **Could Relatron also include KumoRFM, the current state-of-the-art foundation model, into its selection pool?**
>
> Thanks for this good question. It's possible to route foundation models. First, we want to discuss what the main challenge is here. Second, we show some preliminary results.
>
> **Challenge of extension:** In principle, we can extend Relatron to support selection across different foundation models. The main challenge lies in 1. pre-training data. It's hard to check whether there's downstream task leakage. Different foundation models use totally different pre-training data. 2. Technical details of some foundation models haven't been revealed—for example, the context length of KumoRFM.
> 1. For TabPFN, the pretraining is fully conducted on synthetic data. The pre-training data is only for prior fitting, so there's no leakage problem.
> 2. For the Relational transformer [1], the pre-training is conducted on the Relbench datasets with leave-one-out evaluation.
> 3. For KumoRFM [2], no details have been revealed. As shown in their whitepaper, they may use a mix of synthetic and real-world data. On tasks like driver-top3, the ROC-AUC is already 99, suggesting some overlap between pre-training and downstream tasks. Moreover, the fine-tuning effects for RT and KumoRFM are not the same. For RT, fine-tuning has no clear help (see their Table 7). While for KumoRFM, fine-tuning greatly surpasses ICL performance. Data leakage and "magic finetuning" are not something a router module can capture until KumoRFM releases more details.
>
> **Some preliminary experiments:** We then apply Relatron to the selection of relational foundation models across two controlled settings (with similar pre-training datasets).
> 1. The first experiment is to extend Relatron to the backbone models of these foundation models, say RelGT (for KumoRFM) and RT.
> 2. The second experiment is to test Relatron for real foundation models (ICL settings).
>
>
> | RDB name         | rel-f1     | rel-f1      | rel-hm     | rel-trial     | rel-stack  | rel-stack       | rel-avito  | rel-avito   |
> | ---------------- | ---------- | ----------- | ---------- | ------------- | ---------- | --------------- | ---------- | ----------- |
> | task name        | driver-dnf | driver-top3 | user-churn | study-outcome | user-badge | user-engagement | user-visit | user-clicks |
> | RT               | 78.7       | 84.9        | 69.9       | 68.6          | 88.5       | 90.0            | 65.0       | 63.6        |
> | RT-no-target     | 76.46      | 83.98       | 66.63      | 68.6          | 83.74      | 85.38           | 63.87      | 64.9        |
> | RelGT            | 75.87      | 83.52       | 69.27      | 68.61         | 86.32      | 90.53           | 66.78      | 68.30       |
>
> When checking the winner between RT-no-target (we wanna a fair comparison with both not using this trick) and RelGT, our features achieve 0.875 LOO AUC. This is good performance considering there are very close cases like study-outcome.
>
>
> | RDB name    | rel-f1     | rel-f1      | rel-hm     | rel-trial     | rel-stack  | rel-stack       | rel-avito  | rel-avito   |
> | ----------- | ---------- | ----------- | ---------- | ------------- | ---------- | --------------- | ---------- | ----------- |
> | task name   | driver-dnf | driver-top3 | user-churn | study-outcome | user-badge | user-engagement | user-visit | user-clicks |
> | KumoRFM     | 81.98      | 90.64       | 68.82      | 71.54         | 84.09      | 86.8            | 64.85      | 64.11       |
> | RT-ICL      | 81.2       | 89.3        | 63.3       | 54.6          | 81.1       | 86.9            | 62.6       | 60.9        |
> | DFS+TabPFN  | 76.87      | 82.24       | 67.72      | 71.82         | 84.79      | 82.98           | 66.28      | 66.44       |
> | Griffin [3] | 70.91      | 77.95       | 68.04      | 69.08         | 85.99      | 87.56           | 64.68      | 63.30       |
>
> * For KumoRFM vs RT-ICL, our method achieves 100% accuracy, which is reasonable since RT-ICL only wins on one task.
> * For DFS+TabPFN vs Griffin, our method achieves 0.867 AUC if we remove the train size features there. This is reasonable since TabPFN and Griffin use different context sizes (Griffin is in the fine-tuning setting), so train size can be misleading there.
>
> As a result, we can see Relatron presents potential for the selection. In a nutshell, the principles for selection are consistent: RDL presents better expressiveness while requiring more training data. The problem is just how we learn a decision boundary based on these performance data.
>
> [1] Ranjan R, Hudovernik V, Znidar M, et al. Relational Transformer: Toward Zero-Shot Foundation Models for Relational Data[J]. arXiv preprint arXiv:2510.06377, 2025.
> [2] Fey, Matthias, et al. "KumoRFM: A Foundation Model for In-Context Learning on Relational Data."
> [3] Wang, Yanbo, et al. "Griffin: Towards a Graph-Centric Relational Database Foundation Model." arXiv preprint arXiv:2505.05568 (2025).

---

### Official Review · Reviewer_Eebt · 2025-11-01

**Soundness:** 4
**Presentation:** 4
**Contribution:** 3
**Rating:** 8
**Confidence:** 4

**Summary:**

This paper lays out a design space of modeling approaches for machine learning (ML) on relational database (RDB) tasks, and does a comprehensive benchmarking study on it. The findings suggest that different tasks require different design choices (including model architecture). To automate this choice the paper proposes a few metrics which have high correlation with the performance of various approaches on different tasks and can be used to predict the best method. This AutoML system (Relatron) achieves performance improvements over hyperparameter-tuning and auto-transfer baselines, while being more efficient computationally.

**Strengths:**

* The paper is thorough in its parameterization of the design space.
* The findings are interesting. I like the experiment and finding that validation metrics are unreliable as this is very important for temporal splits as in leading relational benchmarks.
* The metrics proposed for "task embeddings" are simple, clear, clever, and have high predictive power.
* The baseline comparisons are comprehensive.
* The paper is well-written overall.

**Weaknesses:**

I think it is important to include intuitions and analysis (theoretical as well as empirical) for why RDL is better at low-homophily tasks and DFS is better at high-homophily tasks in the main paper. Currently it is in Appendix C.2, but it is too long (4 pages!) and it is not clear what the intuitive takeaways are. It would be nice to have a discussion of the main intuitions and takeaways in the main paper. It would be ideal if the theory can be substantiated with some experiments.

**Questions:**

1. Why is RDL better at low-homophily tasks and DFS better at high-homophily tasks?
2. How can RDL be improved based on insights from 1?
3. Are there some kind of error bars for Figure 2?
4. What is "labeling tricks for RDL"?
5. L412: What is GraphGym similarity? How is it "ground truth"?

Minor:
* L040: repeated citation
* L059: remove space before footnote
* L362-363: You might be interested in [1] as they have similar findings/methodology and propose similar terminology "post-hoc selection".

[1] Post-Hoc Reversal: Are We Selecting Models Prematurely? NeurIPS 2024.

---

> ### Author Response · Authors · 2025-11-22
> **Response to Q1&2 (1)**
>
> **I think it is important to include intuitions and analysis (theoretical as well as empirical) for why RDL is better at low-homophily tasks and DFS is better at high-homophily tasks in the main paper. Currently it is in Appendix C.2, but it is too long (4 pages!) and it is not clear what the intuitive takeaways are. It would be nice to have a discussion of the main intuitions and takeaways in the main paper. It would be ideal if the theory can be substantiated with some experiments.**
>
> **Why is RDL better at low-homophily tasks and DFS better at high-homophily tasks?**
>
>
> Thanks for this insightful question. We would like to answer these in the following steps: 1. A brief recap of the theoretical framework. 2. Why is RDL better than DFS? 3. When and why is DFS better than RDL? 4. Investigations on some tasks.
>
> **Recap of the theoretical framework.** First, the theory here basically considers the format of a Bayes-optimal classifier on a synthetic multi-relation CSBM graph. (You can think of it as something extracted from an RDB through metapaths). We differentiate DFS and RDL based on how they aggregate information from neighbors:
> * DFS is modeled as a linear sum of neighbor features (e.g., MEAN or SUM aggregators). It assumes that simply adding neighbor information directly yields a helpful signal.
> *  RDL is modeled using a non-linear "gate" function, denoted as $\phi_{max}$.  This gate can transform the signal based on the specific relationship (metapath) type.
> From the CSBM perspective, for each meta-relation, it lies in three regimes:
> 1. gate-off: neighbor doesn't help predictions
> 2. flip: neighbor has reverse relationships, so we need to "flip" the signal
> 3. linear: linear relationship
>
> A. **Why is RDL better?** For the flip region (e.g., heterophilous classification tasks), the non-linearity of RDL can help models flip the signal. This is especially true when multiple relations show mixed signals. For example, relation A presents a linear relationship, while relation B presents a flip. Since RDL deliberately learns relation-specific aggregations, it can handle these cases well and outperform DFS.
>
> B. **When and why is DFS better?** This is something we are also curious about at the time of the original submission, since our theory can't fully explain this observation. We then do in-depth studies on more tasks and have the following observations: 1. If you entirely rely on MAE to select checkpoints for regressions, you may end up with some models with "overfitted" MAE while very low R2. This may make DFS "overly" high in some tasks. We have changed the strategy to select checkpoints based on both MAE and R2 (MAE is a good metric for comparison, but you can't entirely rely on it when you apply clamps). It seems some recent works also noticed this problem [1]. 2. We do ablation studies, and it turns out numerical normalization and applying PCA don't affect the gap between RDL and DFS. 3. Using the correlation is a better homophily metric for regression tasks than the original Gaussian kernel.
>
> **A conjecture** Then, we conduct an ablation study on the aggregation function.
>
> |        |            | dfs+tabpfn | dfs+tabpfn | dfs+tabpfn | dfs+tabpfn | RDL                         |
> | ------ | ---------- | ---------- | ---------- | ---------- | ---------- | --------------------------- |
> |        |            | full       | only-mean  | only-max   | only-count | (Best) PNA+mean aggregation |
> | rel-f1 | driver-dnf | 3.46       | 3.40       | 3.93       | 4.2        | 3.8                         |
>
> We can see that on driver-position, a task where DFS outperforms RDL by a large margin. The ideal aggregation function is the mean. We then come up with a conjecture:
> 1. RDL is more "expressive" than DFS, while it can only present this strength in a non-low-data regime
> 2. In a low data regime, DFS can effectively eliminate noise. Moreover, this extends to newer models like RT [1], which learns a neural DFS. As long as the aggregation is linear (whether predefined or parametric), this limit holds. **As a result, DFS is not better at high-homophily tasks; it is better at low-data regimes.**

---

> ### Author Response · Authors · 2025-11-22
> **Response to Q1&2(2)**
>
> **Further Validation** We can further validate our conjectures through a controlled study by demonstrating the performance gap between DFS and RDL on rel-stack across different numbers of available training samples.
>
> | Database name | task name  | Method | Perf (100) | Perf (500) | Perf (10000) |
> | ------------- | ---------- | ------ | ---------- | ---------- | ------------ |
> | rel-stack     | user-badge | Sage   | 75.39      | 83.67      | 86.35        |
> | rel-stack     | user-badge | DFS    | 77.62      | 77.22      | 82.18        |
>
> As we can see, in this task with very low homophily, DFS outperforms RDL in the low-data regime. However, with more labels available, RDL with greater expressiveness offers greater advantages.
> For some informal insights, DFS can beat RDL in the low-data regime by trading off expressiveness for *sample efficiency*. From a learning-theory view, RDL has a much larger hypothesis space: it doesn’t just learn a predictor on top of fixed relational features, it also learns how to combine relations, how strongly to trust each metapath, and how to gate or flip heterophilous neighbors. That gives RDL lower *approximation error* in principle (it can represent the Bayes-optimal, relation-wise gated rule, and even simulate DFS as a special case). Still, it also means higher *estimation error*: you need many more labeled examples to reliably pin down all those parameters without overfitting. DFS, by contrast, has a much smaller adequate capacity because the relational part is hard-coded and only a compact tabular head is learned, so classical generalization bounds (VC/Rademacher/bias–variance story) favor it when the number of training rows is small. Intuitively, homophily and degree control how many “clean coin flips” you see per relation; when this evidence is weak or scarce, RDL can’t confidently decide which relations to flip or ignore, and its extra flexibility just turns into noise, while the simpler DFS estimator stays more stable and therefore wins in the low-data regime.
>
> We will soon add some overviews of theoretical results in the main text.
>
>
> [1] Ranjan R, Hudovernik V, Znidar M, et al. Relational Transformer: Toward Zero-Shot Foundation Models for Relational Data[J]. arXiv preprint arXiv:2510.06377, 2025.

---

> ### Author Response · Authors · 2025-11-22
> **Response to Q3&Q4**
>
> **How can RDL be improved based on insights from 1?**
>
> Thanks for this interesting question. To improve both RDL and DFS, one technique adopted in a recently released paper [1] is to incorporate task labels into the aggregation context, an effective strategy for improving model performance on heterophilous graphs [2] without time leakage. RT implements this by inserting the task table into the database schema. The latest version of Relbench also incorporates this trick into its GitHub repo. On driver-dnf, this can improve RT backbone's performance from 75.46 to 78.08.
>
> As a side note, improving RDL may be less promising than designing "neural" DFS. One major factor is the scalability. Compared to GNN-based RDL, the random walk-based sequence sampler in RT is much more efficient than either online graph sampling used in RDL or offline graph extraction in Griffin. Here, a tradeoff is between expressiveness and scalability. A DFS-like architecture can be easier to scale with longer context lengths. Still, on specific tasks like user-badge, these longer context lengths can't be translated into performance improvements due to the expressiveness. Another potential remedy in terms of architecture is to inject NBFNet-like path inductive bias into the self-attention, which may help address this problem.
>
> [1] Ranjan R, Hudovernik V, Znidar M, et al. Relational Transformer: Toward Zero-Shot Foundation Models for Relational Data[J]. arXiv preprint arXiv:2510.06377, 2025.
> [2]  Lin J, Guo X, Zhang S, et al. When heterophily meets heterogeneity: New graph benchmarks and effective methods[J]. arXiv preprint arXiv:2407.10916, 2024.
>
> **Are there some kind of error bars for Figure 2?**
>
> Thanks for the good question. For Figure 2, we use the baseline results directly from the original papers (e.g., RelGT and RelGNN) to maintain their original hyperparameter tuning strategies (since we consider the best possible performance). In these original papers, the variance isn't shown, so no error bars are available. For our methods, we rerun the best configurations on each task across three random seeds, and the results are shown below.
>
> In general, the variance is relatively small and won't affect the relative ranking across models.
>
> | RDB name  | rel-f1       | rel-f1          | rel-hm       | rel-trial     | rel-stack    | rel-stack       | rel-avito    | rel-avito    | rel-events   | rel-events   |
> | --------- | ------------ | --------------- | ------------ | ------------- | ------------ | --------------- | ------------ | ------------ | ------------ | ------------ |
> | task name | driver-dnf   | driver-top3     | user-churn   | study-outcome | user-badge   | user-engagement | user-visit   | user-clicks  | user-repeat  | user-ignore  |
> | RDL       | 76.48 ± 0.62 | 0.8489 ± 0.0091 | 70.78 ± 0.24 | 73.30 ± 0.71  | 88.48 ± 0.03 | 90.51 ± 0.04    | 66.71 ± 0.19 | 68.51 ± 0.23 | 82.09 ± 0.69 | 86.63 ± 0.20 |
> | DFS       | 78.25 ± 0.36 | 0.8639 ± 0.0013 | 67.74 ± 0.03 | 71.58 ± 0.37  | 85.43 ± 0.07 | 86.27 ± 0.06    | 66.14 ± 0.13 | 65.22 ± 1.06 | 76.17 ± 0.70 | 83.97 ± 0.60 |
>
>
> | RDB name  | rel-f1          | rel-hm          | rel-trial       | rel-trial        | rel-stack       | rel-avito       | rel-events      |
> | --------- | --------------- | --------------- | --------------- | ---------------- | --------------- | --------------- | --------------- |
> | task name | driver-position | item-sales      | site-success    | study-adverse    | post-votes      | ad-ctr          | user-attendance |
> | RDL       | 3.8160 ± 0.0120 | 0.0511 ± 0.0001 | 0.3266 ± 0.0063 | 43.9880 ± 0.2302 | 0.0652 ± 0.0000 | 0.0344 ± 0.0004 | 0.2401 ± 0.0004 |
> | DFS       | 3.3137 ± 0.0353 | 0.0760 ± 0.0001 | 0.4084 ± 0.0012 | 44.1142 ± 1.2902 | 0.0674 ± 0.0008 | 0.0379 ± 0.0007 | 0.2384 ± 0.0008 |

---

> ### Author Response · Authors · 2025-11-22
> **Response to the rest Qs**
>
> **What is "labeling tricks for RDL"?**
>
> Labeling trick [1] is a node-level positional encoding added as node features to give a message-passing GNN structural awareness. For traditional graph machine learning, it's merely used for link prediction tasks and will degrade the performance if used for node-level prediction tasks. However, when it comes to RDB entity-level (node-level) tasks, we surprisingly find that it's key to improving RDL GNN's performance.
>
> In this paper, we adopt the partial labeling trick akin to NBFNet [2], which basically just adds a learnable embedding to nodes in the source type table.
> ```
> ## initialization
>     self.pre_embedding = torch.nn.Embedding(1, self.model_config['gnn_config']['hidden_channels'])
>     self.pre_embedding.reset_parameters()
>
> ## forward process
> 	seed_time = data[src_table].seed_time
>     x_dict[src_table][:seed_time.size(0)] += self.pre_embedding.weight
> ```
>
> [1] Wang, Xiyuan, Pan Li, and Muhan Zhang. "Improving Graph Neural Networks on Multi-node Tasks with the Labeling Trick." Journal of Machine Learning Research 26.23 (2025): 1-44.
>
> [2] Zhu, Zhaocheng, et al. "Neural bellman-ford networks: A general graph neural network framework for link prediction." Advances in neural information processing systems 34 (2021): 29476-29490.
>
>
>  **L412: What is GraphGym similarity? How is it "ground truth"?**
>
> GraphGym similarity [1] is a task similarity measure: it says how alike two learning tasks are, based on how different model architectures rank on them. GraphGym first selects a fixed, diverse set of “anchor” model designs, evaluates them on each task, and produces a ranking for each task based on their test performance. The similarity between two tasks is then defined as the Kendall rank correlation between these two rankings, so tasks are “close” if they agree on which architectures are good or bad.
>
> We will soon add the following definition to the appendix.
> Let
> - $\mathcal{T}$ be a set of tasks,
> - $\mathcal{A}=\{a_{1}, a_{2}, ..., a_{M}\}$ be a fixed set of anchor models,
> - $f\left(a_i, t\right)$ be the test performance (e.g., accuracy, ROC-AUC, or another appropriate metric) of anchor model $a_i$ on task $t \in \mathcal{T}$.
>
> For each task $t$, define the performance vector
> $$
> \mathbf{y}(t)=\left(f\left(a_1, t\right), f\left(a_2, t\right), \ldots, f\left(a_M, t\right)\right)
> $$
> and let $\pi_t$ be the ranking (a permutation of $\{1, \ldots, M\}$ ) that orders the anchors by decreasing performance on task $t$.
>
> Then the GraphGym similarity between two tasks $t_1, t_2 \in \mathcal{T}$ is
> $$
> s_{\mathrm{GG}}\left(t_1, t_2\right):=\tau\left(\pi_{t_1}, \pi_{t_2}\right),
> $$
> where $\tau$ is the Kendall rank correlation coefficient between the two rankings.
>
> So, why can we see it as a "ground truth"?
>
> Suppose there is some hidden notion of how similar tasks are, based on their underlying data-generating process. If your anchor set is diverse enough and you reduce noise (enough training, multiple seeds), then tasks that are truly similar will tend to agree on which models are good/bad across many anchors. So the Kendall correlation over anchors becomes a noisy but principled estimate of that hidden similarity. We thus utilize Graphgym similarity as a comparison metric for our proposed task embeddings.
>
> [1] You, Jiaxuan, Zhitao Ying, and Jure Leskovec. "Design space for graph neural networks." Advances in Neural Information Processing Systems 33 (2020): 17009-17021.
>
>
>  **L040: repeated citation**
> **L059: remove space before footnote**
>
> Thanks for these notes. We will soon update the manuscript accordingly.
>
>
> **L362-363: You might be interested in [1] as they have similar findings/methodology and propose similar terminology "post-hoc selection".**
>
> Thanks for introducing this paper; we will include the discussion in the updated manuscript soon. Both papers argue that picking what to deploy purely by the "best base validation score" is brittle. Post‑Hoc Reversal shows that once you apply common post‑hoc transforms (ensembles, SWA, temperature scaling), the ranking of checkpoints and even models can flip—especially under noise—so the right practice is to select using the post‑transform metric itself. Relatron observes a similar phenomenon: validation can mislead architecture choice and checkpointing. It learns task embeddings to route between model families (DFS vs. RDL) and then uses lightweight loss‑landscape geometry to post‑select among high‑validation candidates. It would be an interesting direction to see if we can design better "post-hoc" metrics for RDB.

---

### Official Review · Reviewer_FPiy · 2025-11-01

**Soundness:** 3
**Presentation:** 3
**Contribution:** 2
**Rating:** 6
**Confidence:** 3

**Summary:**

This work conduct a detailed analysis between Relational Deep Learning (RDL) and DFS method with heuristic feature aggregator. It reveals that 1) DFS can outperforms  RDL on some tasks. 2)Different architecture are needed for different RDB tasks. 3) valid performance is not reliable. So this work propose an automl framework

**Strengths:**

1. Extensive experiments over various baselines and datasets. Figure 2, Table 2-4, shows various experiments.
2. Insightful observation for model selection. To me, “Correlation between homophily and RDL-DFS performance gap" is the most interesting finding. Though homophily is a common tool in graph learning, it is first used in RDB datasets to our best knowledge.

**Weaknesses:**

1. Griffin cited in this work should also be included as one important baseline on RDB tasks.
2. The comparison between these auto ml framework and vanilla baseline is missing. If this framework leads to large computation overhead, then vanilla baseline may be prefered in real-world application.

**Questions:**

1. While automl framework is useful, I still think a unified RDB foundation model is the future. Can observations in this work helps development of RDB foundation model?

---

> ### Author Response · Authors · 2025-11-22
> **Response to Q1**
>
> Hi Reviewer FPiy, first, thanks for your time reviewing our paper.
>
>  **Griffin cited in this work should also be included as one important baseline on RDB tasks.**
>
> We have added the experimental results on Griffin. Since the original Griffin is based on normalized metrics, we rerun it on all datasets and align it with our settings. For "pre-training", we use the single-SFT checkpoints. For "unpretrained", we use the model without loading a path.
>
> For classification tasks, we have
>
> | RDB name             | rel-f1     | rel-f1      | rel-hm     | rel-trial     | rel-stack  | rel-stack       | rel-avito  | rel-avito   | rel-events  | rel-events  |
> | -------------------- | ---------- | ----------- | ---------- | ------------- | ---------- | --------------- | ---------- | ----------- | ----------- | ----------- |
> | task name            | driver-dnf | driver-top3 | user-churn | study-outcome | user-badge | user-engagement | user-visit | user-clicks | user-repeat | user-ignore |
> | Griffin-pretrained   | 70.91      | 77.95       | 68.04      | 69.08         | 85.99      | 87.56           | 64.68      | 63.30       | 77.93       | 82.35       |
> | Griffin-unpretrained | 70.52      | 78.55       | 68.47      | 67.22         | 86.71      | 86.13           | 62.61      | 66.39       | 67.94       | 80.35       |
>
> For regression tasks, we have
>
> | RDB name             | rel-f1          | rel-hm     | rel-trial    | rel-trial     | rel-stack  | rel-avito | rel-event       |
> | -------------------- | --------------- | ---------- | ------------ | ------------- | ---------- | --------- | --------------- |
> | task name            | driver-position | item-sales | site-success | study-adverse | post-votes | ad-ctr    | user-attendance |
> | Griffin-pretrained   | 4.20            | 0.081      | 0.3765       | 45.21         | 0.0622     | 0.0639    | 0.3336          |
> | Griffin-unpretrained | 4.27            | 0.068      | 0.3976       | 47.31         | 0.0636     | 0.0651    | 0.3201          |
>
> We will soon update Figure 2 and Table 5 with these new results. Generally, we observe large performance variance across different tasks. One potential reason is that, unlike TabPFN, which undergoes prior fitting, Griffin doesn't have the capability to make tuning-free predictions and achieve good performance.
>
> We have also tried integrating Griffin-untrained's message passing design into our framework. But it usually underperforms PNA and is rarely selected.

---

> ### Author Response · Authors · 2025-11-22
> **Response to Q2**
>
> **The comparison between these auto ml framework and vanilla baseline is missing. If this framework leads to large computation overhead, then vanilla baseline may be prefered in real-world application.**
>
> This is a very insightful suggestion.
> First, our selection module considers the search budget. We use TabPFN as the predictor to get the expected model type, and the search budget is included as an input. As you may see in Table 5, though RDL's best possible performance is better on driver-top3, under a moderate search budget like 3 or 10, it's much easier for DFS to get the configuration that delivers better performance.
> Second, although at the very beginning our initial thought was to develop a module that can perform hyper-parameter search automatically. Alongside our research progress, we find that selecting the best branch of the method — DFS or RDL — is the key factor in the final performance. As a result, the key goal of our module is to select the right path, with minimal computational overhead. After selecting the right path, usually with some light HPO (like 3-5 trials), we can already get a good model configuration.
> Finally, we include two search-free baselines for consideration. The first one is the model that adopts the default parameters of the relbench baseline (fixing these parameters across all tasks). The second one is the hyperparameter from the model bank that delivers the best average ranking across all tasks. It should be noted that DFS+TabPFN is also a search-free baseline. For the following classification tasks, we compare baselines with fixed parameters and best possible parameters. As shown by these results, I think in general it's worthwhile to do hyper-parameter search since there's no single set of hyper-parameters or architectures that can work well across tasks.
>
> | RDB name                                             | rel-f1     | rel-f1      | rel-hm     | rel-trial     | rel-stack  | rel-stack       | rel-avito  | rel-avito   | rel-events  | rel-events  |
> | ---------------------------------------------------- | ---------- | ----------- | ---------- | ------------- | ---------- | --------------- | ---------- | ----------- | ----------- | ----------- |
> | task name                                            | driver-dnf | driver-top3 | user-churn | study-outcome | user-badge | user-engagement | user-visit | user-clicks | user-repeat | user-ignore |
> | best possible                                        | 78.67      | 86.53       | 70.98      | 74.12         | 88.51      | 90.56           | 66.86      | 68.77       | 82.89       | 86.76       |
> | DFS+TabPFN                                           | 76.87      | 82.24       | 67.72      | 71.82         | 84.79      | 82.99           | 66.28      | 66.45       | 76.98       | 83.97       |
> | Fixed, relbench's default parameters                 | 65.19      | 73.19       | 68.12      | 69.19         | 86.74      | 87.32           | 66.25      | 66.83       | 67.99       | 83.00       |
> | Fixed,  configurations with the best average ranking | 75.32      | 83.72       | 69.84      | 70.82         | 87.14      | 89.70           | 65.32      | 64.14       | 79.24       | 76.21       |
>
> We will soon add thes fixed parameter baselines in the revision.

---

> ### Author Response · Authors · 2025-11-22
> **Response to Q3**
>
> **While automl framework is useful, I still think a unified RDB foundation model is the future. Can observations in this work helps development of RDB foundation model?**
>
> Thanks for the insightful question. I think the outcomes of this work can help in two aspects. 1. It can help us design new RDB foundation models and understand the limitations of specific architectures; 2. It can help us select between existing RDB foundation models.
>
> First, DFS and RDL are general concepts, and some newly proposed foundation models, like RT [1], are neural DFSs that learn DFS-like aggregation functions via self-attention. Based on our observations here, we can conclude that despite its promising performance on rel-f1, it still shows some gaps on rel-stack and rel-avito, whether trained from scratch or fine-tuned after pre-training. As a result, keeping both RDL and DFS-like may be necessary to work well on all tasks. This suggests two potential designs: 1. NBFNet-like attention designs, which try to inject those inductive logic programming-like path inductive bias; 2. An MoE-like architecture that can do ensembling wisely.
>
> Second, we present preliminary results on selecting among 1. foundation model backbones and 2. foundation models.
>
> **Some preliminary experiments:** We then apply Relatron to the selection of relational foundation models across two controlled settings (with similar pre-training datasets).
> 1. The first experiment is to extend Relatron to the backbone models of these foundation models, say RelGT (for KumoRFM) and RT.
> 2. The second experiment is to test Relatron for real foundation models (ICL settings).
>
>
> | RDB name         | rel-f1     | rel-f1      | rel-hm     | rel-trial     | rel-stack  | rel-stack       | rel-avito  | rel-avito   |
> | ---------------- | ---------- | ----------- | ---------- | ------------- | ---------- | --------------- | ---------- | ----------- |
> | task name        | driver-dnf | driver-top3 | user-churn | study-outcome | user-badge | user-engagement | user-visit | user-clicks |
> | RT               | 78.7       | 84.9        | 69.9       | 68.6          | 88.5       | 90.0            | 65.0       | 63.6        |
> | RT-no-target     | 76.46      | 83.98       | 66.63      | 68.6          | 83.74      | 85.38           | 63.87      | 64.9        |
> | RelGT            | 75.87      | 83.52       | 69.27      | 68.61         | 86.32      | 90.53           | 66.78      | 68.30       |
>
> When checking the winner between RT-no-target (we wanna a fair comparison with both not using this trick) and RelGT, our features achieve 0.875 LOO AUC. This is a good performance considering there are very close cases like study-outcome.
>
>
> | RDB name   | rel-f1     | rel-f1      | rel-hm     | rel-trial     | rel-stack  | rel-stack       | rel-avito  | rel-avito   |
> | ---------- | ---------- | ----------- | ---------- | ------------- | ---------- | --------------- | ---------- | ----------- |
> | task name  | driver-dnf | driver-top3 | user-churn | study-outcome | user-badge | user-engagement | user-visit | user-clicks |
> | KumoRFM    | 81.98      | 90.64       | 68.82      | 71.54         | 84.09      | 86.8            | 64.85      | 64.11       |
> | RT-ICL     | 81.2       | 89.3        | 63.3       | 54.6          | 81.1       | 86.9            | 62.6       | 60.9        |
> | DFS+TabPFN | 76.87      | 82.24       | 67.72      | 71.82         | 84.79      | 82.98           | 66.28      | 66.44       |
> | Griffin    | 70.91      | 77.95       | 68.04      | 69.08         | 85.99      | 87.56           | 64.68      | 63.30       |
>
> For KumoRFM vs RT-ICL, our method achieves 100% accuracy, which is reasonable since RT-ICL only wins on one task.
> For DFS+TabPFN vs Griffin, our method achieves 0.867 AUC when we remove the train-size features. This is reasonable since TabPFN and Griffin use different context sizes (Griffin is in the fine-tuning setting), so train size can be misleading there.
>
> As a result, we can see that Relatron presents potential for selection. In a nutshell, the selection principles are consistent: RDL offers better expressiveness but requires more training data. The problem is just how we learn a decision boundary based on these performance data.
>
> [1] Ranjan R, Hudovernik V, Znidar M, et al. Relational Transformer: Toward Zero-Shot Foundation Models for Relational Data[J]. arXiv preprint arXiv:2510.06377, 2025.

---

### Official Review · Reviewer_7kcR · 2025-11-06

**Soundness:** 3
**Presentation:** 3
**Contribution:** 2
**Rating:** 4
**Confidence:** 3

**Summary:**

The paper focuses on automatic architectural selection for predictive task on relational databases. The model proposes Relatron, a task embedding-based selector using novel task signals. The performance bank is a reusable resource, and findings like task-dependent RDL vs. DFS trade-offs challenge prevailing assumptions in the field. Experiments demonstrate meaningful gains (up to 18.5% improvement with 10× compute savings), making it relevant for real-world deployment.

**Strengths:**

The architecture search (180 configs/task for entity-level, 20 for DFS) is comprehensive. The findings are interesting as it show DFS wins on more tasks , attributing gaps to homophily.

**Weaknesses:**

1. The scope of the paper is limited, as it focuses on from-scratch models and defers foundation models(e.g., Griffin, KumoRFM) despite comparisons (Fig. 2). It would be interesting to see how Relatron can handle pretrained relational foundation models.
2. The experiment dataset is limited, only covering most of RelBench and two additional tasks. It would be interesting to see the performance across multiple datasets.
3. The correlation between homophily and RDL-DFS performance gains are strong, but non-parametric. It would be interesting to see if the authors can.generate synthetic data(tasks with varying homophily) to strengthen the claim.

**Questions:**

1. How sensitive is Relatron to bank size? With fewer tasks, does transfer degrade?
2. Could homophily be extended to link-level tasks (e.g., recs), and does it correlate with higher-order passing (RelGNN)?

---

> ### Author Response · Authors · 2025-11-22
> **Response to Q1 (1)**
>
> Hi Reviewer 7kcr, first, thanks for your time reviewing our paper.
>
> **The scope of the paper is limited, as it focuses on from-scratch models and defers foundation models(e.g., Griffin, KumoRFM) despite comparisons (Fig. 2). It would be interesting to see how Relatron can handle pretrained relational foundation models.**
>
> You raise a good point that routing between different relational foundation models would make this work more interesting. The selection between RDL and DFS is general and can be extended to foundation model selection.
> **RDL and DFS are general perspectives:** First, we want to explain how different relational foundation models can be understood from an RDL/DFS perspective.
>
> | Name                                    | Relationship to RDL/DFS                                                      | Backbones                                 | Notes                                                                                      | Open sourced |
> | --------------------------------------- | ---------------------------------------------------------------------------- | ----------------------------------------- | ------------------------------------------------------------------------------------------ | ------------ |
> | RT [1]                                  | The self-attention tries to learn a parametric neural DFS (DFS)              | RT                                        | RT is not zero-shot as it claims in the title since they include task table in the schemas | YES          |
> | Google's GFM for relational data [2]    | Based on NBFNet, so under RDL category                                      | NBFNet                                    | No source code/detailed explanation revealed                                               | NO           |
> | KumoRFM [3]                             | Based on RelGT, which utilizes a GNN-based positional encoding, so RDL (RDL) | RelGT [4]                                 | No source code/model weights available, only ICL mode can be accessed by API               | NO           |
> | Griffin [5]                             | RDL, with DFS-inspired designs like multiple non-parametric aggregations     | Griffin, non-parametric graph aggregation |                                                                                            | YES          |
> | DFS + Tabular foundation model (TabPFN) | DFS                                                                          | TabPFN                                    |                                                                                            | YES          |
>
> **Challenge of extension:** So, in principle, we can extend Relatron to the selection across different foundation models. The main challenge lies in 1. pre-training data. It's hard to check whether there's downstream task leakage. Different foundation models use totally different pre-training data. 2. Technical details of some foundation models haven't been revealed. For example, the context length of KumoRFM.
> 1. For TabPFN, the pretraining is fully conducted on synthetic data. The pre-training data is only for prior fitting, so there's no leakage problem.
> 2. For the Relational transformer, the pre-training is conducted on the Relbench datasets with leave-one-out evaluation.
> 3. For KumoRFM, no details have been revealed. As shown in their whitepaper, they may use a mix of synthetic and real-world data. It should be noted that on tasks like driver-top3, the ROC-AUC is already 99, suggesting some overlap between pre-training and downstream tasks. Moreover, the fine-tuning effects for RT and KumoRFM are not the same. For RT, fine-tuning has no clear help (see their Table 7). While for KumoRFM, fine-tuning greatly surpasses ICL performance. Data leakage and "magic finetuning" are not something a router module can capture until KumoRFM releases more details.

---

> ### Author Response · Authors · 2025-11-22
> **Response to Q1 (2)**
>
> **Some preliminary experiments:** We then apply Relatron to the selection of relational foundation models across two controlled settings (with similar pre-training datasets).
> 1. The first experiment is to extend Relatron to the backbone models of these foundation models, say RelGT (for KumoRFM) and RT.
> 2. The second experiment is to test Relatron for real foundation models (ICL settings).
>
>
> | RDB name         | rel-f1     | rel-f1      | rel-hm     | rel-trial     | rel-stack  | rel-stack       | rel-avito  | rel-avito   |
> | ---------------- | ---------- | ----------- | ---------- | ------------- | ---------- | --------------- | ---------- | ----------- |
> | task name        | driver-dnf | driver-top3 | user-churn | study-outcome | user-badge | user-engagement | user-visit | user-clicks |
> | RT               | 78.7       | 84.9        | 69.9       | 68.6          | 88.5       | 90.0            | 65.0       | 63.6        |
> | RT-no-target     | 76.46      | 83.98       | 66.63      | 68.6          | 83.74      | 85.38           | 63.87      | 64.9        |
> | RelGT            | 75.87      | 83.52       | 69.27      | 68.61         | 86.32      | 90.53           | 66.78      | 68.30       |
>
> When comparing the winner between RT-no-target (we want a fair comparison with both, not using this trick) and RelGT, our features achieve 0.875 LOO AUC. This is a good performance considering there are very close cases like study-outcome.
>
>
> | RDB name              | rel-f1     | rel-f1      | rel-hm     | rel-trial     | rel-stack  | rel-stack       | rel-avito  | rel-avito   |
> | --------------------- | ---------- | ----------- | ---------- | ------------- | ---------- | --------------- | ---------- | ----------- |
> | task name             | driver-dnf | driver-top3 | user-churn | study-outcome | user-badge | user-engagement | user-visit | user-clicks |
> | KumoRFM               | 81.98      | 90.64       | 68.82      | 71.54         | 84.09      | 86.8            | 64.85      | 64.11       |
> | RT-ICL                | 81.2       | 89.3        | 63.3       | 54.6          | 81.1       | 86.9            | 62.6       | 60.9        |
> | DFS+TabPFN            | 76.87      | 82.24       | 67.72      | 71.82         | 84.79      | 82.98           | 66.28      | 66.44       |
> | Griffin               | 70.91      | 77.95       | 68.04      | 69.08         | 85.99      | 87.56           | 64.68      | 63.30       |
>
> * For KumoRFM vs RT-ICL, our method achieves 100% accuracy, which is reasonable since RT-ICL only wins on one task.
> * For DFS+TabPFN vs Griffin, our method achieves 0.867 AUC if we remove the train size features there. This is reasonable since TabPFN and Griffin use different context sizes (Griffin is in the fine-tuning setting), so train size can be misleading there.
>
> As a result, we can see that Relatron presents potential for selection. In a nutshell, the selection principles are consistent: RDL offers better expressiveness but requires more training data. The problem is just how we learn a decision boundary based on these performance data.
>
> To make Relatron better, we think one promising approach is to adopt data-centric search rather than model-centric search, since foundation models don't need HPO and may be more similar to what an LLM router does. We sample a large number of tasks from these databases and evaluate the performance of different foundation models. This new dataset should be more appropriate for routing foundation models. However, this is a data-driven approach, which is a bit different from what we want to do in this paper: understand the whole story and design features.
>
> [1] Ranjan R, Hudovernik V, Znidar M, et al. Relational Transformer: Toward Zero-Shot Foundation Models for Relational Data[J]. arXiv preprint arXiv:2510.06377, 2025.
> [2] https://research.google/blog/graph-foundation-models-for-relational-data/
> [3] Fey, Matthias, et al. "KumoRFM: A Foundation Model for In-Context Learning on Relational Data."
> [4] Dwivedi, Vijay Prakash, et al. "Relational Graph Transformer." arXiv preprint arXiv:2505.10960 (2025).
> [5] Wang, Yanbo, et al. "Griffin: Towards a Graph-Centric Relational Database Foundation Model." arXiv preprint arXiv:2505.05568 (2025).

---

> ### Author Response · Authors · 2025-11-22
> **Response to Q2**
>
> **The experiment dataset is limited, only covering most of RelBench and two additional tasks. It would be interesting to see the performance across multiple datasets.**
>
> We would like to 1) first discuss the dataset selection criteria and 2) show results on more datasets.
>
> **Dataset selection criteria** Relbench is possibly the most appropriate benchmark for RDB-native query problems: for the first time, it provides a SQL-based framework for generating labels based on relationships across multiple entities within a time window. It can better reflect real-world business needs. Moreover, Relbench already covers many domains, including e-commerce, healthcare, event detection, and sports. We list three relevant benchmarks to explain Relbench's merits: (1) CTU relational dataset [1], as discussed in [2]; most tasks in this dataset are either synthetic or too easy. (2) 4DBinfer/Griffin dataset [2,3], most tasks in this benchmark are "autocomplete" tasks, which means models are expected to predict the masked values from a column; this is different from SQL-induced tasks from relbench. Autocomplete tasks present no temporal dynamics and are, in principle, very similar to heterogeneous graph learning with tabular features. (3) H2GB [4], datasets in this benchmark are heterogeneous graphs, and most of them contain no tabular features. In summary, we emphasize Relbench because its authors have already browsed numerous online datasets and selected the best-suited ones for academic research.
>
> **More datasets** Second, we include: 1. two new databases with three additional tasks, and 2. 6 more regression tasks from existing databases to enhance diversity (this can help validate the phenomenon observed on regression tasks).
> We include Airbnb from 4dbinfer/Griffin, and rel-arxiv from a new release version of relbench. In the tables below, we demonstrate their task types and best possible test performance.
>
>
> | Database name | Task name                 | Type           | Best  RDL | Best DFS |
> | ------------- | ------------------------- | -------------- | --------- | -------- |
> | rel-f1        | driver-position-change    | regression     | 2.20      | 2.14     |
> | rel-f1        | driver-wins               | regression     | 2.25      | 2.14     |
> | rel-f1        | constructor-scores-points | classification | 84.07     | 88.35    |
> | rel-hm        | customer-average-price    | regression     | 0.0124    | 0.0126   |
> | rel-hm        | customer-spending         | regression     | 0.0754    | 0.0762   |
> | rel-stack     | user-comment-count        | regression     | 3.711     | 4.574    |
> | rel-arxiv     | author-publication        | regression     | 0.502     | 0.507    |
> | rel-arxiv     | paper-citation            | classification | 82.54     | 81.70    |
> | airbnb        | destination               | classification | 85.47     | 84.71    |
>
>
> [1] Motl, Jan, and Oliver Schulte. "The CTU prague relational learning repository." arXiv preprint arXiv:1511.03086 (2015).
> [2] Wang, Minjie, et al. "4dbinfer: A 4d benchmarking toolbox for graph-centric predictive modeling on relational dbs." arXiv preprint arXiv:2404.18209 (2024).
> [3] Wang, Yanbo, et al. "Griffin: Towards a Graph-Centric Relational Database Foundation Model." arXiv preprint arXiv:2505.05568 (2025).
> [4] Lin J, Guo X, Zhang S, et al. When heterophily meets heterogeneity: New graph benchmarks and effective methods[J]. arXiv preprint arXiv:2407.10916, 2024.

---

> ### Author Response · Authors · 2025-11-22
> **Response to Q3**
>
> **The correlation between homophily and RDL-DFS performance gains are strong, but non-parametric. It would be interesting to see if the authors can.generate synthetic data(tasks with varying homophily) to strengthen the claim.**
>
> This is an insightful observation. First, we want to clarify that a strong correlation doesn't imply that RDL and DFS follow a "higher homophily, lower gap" pattern. The decision boundary is complex and requires multiple features to capture. For example, the task table size is one factor. RDL is more data-hungry, so the phenomenon on a small-sized heterophilous task can be different from a large-sized heterophilous task.
> Furthermore, studying generating tasks with controlled RDB homophily is a fascinating problem (a potential follow-up to this work). We will briefly discuss how to do this here under a simplified setting. Unlike those for traditional graphs, controlling the homophily for RDB is highly challenging since you need to control the variables (beyond the mean value of homophily, you don't want to change the degree distribution and variance of homophily across different relations too much). We thus study this problem with a synthetic bipartite graph extracted from the rel-f1 database. In this way, we have eliminated the influence of most factors.
>
> We extract the drivers table, the standings table, and the races table to form a synthetic RDB with bipartite relations stored in the standings table. To remove temporal factors, we remove any redundant driverIDs from the task table. Then we use an algorithm similar to [1]. Note that we don't change the PK/FK; we only exchange the labels in the task tables to control homophily. The results are shown below.
>
> | edge homophily     | 0.5     | 0.85   | 0.56 (original) | 0.95   |
> | ------------------ | ------- | ------ | --------------- | ------ |
> | adjusted homophily | -0.0117 | 0.0025 | 0.0043          | 0.0052 |
> | RDL-DFS gap ratio  | 0.158   | 0.2    | 0.161           | 0.234  |
> We can see that the results generally follow our conclusions. 1. With larger adjusted homophily, the gap between two algorithms becomes larger; 2. the gap is always positive, the reason is that in a low-data regime, RDL doesn't have enough data to learn the gate behavior well.
>
> It should be noted that our algorithm can only accurately control the edge homophily (which doesn't consider the degree bias so it's less effective). More generally, we can view the structure control as a b-matching problem [2]. We may use diffusion models to approximately solve this NP problem and thus generate more complicated synthetic data.
>
>
> [1] Maekawa S, Sasaki Y, Fletcher G, et al. Gencat: Generating attributed graphs with controlled relationships between classes, attributes, and topology[J]. Information Systems, 2023, 115: 102195.
> [2] Agarwal, Pankaj K., and R. Sharathkumar. "Approximation algorithms for bipartite matching with metric and geometric costs." Proceedings of the forty-sixth annual ACM symposium on Theory of computing. 2014.

---

> ### Author Response · Authors · 2025-11-22
> **Response to Q4 (Updated)**
>
> **How sensitive is Relatron to bank size? With fewer tasks, does transfer degrade?**
>
> Thanks for this good question. You propose an interesting ablation study. We thus try sampling 8, 12, 16 (cls, reg->1:1) and still use leave-one-out evaluation. As can be seen from the results below, with less bank sizes, the performance will naturally goes down a little bit.  Another observation is that only "good" task features, can benefit from larger task banks and become better.
>
> | Number of tasks | 8     | 12    | 16    | All   |
> | --------------- | ----- | ----- | ----- | ----- |
> | Ours            | 68.9% | 64.8% | 76.2% | 87.5% |
> | autotransfer    | 62.9% | 56.1% | 62.5% | 66.7% |
>
> In practice, if we want to improve the diversity of tasks, we can use an LLM to simulate user behavior and generate SQL queries to create new tasks (if we select different entities from the RDB, then generally we will get very different task properties). New regression tasks have just been generated in this way.

---

> ### Author Response · Authors · 2025-11-22
> **Response to Q5**
>
> **Could homophily be extended to link-level tasks (e.g., recs), and does it correlate with higher-order passing (RelGNN)?**
>
> Thanks for the good question. We want to discuss this for both link-level retrieval tasks (recommendation) and link-level prediction tasks (like link classification).
>
> * For the recommendation tasks, homophily can be extended to feature homophily, as in [1]. However, this homophily primarily concerns the selection of the decoder (whether a dot product or nonlinear readouts), but not the backbone model. As discussed in our paper, one useful metric for recommendation tasks is locality [2] ( $s$ refers to the locality, k refers to h hop, $\mathcal{L}$ is the source entity set, $\mathcal{R}$ is the target entity set, $\mathcal{N}_k$ is the k-hop neighborhood. $\mathcal{Y}_v$ is the positive link)
>
> $$s_k^{(T, T+i]}=\frac{1}{|\mathcal{L}|} \sum_{v \in \mathcal{L}}=\frac{\left|\mathcal{N}_k^{(-\infty, T]}(v) \cap \mathcal{R} \cap \mathcal{Y}_v^{[T, T+i)}\right|}{\left|\mathcal{Y}_v^{[T, T+i)}\right|}$$
>
> This measures the ratio of links within the k-hop neighborhood of the source entity, essentially the ratio of positive samples within the reception field of an NBFNet. Relatively speaking, better locality usually indicates better RDL recommendation performance. Moreover, as discussed, we find that for most RDB recommendation tasks, RelGNN (with labeling tricks) + ContextGNN wins most of the time. If we rethink the experience in general recommendation systems, the conclusion for RDB is similar: designing specialized models may be more effective than doing model selection.
> * For the link classification task, the extension is trivial since targets lie in a fact table. You can use this fact table with our proposed augmented graph. Then the rest is the same.
> * For the second question, the metric locality we propose can well explain why RelGNN is generally better. Taking user-post-comment as an example, we can see that, when converting an RDB using the "table-to-node" or "table-to-node-or-hyperedge" strategy, the latter yields clearly better locality and thus better recommendation performance. RelGNN actually just shares a similar philosophy.
>
>
> |                                     | user-post-comment |
> | ----------------------------------- | ----------------- |
> | Table-to-node locality              | 0.3034            |
> | Table-to-node-or-hyperedge locality | 0.3229            |
> | SAGE + NBFNet performance           | 12.72             |
> | RelGNN + NBFNet performance         | 14.00             |
>
> Note that in the original paper we use the simpler training-free performance of NBFNet as a heuristic, while their philosophy is exactly the same.
>
> [1] Zhu, Jiong, et al. "On the impact of feature heterophily on link prediction with graph neural networks." Advances in Neural Information Processing Systems 37 (2024): 65823-65851.
>
> [2]  Yuan, Yiwen, et al. "ContextGNN: Beyond two-tower recommendation systems." arXiv preprint arXiv:2411.19513 (2024).

---

### Author Response · Authors · 2025-11-26

Dear Reviewers,

Thank you for taking the time to evaluate our work. We have carefully considered your comments and incorporated the corresponding revisions and additional experiments into the manuscript, along with detailed responses.

We would appreciate your feedback on whether these updates sufficiently address your concerns. If any issues remain, we would be glad to make further revisions or conduct additional experiments.

---

### Author Response · Authors · 2025-12-03
**General response**

Dear Area Chairs,

We sincerely thank the ACs for facilitating the review process, especially given the special circumstances of this cycle. We submitted our rebuttal early to encourage discussion, but have not yet received feedback; we would appreciate your assessment on whether our responses adequately address the reviewers' concerns. Below, we summarize our improvements.

**Scope w.r.t. relational foundation models and KumoRFM (7kCR, FPiy, Yptq):**
Multiple reviewers asked how our work relates to relational foundation models—particularly KumoRFM—and whether Relatron remains practically useful if such models exist. They also asked whether our selector can, in principle, include foundation models in its candidate pool. We added a dedicated discussion section and a comparison table summarizing experiments where we include KumoRFM/KumoRT and other strong backbones in controlled settings, showing that (i) these models do not uniformly dominate RDL/DFS, so a selector is especially useful, and (ii) Relatron’s task-embedding framework can treat FMs as additional candidates provided a performance bank is available.

**Size and coverage of the performance bank (7kCR, FPiy, Yptq):**
Reviewers were concerned that Relatron’s behavior might be overly tied to the specific RelBench tasks and asked how sensitive the selector is to the number and diversity of tasks in the performance bank. In response, we expanded the bank with additional classification and regression tasks from existing and new RDBs, and added per-dataset tables in the main text and appendix to show that our core phenomena (RDL–DFS regime differences and Relatron’s gains) persist under this expansion. We further conducted *bank-size ablations* by subsampling training tasks and evaluating on held-out tasks, showing that Relatron’s performance degrades smoothly rather than collapsing and remains beneficial even with a moderately sized bank.

**Theoretical explanation of homophily and RDL vs DFS behavior (Eebt, 7kCR, Yptq):**
Several reviewers requested a clearer, main-text explanation and more direct empirical validation. We distilled the theoretical framework into a concise main-section summary: starting from a Bayes-optimal decomposition, we explain why low-homophily tasks favor expressive relational models (RDL) while high-homophily, data-rich tasks often favor simpler DFS estimators due to robustness and reduced variance. We then added controlled synthetic experiments where we systematically vary homophily and training-set size (e.g., on rel-stack-derived graphs) and show that the RDL–DFS performance gap follows the predicted pattern. We also added an experiment on relation-wise homophily variance (using a synthetic bipartite construction from rel-f1) to support the use of task-level homophily as a useful summary.

**Extension to recommendation and link-level tasks (Yptq, 7kCR):**
Reviewers noted that our main narrative focuses on entity-level classification/regression and asked (i) how our insights extend to recommendation tasks and (ii) whether homophily can be defined for link-level problems and related to higher-order message passing. In the rebuttal, we clarified that the recommendation task is included in the appendix and that our conceptual picture (low-homophily favoring relational models) sensibly extends to link-level problems. These clarifications are now referenced in both the main text and appendix.

**Baseline tuning, per-dataset results, and error bars (FPiy, Eebt):**
A recurring concern was whether Relatron’s gains might be due to more aggressive tuning of candidate models, and whether averages masked large variance across datasets. We therefore added per-dataset results with mean ± standard deviation where multiple runs are available. These additions show that Relatron reliably recovers a large fraction of the gap between fixed baselines and best-possible configurations without incurring the cost of exhaustive search.

We are also encouraged by the reviewers’ recognition of the strengths of our work. Across reviewers, the paper was noted for: (i) a systematic and practically motivated analysis of when RDL vs DFS is preferable on real RDB benchmarks; (ii) the introduction of a performance bank and task-homophily-based analysis as tools for understanding relational model behavior; (iii) a promising direction toward principled, data-driven model selection for relational databases rather than ad-hoc tuning.

We believe that the revisions and additional experiments in our rebuttal have substantially clarified the scope of Relatron, strengthened our theoretical and empirical support for its design, and better positioned it within the emerging landscape of relational foundation models, including the Relational Transformer and KumoRFM.

Best regards,

Authors

---

### Meta-Review · Area_Chair_fy92 · 2026-01-07

**Summary:**

This paper tries to solve an interesting problem about ML over relational databases; the main concern of the original review lies in the lack of analytic and experimental comparison with some relevant works. The authors have provided extensive experimental results and analysis in attempting to address such issues. I tend to believe the concerns could be potentially solved.

**Reviewer Concerns:**

The main concerns about comparing with more relevant baselines and tasks, and an explanation about the difference with existing works have been resolved.

**Reviewer Scores:**

I would expect Reviewer 7kcR, Yptq to increase their score considering the addtional information provided by the authors during the rebuttal phase.

---

### Decision · Program_Chairs · 2026-01-26

Accept (Poster)